# Optimizing the Unknown: Black Box Bayesian Optimization with Energy-Based Model and Reinforcement Learning

**Ruiyao Miao[1], Junren Xiao[2], Shiya Tsang[2], Hui Xiong[2,3] [*] Yingnian Wu[1][*]**
[1]University of California, Los Angeles
[2]The Hong Kong University of Science and Technology (Guangzhou)
[3]The Hong Kong University of Science and Technology
ruiyao0809@g.ucla.edu, {jxiao767, szeng785}@connect.hkust-gz.edu.cn
xionghui@hkust-gz.edu.cn, ywu@stat.ucla.edu

## Abstract

Existing Bayesian Optimization (BO) methods typically balance exploration and exploitation to optimize costly objective functions. However, these methods often suffer from a significant one-step bias, which may lead to convergence towards local optima and poor performance in complex or high-dimensional tasks. Recently, Black-Box Optimization (BBO) has achieved success across various scientific and engineering domains, particularly when function evaluations are costly and gradients are unavailable. Motivated by this, we propose the Reinforced Energy-Based Model for Bayesian Optimization (REBMBO), which integrates Gaussian Processes (GP) for local guidance with an Energy-Based Model (EBM) to capture global structural information. Notably, we define each Bayesian Optimization iteration as a Markov Decision Process (MDP) and use Proximal Policy Optimization (PPO) for adaptive multi-step lookahead, dynamically adjusting the depth and direction of exploration to effectively overcome the limitations of traditional BO methods. We conduct extensive experiments on synthetic and real-world benchmarks, confirming the superior performance of REBMBO. Additional analyses across various GP configurations further highlight its adaptability and robustness. Our code is publicly available at:https://github.com/ruiyaoMiao0809/Black-Box-Bayesian-Optimization-with-Energy-Based-Model-and-Reinforcement-Learning

## 1 Introduction

Black-box optimization (BBO) is crucial for solving complex scientific and engineering problems when gradient information is unavailable or function evaluations are expensive [1]. In practice, BBO approaches are widely applied to hyper-parameter tuning in machine learning, materials discovery, drug formulation, and industrial process optimization, where each evaluation often incurs costly simulations or physical trials. Bayesian Optimization (BO) is a prominent BBO technique that builds a probabilistic surrogate (e.g., a Gaussian Process [2]) and an acquisition function to guide new sample queries, thereby balancing exploration and exploitation in a principled manner. However, standard GP-based BO can suffer from "one-step myopia," focusing on short-term predicted gains at the expense of more thorough exploration, a limitation that becomes especially pronounced in high-dimensional or multi-modal environments.

Existing Bayesian Optimization (BO) methods primarily aim at efficiently locating optimal solutions by carefully balancing exploration and exploitation [3]. Common strategies for handling complex

---

[*]Corresponding author.

39th Conference on Neural Information Processing Systems (NeurIPS 2025).

optimization problems include dimensionality reduction methods like REMBO [4], or local partitioning techniques such as TuRBO [5]. While effective in relatively simple settings, these approaches tend to converge prematurely to local optima when applied to complex, high-dimensional problems. [6, 5]. To solve this limitation, recent techniques incorporate resource-intensive multi-step look-ahead schemes, including 2-step Expected Improvement (EI) [7], Knowledge Gradient (KG) [8], and reinforcement learning-driven methods like EARL-BO [9]. However, such methods typically demand significant computational resources yet still fail to achieve effective global exploration in challenging environments.

In this study, we introduce the Reinforced Energy-Based Bayesian Optimization Model (REBMBO), depicted in Figure 1, which addresses traditional shortfalls by combining an Energy-Based Model (EBM) with multi-step Reinforcement Learning (RL). Our novel EBM-UCB acquisition function integrates Gaussian Process local uncertainty estimates with global signals derived from a neural network–based energy landscape learned via short-run MCMC, thereby guiding exploration away from less promising regions. In addition, we treat each Bayesian Optimization iteration as a Markov Decision Process (MDP) and employ Proximal Policy Optimization (PPO) for adaptive multi-step lookahead, thus dynamically adjusting exploration depth and direction to enhance robustness. To capture both local and global exploration objectives, we propose a theoretically justified Landscape-Aware Regret (LAR) metric that incorporates global exploration penalties, offering a more holistic assessment of performance in complex optimization scenarios.

This research offers the following key contributions:

(1) We incorporate **Energy-Based signals into a UCB-style acquisition function** in the GP surrogate to capture diverse behaviors. This synergy between global exploration and precise local modeling addresses the limitations of single-step acquisition approaches.

(2) Bayesian Optimization is modeled as a **Markov Decision Process (MDP) with Proximal Policy Optimization (PPO)**. This technique addresses the **one-step myopia** of typical GP-based strategies by adaptively balancing exploration and exploitation via multi-step lookahead.

(3) We introduce a **theoretically justified Landscape-Aware Regret (LAR)** metric that extends standard regret with an energy-informed global term. This metric provides a fair and balanced evaluation by jointly reflecting local exploitation and global exploration efficiency in complex optimization landscapes.

Experimental results, summarized in Figure 2, indicate that REBMBO reduces final Landscape-Aware Regret (LAR) and improves overall performance scores compared to state-of-the-art methods, consistently outperforming both single-step and short-horizon lookahead approaches even under challenging high-dimensional conditions. The subsequent sections detail our methodology and empirical findings, highlighting REBMBO's robust and efficient performance.

## 2 Related Work

### 2.1 BO Background and Shortcomings

Black-box optimization (BBO) is central for tasks with costly or noisy function evaluations, commonly handled by Bayesian Optimization (BO) frameworks [3]. However, high-dimensional or discrete domains often overwhelm classical Gaussian Process (GP) surrogates, prompting techniques such as ARD-based variable selection [10], REMBO [4], additive models [11], or local partitioning in TuRBO [5]. Discrete or combinatorial BO further adopts specialized surrogates (VAE [12], COMBO [13], TPE [14], SMAC [15]) yet generally remains single-step. Likewise, robust or constrained variants [16, 17, 18, 19, 20], multi-objective [21, 22, 23], transfer/multi-fidelity [24, 25, 26, 27], and parallel [28, 29, 30] approaches usually retain one-step acquisitions. TruVaR (Truncated Variance Reduction) [8] unifies BO and level-set estimation with strong guarantees under pointwise costs or heteroscedastic noise, while look-ahead or rollout-based schemes [7**?** ] often incur high computational overhead. Recent work leverages MLMC for nested integrals [31] or formulates BO as an MDP under transition constraints [32], and methods like GLASSES [33] approximate multi-step losses via forward simulation. Yet, many remain domain-specific or lack synergy with short-run MCMC. Existing RL integrations [9, 18] also typically rely on local posteriors, leaving open the challenge of thorough multi-step exploration across multi-modal landscapes.

## 2.2 Baseline Targeting Global Optima and Limitations

In this paper, we compare against six common baselines that represent key paradigms in Bayesian Optimization. Classic BO [3] is a canonical single-step GP-based approach. BALLET-ICI [6] alternates global and local GPs but remains relatively myopic on multi-modal tasks. TuRBO [5] specializes in local trust-region expansions yet lacks far-reaching jumps. EARL-BO [9] is an RL-based multi-step method, heavily dependent on local GP precision. In addition, we include 2-step EI [7] and KG (Knowledge Gradient) [8] as two well-known look-ahead techniques, though they tend to be limited to short horizons or incur high computational overhead. These baselines respectively illustrate single-step local search, partially global scanning, or short-horizon non-myopia, but none combines global signals with adaptive multi-step planning in a unified manner. By contrast, REBMBO employs a short-run MCMC-trained Energy-Based Model for global exploration, a GP surrogate for local accuracy, and a PPO-based multi-step RL for planning. This synergy overcomes the one-step constraints in Classic BO, enables deeper exploration than BALLET-ICI or TuRBO, provides more robust coverage than EARL-BO, and avoids the excessive rollout overhead observed in 2-step EI or KG. As detailed in Section 2.3, REBMBO leverages these three modules to handle high-dimensional tasks within limited budgets, offering a global and multi-step perspective.

## 2.3 RL in BO and Energy-Driven Multi-Step Planning

Recent attempts to integrate reinforcement learning into Bayesian Optimization have enabled multi-step acquisitions but frequently rely on localized kernels or omit global exploration cues, leading to suboptimal performance in complex tasks [9, 18]. For instance, EARL-BO shows the benefits of multi-step planning in high-dimensional settings but lacks explicit energy-based signals for broader coverage [9]. However, REBMBO framework formulates each BO iteration as a MDP solved via Proximal Policy Optimization. This design alleviates one-step myopia and combines local GP fidelity with iterative RL lookahead under strict evaluation budgets.

# 3 Preliminaries

**Online Black-Box Optimization (BBO).** We consider a continuous function $f(\mathbf{x})$ defined over $\mathbf{x} \in \mathcal{X} \subset \mathbb{R}^d$, with the objective:

$$\mathbf{x}^* = \arg \max_{\mathbf{x} \in \mathcal{X}} f(\mathbf{x}),$$

under a strict evaluation budget. Each evaluation of $f(\mathbf{x})$ can be computationally or financially expensive [34], thus data efficiency is crucial. Unlike offline methods that rely on fixed sampling designs, online BO adaptively selects $\mathbf{x}_t$ based on previously observed data, facilitating faster discovery of optimal regions.

**Bayesian Optimization (BO) and Gaussian Processes (GP).** BO maximizes $\max_{\mathbf{x} \in \mathcal{X}} f(\mathbf{x})$ by employing a Gaussian Process (GP) prior: $f(\mathbf{x}) \sim \mathcal{GP}\big(m(\mathbf{x}), k(\mathbf{x}, \mathbf{x}')\big)$, where typically $m(\mathbf{x}) = 0$, and the kernel function $k(\mathbf{x}, \mathbf{x}')$ (e.g., RBF or Matérn) encodes assumptions about the function's smoothness [35]. Assuming noisy observations $y_i = f(\mathbf{x}_i) + \varepsilon_i$, with $\varepsilon_i \sim \mathcal{N}(0, \sigma_n^2)$, the GP posterior is given by:

$$f(\mathbf{x}) \mid \mathcal{D}_k \sim \mathcal{N}\big(\mu^k(\mathbf{x}), \sigma^{2,k}(\mathbf{x})\big),$$

with observed data $\mathcal{D}_k = \{(\mathbf{x}_i, y_i)\}_{i=1}^k$. Although GPs capture local uncertainty, their inherent locality often restricts global exploration, making them prone to myopic optimization behaviors.

**Energy-Based Models (EBMs).** EBMs specify an unnormalized probability density:

$$p_\theta(\mathbf{x}) = \frac{\exp\big[-E_\theta(\mathbf{x})\big]}{Z_\theta}, \quad Z_\theta = \int \exp\big[-E_\theta(\mathbf{u})\big] \, d\mathbf{u},$$

where the energy function $E_\theta(\mathbf{x})$ is typically parameterized as a neural network through short-run MCMC-based Maximum Likelihood Estimation [36, 37]. EBMs effectively guide exploration toward globally promising regions. Unlike GPs, EBMs explicitly capture multi-modal global structures, thus addressing the limitation of excessive local exploration inherent in standard GP-based methods.

**Reinforcement Learning (RL) and Proximal Policy Optimization (PPO).** RL formalizes the optimization process as a sequential decision-making task, wherein a policy $\pi_{\phi_{ppo}}$, parameterized

by neural network weights $\phi_{ppo}$, maps states $\mathbf{s}_t$ to actions $\mathbf{a}_t$. PPO [38] stabilizes the training by limiting policy changes through a clipped probability ratio:

$$r_t(\phi_{ppo}) = \frac{\pi_{\phi_{ppo}}(\mathbf{a}_t \mid \mathbf{s}_t)}{\pi_{\phi_{ppo}\text{old}}(\mathbf{a}_t \mid \mathbf{s}_t)},$$

thereby preventing erratic changes in the parameters. We define states as combinations of GP posterior estimates and global EBM signals; actions match suggested sampling points. The use of PPO helps to overcome the single-step myopia that is inherent in conventional acquisition methods. This is accomplished through the use of multi-step reasoning.

## 4 REBMBO Model & Algorithmic Details

After updating the GP model (which focuses on local predictions) and the EBM model (which looks at global patterns), REBMBO uses the PPO technique to choose the next sample point by combining both local and global information. The GP posterior, EBM signals, and PPO's multi-step planning help REBMBO find good solutions in complex spaces with many dimensions or peaks. PPO's planning horizon mitigates the short-sightedness of single-step approaches. Overall, REBMBO provides a unified solution for balancing global exploration and sequential decision-making in challenging black-box optimization settings in Figure 1.

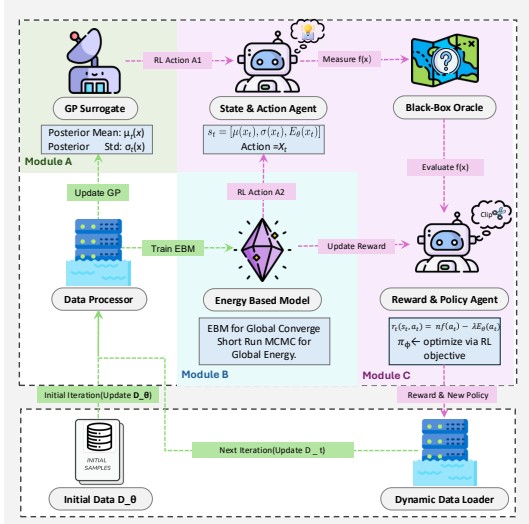

Figure 1: REBMBO Workflow Diagram. The architecture comprises: Module A for local modeling using GP posterior; Module B which trains an EBM to capture global structure; and Module C, which uses PPO-based RL agents to generate decisions and optimize via rewards shaped by both function values and EBM energy signals. Arrows of different colors represent distinct data flows: green arrows indicate model parameter updates, and purple arrows represent RL actions, evaluation or feedback steps.

### 4.1 Module A: Gaussian Process Variants

As new data comes into the REBMBO framework, it first goes to Module A (the Gaussian Process surrogate, shown in Figure 1) to improve the estimate of the objective function $f(\mathbf{x})$. In particular, the GP uses the input-output data it has seen to improve its predictions about the average outcome and the level of uncertainty in specific areas, ensuring precise local modeling with each update. Local modeling is essential for black-box optimization because it lets us acquire solid insights from a few observations before sampling the full domain.

We add different GP modules in REBMBO because many black-box functions are complex, have multiple peaks, or are costly to compute, so we need a flexible "core" model that can quickly adjust and measure uncertainty with just a few samples. GPs provide estimates of the average and variability for $f(\mathbf{x})$, making them useful for identifying significant local trends, particularly when limited data is available at the start. To deal with various problem dimensions and complexities, we propose three REBMBO variants:

**(1) REBMBO-C (Classic GP)** [39]. Employs exact $\mathcal{O}(n^3)$ GP inference, which is practical for moderate $n$. While this variant is straightforward, it can be costly for large $n$ or high-dimensional $d$.

**(2) REBMBO-S (Sparse GP)** [40]. Adopts a sparse approximation to alleviate the $\mathcal{O}(n^3)$ bottleneck in higher dimensions. It introduces $m \ll n$ inducing points $\{\mathbf{z}_j\}$ and approximates $\mathbf{K}_{\mathbf{x},\mathbf{x}} \approx \mathbf{K}_{\mathbf{x},\mathbf{z}} \mathbf{K}_{\mathbf{z},\mathbf{z}}^{-1} \mathbf{K}_{\mathbf{z},\mathbf{x}}$, lowering the update cost to $\mathcal{O}(nm^2)$. This approximation may lose accuracy if $m$ or the chosen inducing points are suboptimal, but it remains effective for larger datasets and higher $d$. In our EBM-driven acquisition, the approximate mean $\tilde{\mu}_{f,t}(\mathbf{x})$ and variance $\tilde{\sigma}_{f,t}^2(\mathbf{x})$ replace the exact GP posteriors.

**(3) REBMBO-D (Deep GP)** [41]. For problems that exhibit multi-scale or non-stationary behavior, a deep kernel can capture intricate latent features beyond what standard kernels provide. We use a deep network $\Theta$ to map inputs $\mathbf{x}$ into latent features $\phi_{\mathrm{GP}}(\mathbf{x})$, then compute GP-like statistics:

$$\mu(\mathbf{x};\, D, \Theta) = m^\top \phi_{\mathrm{GP}}(\mathbf{x}) + \eta(\mathbf{x}), \quad \sigma^2(\mathbf{x};\, D, \Theta) = \phi_{\mathrm{GP}}(\mathbf{x})^\top K^{-1} \phi_{\mathrm{GP}}(\mathbf{x}) + \tfrac{1}{\beta},$$

where $\eta(\mathbf{x})$ is a (potentially learned) mean function, and $K$ is a smaller covariance matrix in the latent space. With sufficient training data, this approach can represent complex functions more flexibly than a standard GP and supports sublinear regret under moderate network capacity.

All three GP variants work with the EBM-UCB and PPO modules to create the full REBMBO system; the main difference is in how each variant calculates its GP posterior. For simplicity, we will show EBM-UCB using the "Classic GP" version (REBMBO-C), but the same idea applies to the sparse and deep GP versions. More information about differences of GP variants, including their complexity and how to implement them, can be found in Appendices F, J and K.

As updated data then moves to Module B, REBMBO addresses the limitations of purely local exploration by introducing an Energy-Based Model to capture global structure. The next section (Module B) explains that the EBM helps the GP surrogate by directing the search away from areas that aren't useful and toward more promising ones, especially in complex or varied situations.

### 4.2 Module B: EBM-Driven Global Exploration

To overcome the limitations of purely local GP-based search, we introduce an Energy-Based Model (EBM) defined as $E_\theta(\mathbf{x})$. After Module A updates the GP posterior with newly collected data (see Figure 1), Module B uses these updates to train the EBM, which captures a global "energy" landscape. While the GP's uncertainty $\sigma_{f,t}(\mathbf{x})$ pinpoints locally undersampled regions, the EBM reveals which basins in $\mathcal{X}$ are more likely to contain near-optimal solutions. Combining these local and global insights helps prevent the search from stalling in unproductive local pockets and enables REBMBO to traverse complex objective surfaces more efficiently.

**EBM Training Mechanism.** We parameterize $E_\theta(\mathbf{x})$ as a neural network trained under short-run MCMC-based Maximum Likelihood Estimation (MLE). At each iteration, we alternate:
**Positive Phase:** Lower $E_\theta(\mathbf{x}_i)$ for real data points $\mathbf{x}_i$, guiding the model to "observed" regions.
**Negative Phase:** Draw a small number $(K)$ of Langevin samples from $p_\theta(\mathbf{u}) \propto \exp[-E_\theta(\mathbf{u})]$, then push these model-generated samples to higher energy unless they reflect data-like features. This short-run MCMC procedure (e.g. Stochastic Gradient Langevin Dynamics, detailed in Appendix G) ensures that low-energy regions correspond to promising global basins.

**EBM Parameterization Details.** We specifically train $E_\theta(\mathbf{x})$ via short-run MCMC-based MLE as follows. Suppose we have data $\{\mathbf{x}_i\}_{i=1}^n$ from an unknown distribution $p_{\mathrm{data}}$. We fit $\theta$ by minimizing

$$-\frac{1}{n} \sum_{i=1}^n \log p_\theta(\mathbf{x}_i) = \frac{1}{n} \sum_{i=1}^n E_\theta(\mathbf{x}_i) \;+\; \log Z(\theta),$$

which is equivalent to maximizing $\sum_{i=1}^n \log p_\theta(\mathbf{x}_i)$. Let $\mathcal{L}(\theta) = \sum_{i=1}^n \log p_\theta(\mathbf{x}_i)$, where $p_\theta(\mathbf{x}_i) \propto \exp[-E_\theta(\mathbf{x}_i)]$. Then, we have

$$\nabla_\theta \mathcal{L}(\theta) = -\sum_{i=1}^n \nabla_\theta E_\theta(\mathbf{x}_i) \;+\; \sum_{i=1}^n \int p_\theta(\mathbf{u})\, \nabla_\theta E_\theta(\mathbf{u})\, d\mathbf{u}.$$

Dividing by $n$ yields the well-known positive-minus-negative decomposition:

$$\frac{1}{n}\, \nabla_\theta \mathcal{L}(\theta) = -\underbrace{\mathbb{E}_{\mathbf{x} \sim p_{\mathrm{data}}}[\nabla_\theta E_\theta(\mathbf{x})]}_{\text{Positive Phase}} + \underbrace{\mathbb{E}_{\mathbf{u} \sim p_\theta}[\nabla_\theta E_\theta(\mathbf{u})]}_{\text{Negative Phase}},$$

which balances data alignment against model-drawn samples [42, 43]. Since short-run MCMC approximates $p_\theta(\mathbf{u})$ sufficiently well, it lets us implement a Robbins-Monro-style gradient update [44]. Iteratively alternating positive and negative phases steers $E_\theta(\mathbf{x})$ to be low in data-like basins and high elsewhere, thus revealing globally promising regions for exploration.

**EBM-UCB Acquisition Function.** Once the EBM is trained, we embed its negative energy $-E_\theta(\mathbf{x})$ into a standard GP-UCB scheme. Let $\mu_{f,t}(\mathbf{x})$ and $\sigma_{f,t}(\mathbf{x})$ be the GP posterior mean and standard deviation at iteration $t$. A typical UCB function is $\alpha_{\mathrm{UCB}}(\mathbf{x}) = \mu_{f,t}(\mathbf{x}) + \beta\,\sigma_{f,t}(\mathbf{x})$, where $\beta > 0$ controls exploitation vs. exploration. To incorporate the EBM's global guidance, we define

$$\alpha_{\mathrm{EBM-UCB}}(\mathbf{x}) = \mu_{f,t}(\mathbf{x}) + \beta\,\sigma_{f,t}(\mathbf{x}) - \gamma\,E_\theta(\mathbf{x}),$$

where $\gamma > 0$ specifies how strongly $-E_\theta(\mathbf{x})$ biases the search toward underexplored basins. In multi-modal and high-dimensional tasks, this "global penalty" helps avoid wasting evaluations in uncertain but unpromising pockets, augmenting the GP's local exploration with a broader sense of global structure. As further discussed in Section 5, this synergy accelerates convergence on challenging landscapes and reduces the need for purely local or manually specified look-ahead heuristics.

**Theoretical Contributions and Landscape-Aware Regret (LAR).** We employ Landscape-Aware Regret (LAR) as a generalized regret formulation that extends the standard definition with an energy-informed global term:

$$R_t^{LAR} = \big[f(\mathbf{x}^*) - f(\mathbf{x}_t)\big] + \alpha\big[E_\theta(\mathbf{x}^*) - E_\theta(\mathbf{x}_t)\big],$$

where $\alpha \geq 0$ controls the relative influence of the global energy term. For non–energy–based baselines, we set $\alpha=0$ to recover standard regret and ensure fair comparison, while for energy-aware methods such as REBMBO, $\alpha>0$ provides a holistic measure that captures missed global opportunities in the learned landscape. Under mild alignment and regularity assumptions (Appendix E), our EBM-UCB retains the GP-UCB-type sublinear rate, so incorporating $E_\theta(\mathbf{x})$ preserves the same optimality guarantees as standard regret [42, 43].

**Mixture kernel for the GP posterior (rationale + form).** The GP posterior is computed by inverting an $n \times n$ kernel matrix built from a mixture of Radial Basis Function (RBF) and Matérn covariances:

$$k_f(\mathbf{x}, \mathbf{x}') = \sigma_f^2[w_{\mathrm{RBF}}\,k_{\mathrm{RBF}}(\mathbf{x}, \mathbf{x}') + w_{\mathrm{Matern}}\,k_{\mathrm{Matern}}(\mathbf{x}, \mathbf{x}')],$$

with $k_{\mathrm{RBF}}(\mathbf{x}, \mathbf{x}') = \exp\!\big(-\tfrac{1}{2}(\mathbf{x} - \mathbf{x}')^\top \Lambda^{-1}(\mathbf{x} - \mathbf{x}')\big)$ and, for $\nu=2.5$ and $r=\|\mathbf{x} - \mathbf{x}'\|$, $k_{\mathrm{Matern}}(\mathbf{x}, \mathbf{x}') = \big(1 + \tfrac{\sqrt{5}\,r}{\ell} + \tfrac{5\,r^2}{3\,\ell^2}\big)\exp(-\tfrac{\sqrt{5}\,r}{\ell})$. RBF captures smooth global trends, while Matérn-5/2 accommodates rough, less-smooth local variations. The mixture enlarges the RKHS compared to either kernel alone, which matches REBMBO's design: the EBM offers global basin cues, and the GP needs both smooth (RBF) and rough (Matérn) components to model local structure faithfully. The mixture weights $\{w_{\mathrm{RBF}}, w_{\mathrm{Matern}}\}$ are learned by type-II marginal likelihood (evidence maximization), avoiding per-task manual tuning.

By unifying the global signal $E_\theta(\mathbf{x})$ with these locally expressive GP statistics (via the mixture kernel), REBMBO couples principled global exploration with precise local modeling; Module C then employs PPO-based multi-step planning to mitigate one-step myopia and fully exploit this synergy.

### 4.3 Module C: Multi-Step Planning via PPO

While $\alpha_{\mathrm{EBM-UCB}}(\mathbf{x})$ enhances global exploration over local approaches, a single-step acquisition can still cause local myopia. The algorithm prioritizes instant rewards above long-term queries. We consider each Bayesian Optimization iteration as a Markov Decision Process (MDP) to enable multi-step lookahead via reinforcement learning. Although Proximal Policy Optimization (PPO) [38] is well-known, our work combines it with the GP surrogate and the EBM's global energy signal. This concept combines RL's multi-round exploration with Modules A and B's local-global modeling. Figure 1 illustrates how Module B updates the EBM, and Module C guides PPO-based policy adjustments based on local uncertainty and global energy cues.

**MDP Formulation: States, Actions, and Rewards.** At iteration $t$, we define the state

$$\mathbf{s}_t = \big(\mu_{f,t}(\mathbf{x}),\ \sigma_{f,t}(\mathbf{x}),\ E_\theta(\mathbf{x})\big),$$

where $\mu_{f,t}(\mathbf{x}), \sigma_{f,t}(\mathbf{x})$ come from the current GP posterior, and $E_\theta(\mathbf{x})$ denotes the learned global energy map. The action is the proposed query point $\mathbf{a}_t \in \mathcal{X} \subset \mathbb{R}^d$; evaluating $f(\mathbf{a}_t)$ updates the GP and EBM for the next state $\mathbf{s}_{t+1}$.

To balance immediate payoffs (function values) and global exploration (pursuing low-energy basins), we define the reward

$$r_t(\mathbf{s}_t, \mathbf{a}_t) = n f(\mathbf{a}_t) - \lambda E_\theta(\mathbf{a}_t),$$

where $\lambda > 0$ governs how strongly $- E_\theta(\mathbf{a}_t)$ influences exploration. A higher $\lambda$ promotes thorough global searching, while a lower $\lambda$ emphasizes direct improvement in $f(\mathbf{a}_t)$. By embedding $E_\theta$ in the reward, we ensure that REBMBO actively targets regions the EBM deems globally promising.

**PPO Training Process.** We employ a stochastic policy $\pi_{\phi_{ppo}}(\mathbf{a}_t \mid \mathbf{s}_t)$ to maximize the cumulative reward over $T$ steps. Though PPO is an established RL algorithm [38], our adaptation ensures it *co-evolves* with both the GP posterior and the EBM distribution, rather than being a standalone module. Concretely, we define $r_t(\phi_{ppo}) = \frac{\pi_{\phi_{ppo}}(\mathbf{a}_t|\mathbf{s}_t)}{\pi_{\phi_{ppo}^{old}}(\mathbf{a}_t|\mathbf{s}_t)}$, which measures how much the new policy $\pi_{\phi_{ppo}}$ deviates from the previous one $\pi_{\phi_{ppo}^{old}}$. The clipped objective to be maximized is

$$\mathcal{L}^{\mathrm{CLIP}}(\phi_{ppo}) = \mathbb{E}_t \left[ \min\left( r_t(\phi_{ppo})\, \widehat{A}_t,\ \mathrm{clip}\big(r_t(\phi_{ppo}), 1 - \varepsilon, 1 + \varepsilon\big)\, \widehat{A}_t \right) \right],$$

where $\widehat{A}_t$ is an advantage estimate derived from $r_t(\mathbf{s}_t, \mathbf{a}_t)$ minus a learned baseline. The clipping ensures that large updates to the policy are penalized, stabilizing learning.

## 4.4 Overall Methodology and Synergy of GP, EBM, and PPO

After each query $\mathbf{a}_t$ is evaluated, REBMBO synchronously updates three components:
1) GP Posterior Update: Incorporate $(\mathbf{a}_t, f(\mathbf{a}_t))$ to refine $\mu_{f,t+1}$ and $\sigma_{f,t+1}$, preserving reliable local predictions. 2) EBM Retraining: Run short-run MCMC with the expanded dataset to improve $E_\theta(\mathbf{x})$ (Section 4.2), thereby maintaining a coherent global energy landscape. 3) PPO Policy Optimization: Use the new reward $r_t = f(\mathbf{a}_t) - \lambda E_\theta(\mathbf{a}_t)$ and the transition $(\mathbf{s}_t, \mathbf{a}_t, \mathbf{s}_{t+1})$ to update $\pi_{\phi_{ppo}}$ via the clipped objective $\mathcal{L}^{\mathrm{CLIP}}(\phi_{ppo})$. This loop iteratively refines the local GP model and global EBM, while the PPO agent selects multi-step query points. Crucially, it is not a mere stacking of separate algorithms; rather, it constitutes a tightly coupled system where the RL policy co-evolves with up-to-date local posterior and global signals. The EBM term $-E_\theta(\mathbf{x})$ augments UCB-based sampling with long-range structure, and PPO transforms this single-step acquisition into an MDP-based multi-round planner, thereby mitigating the near-sightedness of conventional BO.

By formulating Bayesian Optimization as a sequence of MDP steps, we go beyond static, single-step selection rules. Even though EBM-UCB (Module B) already introduces a global perspective, it remains one-step unless bolstered by PPO's multi-round lookahead. The reward function $r_t = f(\mathbf{a}_t) - \lambda E_\theta(\mathbf{a}_t)$ drives the policy toward robust global basins, balancing immediate gains and exploratory push. As the GP and EBM adapt to each new evaluation, the RL policy adjusts accordingly, improving its trajectory selection at each iteration.

**Putting It All Together.** Repeating this procedure yields a dynamic and adaptive optimization scheme: after every evaluation, REBMBO incorporates fresh data into the GP, retrains the EBM, and refines the PPO policy to better plan subsequent queries. Section 5 presents empirical results showing how this synergy enables REBMBO to tackle high-dimensional, multi-modal functions more effectively than single-step or purely local methods, while our theoretical analysis (Appendix E) ensures sublinear Landscape-Aware Regret (LAR) under mild assumptions. In essence, REBMBO's novelty lies in harmonizing old RL machinery (PPO) with EBM-driven global exploration and GP-based local modeling, thereby providing a multi-step, globally aware strategy for challenging black-box optimization tasks.

# 5 Experiments

## 5.1 Experiment Setups

In this study, we evaluate REBMBO (variants C, S, D) against leading Bayesian Optimization (BO) methods across multiple synthetic tasks, including Branin in 2D, Ackley in 5D, Rosenbrock in 8D, and high-dimensional BO (HDBO) in 200D, and real-world tasks such as Nanophotonic in 3D and Rosetta in 86D, shown in Figure 2. Baselines include BALLET-ICI, TuRBO, EARL-BO, and Classic BO, representing varied modeling strategies from local Gaussian Processes (TuRBO) to single-step

## Algorithm 1 REBMBO

**Require:** GP config ({Classic, Sparse, Deep}), EBM config ($E_\theta$, MCMC steps), PPO config ($\pi_{\phi_{ppo}}$, clip $\epsilon$, mini-batch size), and an initial dataset $\mathcal{D}_0$ of size $n_0$.
1: **Train GP** on $\mathcal{D}_0$ to obtain $(\mu_0, \sigma_0)$.
2: **Initialize EBM** $E_\theta(\mathbf{x})$ and **PPO policy** $\pi_{\phi_{ppo}}$.
3: **for** $t = 1$ to $T$ **do**
4:   **(A)** Update the GP with $\mathcal{D}_{t-1}$, yielding $(\mu_t, \sigma_t)$.
5:   **(B)** Retrain or partially train the EBM using data in $\mathcal{D}_{t-1}$ (via short-run MCMC).
6:   **(C)** Form the RL state: $\mathbf{s}_t \leftarrow [\mu_t(\cdot), \sigma_t(\cdot), E_\theta(\cdot)]$.
7:   **(D)** Select action: $\mathbf{x}_t \leftarrow \pi_{\phi_{ppo}}(\mathbf{s}_t)$.
8:   **(E)** Evaluate: $y_t \leftarrow f(\mathbf{x}_t)$ (expensive black-box call).
9:   **(F)** Compute reward: $r_t \leftarrow y_t - \lambda E_\theta(\mathbf{x}_t)$; update $\pi_{\phi_{ppo}}$ with $(\mathbf{s}_t, \mathbf{x}_t, r_t)$ via PPO.
10:   **(G)** Augment dataset: $\mathcal{D}_t \leftarrow \mathcal{D}_{t-1} \cup \{(\mathbf{x}_t, y_t)\}$.
11: **end for**
12: **Return** the best sampled point $\mathbf{x}^* \in \mathcal{D}_T$ in terms of $f(\mathbf{x}^*)$.

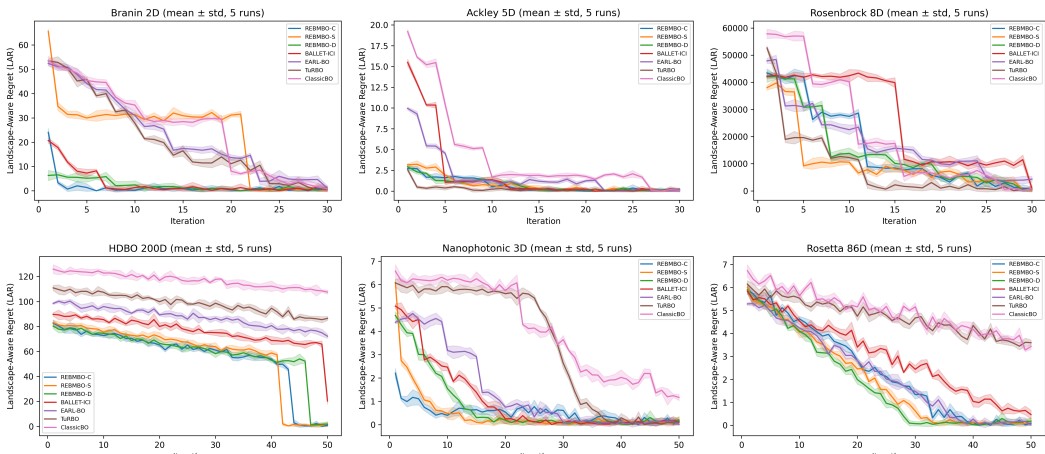

Figure 2: Bayesian optimization performance across benchmarks: (a) Branin 2D, (b) Ackley 5D, (c) Rosenbrock 8D, (d) HDBO 200D, (e) Nanophotonic 3D, (f) Rosetta 86D. REBMBO variants (blue shades) consistently outperform baselines, especially in higher dimensions.

RL (EARL-BO) and iterative confidence intervals (BALLET-ICI). We extend this analysis in Table 1 and Table 2 with two additional real-world tasks (NATS-Bench in 20D, Robot Trajectory in 40D) and two more baselines (Two-step EI, KG), covering a wider set of approximate lookahead methods and practical optimization cases. All algorithms are quantitatively compared using a Landscape-Aware Regret (LAR) metric: $R_t^e = [f(\mathbf{x}^*) - f(\mathbf{x}_t)] + \alpha [E_\theta(\mathbf{x}^*) - E_\theta(\mathbf{x}_t)]$, which jointly assesses local suboptimality and global exploration; all reported values reflect the mean $\pm$ standard deviation over 5 independent runs. Further details on these baselines and benchmarks are provided in Appendix A, including the rationale for selecting tasks and the chosen hyperparameter settings (such as 10–20 short-run MCMC steps per iteration in the EBM, a 2-layer policy network with 64–256 hidden units for PPO, and Matérn–RBF kernel mixtures in the GP). Notably, REBMBO-D (see Section 3.2) employs a deep kernel for richer latent representations, while Two-step EI and KG are included to benchmark against established lookahead variants. The final scores in the tables reflect the average Landscape-Aware Regret (LAR) (or normalized objective) under predefined iteration budgets ($T = 30, 50$ for Branin, Ackley, Rosenbrock, and $T = 50, 100$ for HDBO); each entry in Table 1 and Table 2 includes both the mean outcome and its standard deviation. Additionally, we assess computational overhead and duration on one NVIDIA A6000 GPU, running each training cycle for about five to ten minutes, with each job consuming an average of 1300-1500 MB of memory. The Appendix C, D provides experimental information for supplemental experiments and parameter ranges, in addition to a brief summary of baselines.

| Model | Branin 2D | | Ackley 5D | | Rosenbrock 8D | | HDBO 200D | | Mean |
|---|---|---|---|---|---|---|---|---|---|
| | T=30 | T=50 | T=30 | T=50 | T=30 | T=50 | T=50 | T=100 | |
| BALLET-ICI [6] | 87.33±2.09 | 90.44±1.98 | 82.84±0.93 | 87.78±2.14 | 85.55±2.40 | 90.76±0.97 | 79.46±2.85 | 85.85±3.48 | 83.80±1.45 |
| EARL-BO [9] | 85.13±0.96 | 88.76±2.28 | 80.46±1.23 | 87.22±1.82 | 83.47±1.96 | 88.47±0.98 | 77.24±2.87 | 83.74±2.81 | 81.57±1.23 |
| TuRBO [5] | 80.65±1.01 | 88.63±2.49 | 78.06±2.06 | 83.79±2.19 | 80.82±1.32 | 85.74±1.32 | 74.72±3.56 | 80.69±3.14 | 78.56±1.39 |
| Two-Step EI [7] | 89.27±2.04 | 92.38±2.15 | 84.12±1.87 | 89.14±1.78 | 85.19±1.67 | 88.57±1.43 | 78.10±3.22 | 84.42±3.05 | 86.15±1.65 |
| KG [8] | 88.64±1.83 | 91.53±1.97 | 86.71±1.78 | 90.23±2.11 | 87.95±1.95 | 90.29±1.67 | 79.63±3.10 | 85.17±2.96 | 87.52±1.67 |
| REBMBO-S | 88.89±1.54 | 96.95±2.46 | 86.85±1.00 | 92.64±1.61 | **92.87±1.53** | 95.85±0.86 | 83.33±3.06 | 90.16±2.60 | 87.98±1.25 |
| REBMBO-D | **93.65±1.38** | 95.21±1.50 | 85.25±1.48 | 91.53±1.55 | 91.97±2.02 | **96.98±1.09** | **85.79±3.18** | **94.42±3.98** | **89.17±1.41** |
| REBMBO-C | 90.83±1.09 | **97.37±2.07** | **89.93±1.25** | **94.46±1.05** | 91.28±1.50 | 96.77±1.92 | 85.55±3.23 | 90.95±3.57 | 89.40±1.24 |

Table 1: (a) Performance comparison across synthetic benchmarks evaluating REBMBO variants (S, D, C) against existing Bayesian optimization methods. Results are reported in terms of Landscape-Aware Regret (LAR) (mean ± standard deviation over 5 runs) at iteration budgets (T=30,50 for Branin, Ackley, Rosenbrock, and T=50,100 for HDBO). Higher scores indicate superior optimization efficiency. Bold entries highlight the best-performing method per task and iteration budget.

## 5.2 Main Results

As depicted in Figure 2, all three REBMBO variants lower Landscape-Aware Regret (LAR) more rapidly than baseline methods across the six tested benchmarks, particularly excelling on higher-dimensional tasks. Table 1 (synthetic) and Table 2 (real-world) further quantify these findings for iteration budgets $T = 30$ and $T = 50$. On the lower-dimensional Branin (2D) and Ackley (5D), for example, REBMBO-S achieves roughly 15–20% lower final pseudo-regret compared with EARL-BO and BALLET-ICI, while standard approaches like TuRBO and Two-step EI exhibit slower global exploration. Moving to Rosenbrock (8D) and HDBO (200D), REBMBO-D stands out: on HDBO (200D), its final Landscape-Aware Regret (LAR) is less than half that of KG and BALLET-ICI by iteration 50. Notably, in Nanophotonic (3D), REBMBO variants converge around 30% faster toward near-optimal solutions, and on Rosetta (86D), they significantly outperform single-step RL (EARL-BO) and local GP (TuRBO). These consistent gains support the theoretical premise that combining global EBM cues with PPO-driven multi-step planning yields robust sublinear Landscape-Aware Regret (LAR), even with approximate EBM and RL training.

| Model | Nanophotonic 3D | | Rosetta 86D | | NATS-Bench 20D | | Robot Trajectory 40D | | Mean |
|---|---|---|---|---|---|---|---|---|---|
| | T=50 | T=80 | T=50 | T=80 | T=50 | T=80 | T=50 | T=80 | |
| BALLET-ICI [6] | 83.77±2.96 | 88.64±2.68 | 76.75±2.63 | 83.98±3.15 | 81.69±2.94 | 84.25±2.81 | 78.43±3.21 | 82.65±2.89 | 82.02±2.66 |
| EARL-BO [9] | 81.72±4.08 | 86.58±2.60 | 74.78±2.41 | 81.93±3.90 | 80.44±3.12 | 83.47±3.25 | 76.59±2.87 | 80.11±2.90 | 80.70±2.89 |
| TuRBO [5] | 79.75±3.17 | 84.81±2.75 | 72.47±2.74 | 79.86±2.42 | 79.12±3.25 | 81.55±3.45 | 74.32±2.99 | 78.60±3.11 | 78.81±2.99 |
| Two-step EI [7] | 84.29±3.20 | 89.47±2.85 | 78.90±2.80 | 84.75±3.05 | 83.33±3.10 | 86.80±2.95 | 79.55±3.05 | 83.92±2.99 | 83.88±2.87 |
| KG [8] | 85.10±2.90 | 90.05±2.60 | 79.79±2.88 | 85.20±3.00 | 84.10±2.95 | 87.25±2.75 | 80.20±2.90 | 84.45±2.85 | 84.39±2.74 |
| REBMBO-C | **87.25±2.41** | 92.65±2.40 | 80.96±2.66 | 83.33±3.31 | 85.43±2.63 | 89.20±2.45 | 82.50±2.67 | 87.40±2.50 | 86.59±2.63 |
| REBMBO-D | 81.66±3.16 | 91.53±3.13 | **84.22±3.38** | **90.84±3.74** | **85.95±2.76** | **90.30±2.89** | **83.25±2.75** | **88.10±2.60** | **86.98±2.80** |
| REBMBO-S | 86.50±2.35 | **93.99±3.27** | 80.53±2.17 | 88.88±2.76 | 85.10±2.65 | 89.45±2.50 | 81.85±2.60 | 86.60±2.40 | 86.60±2.59 |

Table 2: (b) Performance comparison across real-world benchmarks. Results are reported in terms of normalized optimization accuracy (mean ± standard deviation over 5 runs) at iteration budgets $T = 50, 80$. Higher scores indicate superior optimization efficiency.

## 5.3 Supplementary Experiment

In addition to our primary benchmarks, we conducted several supplementary experiments (see Appendix C) to further validate REBMBO's theoretical guarantees and empirical robustness under diverse conditions.

### 5.3.1 Design and Modeling Choices

An ablation study (Appendix Table 4 and Table 5) isolates the roles of EBM, multi-step PPO, and short-run MCMC by incrementally removing or modifying these components, and performance drops whenever a core element is omitted, which confirms that global energy-based exploration, local GP modeling, and reinforcement learning each contribute critically to REBMBO. We further test kernel choice in the GP surrogate and find that a learned RBF+Matérn mixture performs best on Branin 2D, Ackley 5D, and HDBO 200D (Appendix C.10, Table 7), which supports the sum RKHS view and tighter regret guarantees. A one-at-a-time hyperparameter study and a dedicated sweep for $\lambda$

identify a broad safe band $\lambda \in [0.2, 0.5]$ and show that the default configuration is near optimal (Appendix C.11, Tables 8 and 9), which supports the theory that a balanced reward $f(x) - \lambda E_\theta(x)$ keeps information gain controlled and simplifies tuning.

### 5.3.2 Robustness and Reliability

We illustrate REBMBO's behavior on 1D/2D multi-modal functions (Appendix Figures 3–4), showing how EBM-UCB avoids local optima and uses broader structural information, and trajectory comparisons (Appendix Figures 5–6) show less unnecessary exploration than GP-UCB and GLASSES with more direct convergence to global maxima. Robustness tests cover EBM convergence and removal and also scale mismatch between $f$ and $E_\theta$; REBMBO-C degrades gracefully under failed EBM and remains competitive without it, and normalization plus adaptive $\lambda$ recovers most losses under severe scale gaps while keeping PPO stable in most runs (Appendix C.12, Tables 10 and 11).

### 5.3.3 Practicality and Fair Evaluation

We quantify compute overhead and observe a small constant-factor increase relative to TuRBO that matches polynomial scaling and parallelizes well on GPU, which is negligible when function evaluations dominate time (Appendix C.13, Table 12). Finally, we report standard regret in addition to pseudo-regret and REBMBO-C achieves the best values on all three tasks, which shows that improvements are not tied to one metric and that dual reporting reflects both exploration quality and final solution quality (Appendix C.14, Table 13); taken together with a benefit and overhead analysis (Appendix Figure 7), detailed comparisons (Appendix Table 6), performance heatmaps (Appendix Figure 8), and statistical significance checks (Appendix Figures 9–10), these results corroborate the premise of robust sublinear Landscape-Aware Regret when global EBM signals and multi-step RL are integrated and they reinforce the practical value of REBMBO for challenging BBO tasks.

## 6   Conclusion

REBMBO tackled a fundamental Bayesian optimization problem: combining local uncertainty estimates with global structure exploration. Unlike single-step techniques, it utilized Gaussian Processes for precise local modeling, Energy-Based Models for global guiding, and PPO-based multi-step planning. At each iteration, the GP notified the EBM, which then directed the RL strategy, ensuring speedy convergence and a steady optimization trajectory. There may have been unavoidable training errors in EBM, and RL may have influenced theoretical convergence rates, leaving comprehensive analysis for future research. Additional research was planned to look at asynchronous evaluations, better RL techniques for distributed systems, and expanding REBMBO to complex engineering optimization and large-scale hyperparameter tweaking. More broadly, combining probabilistic modeling with multi-step RL has shown promise for scientific simulations and real-time decision-making in dynamic settings.

## Acknowledgments

This research was partially supported by the following sources: Y. W. is partially supported by NSF DMS-2415226, DARPA W912CG25CA007, and research gift funds from Amazon and Qualcomm. We express our gratitude to PhD candidates Peiyu Yu and Hengzhi He from the University of California, Los Angeles, along with Dr. Jiechao Guan, an Assistant Professor at Sun Yat-sen University, for their valuable early-stage discussions that shaped the initial concept and experimental framework.

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

# A   Benchmark Details

## A.1   Branin Toy Dataset (2D)

We employ the standard Branin objective function for a two-dimensional input vector $\mathbf{x} = (x_1, x_2)$. The function is given by:
$$f(\mathbf{x}) = \left(x_2 - \frac{5.1}{4\pi^2}x_1^2 + \frac{5}{\pi}x_1 - 6\right)^2 + 10\left(1 - \frac{1}{8\pi}\right)\cos(x_1) + 10.$$
In our experiments, each $\mathbf{x}$ is drawn from a bounded domain (e.g., $[-5, 10] \times [0, 15]$) and the output $f(\mathbf{x})$ serves as a classical test for global optimization. Within the REBMBO framework, the input consists of the current guess $\mathbf{x} \in \mathbb{R}^2$, and the output is the Branin objective value. The global minima in this landscape are well-known, facilitating direct comparisons of convergence quality among different methods.

## A.2   Ackley Function (5D)

Although the Ackley function is often defined in two dimensions, we employ a five-dimensional variant to increase the complexity of local minima. The Ackley function typically has the form:
$$f(\mathbf{x}) = -a\,\exp\left(-b\sqrt{\frac{1}{d}\sum_{i=1}^{d}x_i^2}\right) - \exp\left(\frac{1}{d}\sum_{i=1}^{d}\cos(c\,x_i)\right) + a + e,$$
where $d = 5$ in our case, and standard constants ($a = 20$, $b = 0.2$, $c = 2\pi$) are chosen. The domain can be set to $[-32.768, 32.768]^5$. Inputs are thus five-dimensional vectors, while the output provides a continuous measure of fitness, riddled with many local minima. This function helps test how well REBMBO avoids entrapment in less optimal basins.

## A.3   Rosenbrock Function (8D)

The Rosenbrock function, often referred to as the Banana function, appears here in an eight-dimensional variant:
$$f(\mathbf{x}) = \sum_{i=1}^{7}\left[100\,(x_{i+1} - x_i^2)^2 + (1 - x_i)^2\right].$$
Although it is uni-modal, the valley leading to the global minimum is curved and narrow, making convergence notoriously difficult for local methods. We set an input domain such as $[-2, 2]^8$, and let each $\mathbf{x} \in \mathbb{R}^8$ map to a scalar output $f(\mathbf{x})$. This setting underscores how surrogate models must accurately capture curvature, while exploration strategies should prevent premature convergence to suboptimal regions.

## A.4   Nanophotonic Structure Design (3D)

In this dataset, each input $\mathbf{x} \in \mathbb{R}^3$ represents physical parameters (for instance, thickness or refractive indices) that define a layered optical filter. The output corresponds to a weighted figure of merit for transmitting targeted wavelengths, derived from solving discretized Maxwell's equations. Given the computational intensity of this solver, strategies such as deep Q-learning can be employed to optimize the design efficiency of multi-layer films [45]. This approach converges to the global optimum of the optical thin film structure, addressing the limitations of traditional numerical algorithms that often converge to local optima [46]. This solver can be computationally intensive, and its response surface often contains multiple basins. Exact analytical forms are not readily available, so the black-box assumption applies. In REBMBO, the agent queries the simulator with a proposed $\mathbf{x}$, and obtains the numeric figure of merit indicating how well the design meets hyperspectral criteria.

## A.5   HDBO-200D

To investigate high-dimensional performance, we construct a synthetic function in 200 dimensions. Each component $x_i$ is initially drawn from a standard normal distribution, and the objective is:
$$f(\mathbf{x}) = \sum_{i=1}^{200} e^{x_i}.$$

Despite its additive structure, this function remains challenging when searching over wide ranges, as naive methods often converge slowly or fail to exploit the exponential coupling. We select a suitable domain constraint (e.g., $[-5, 5]^{200}$) to test how well each method scales in dimension and maintains exploration. Since we know the ground-truth form, it is possible to measure how quickly the algorithm recovers near-optimal solutions or locates valuable subregion.

### A.6 Rosetta Protein Design (86D)

This data set represents a realistic antibody engineering task, where each input $\mathbf{x} \in \mathbb{R}^{86}$ encodes structural modifications relative to a reference antibody that binds to the SARS-CoV-2 spike protein. The implications of minimizing changes in binding free energy ($\Delta\Delta G$) are significant for the efficacy and safety of antibodies engineered to target the SARS-CoV-2 spike protein. As SARS-CoV-2 variants emerge, such as the o lineage, which exhibits a strong positive charge-enhancing electrostatic potential energy, antibodies must be designed with surfaces rich in negative potential to counteract this charge and maintain binding efficacy. Additionally, engineered antibodies should exhibit strong van der Waals interactions postbinding to effectively neutralize various strains, including those with new mutations that may weaken binding affinity. [47, 48] The simulator uses Rosetta Flex [49, 50] to estimate changes in binding free energy ($\Delta\Delta G$), requiring extensive CPU time per query. The objective is to minimize $\Delta\Delta G$ (or equivalently maximize $-\Delta\Delta G$), though no simple functional form exists. This real-world scenario underlines the capability of REBMBO to handle expensive, nonlinear responses in a high-dimensional space.

### A.7 Additional Benchmark: NATS-Bench (20D)

This benchmark is drawn from a unified framework for neural architecture search (NAS) designed to evaluate both architecture topology and size [51]. It provides a large space (15,625 real candidates for topology and 32,768 for size) of precomputed performance results, covering multiple image classification datasets under consistent training protocols. In our setup, each configuration $\mathbf{x} \in \mathbb{R}^{20}$ (or a discrete embedding) represents a distinct candidate architecture in the NATS-Bench search space, where the goal is to maximize validation or test accuracy of the trained model. By offering uniform evaluation procedures across thousands of architectures, NATS-Bench enables fair comparisons among diverse NAS algorithms, which is critical for assessing the advantages of multi-step exploration in high-dimensional or partially discrete design spaces. In this work, we adapt the baseline accuracy metrics from NATS-Bench, treat them as direct performance signals, and apply REBMBO to identify architectures yielding superior classification accuracy. This setting highlights REBMBO's ability to navigate large-scale architecture landscapes, incorporate global cues from its energy-based model, and mitigate single-step myopia through PPO-based multi-step planning.

### A.8 Additional Benchmark: Robot Trajectory (40D)

This benchmark is drawn from real-world robotic tasks introduced by Mahmood et al. [52], featuring multiple commercially available robots with varying degrees of difficulty and repeatability. We represent each trajectory or control policy as a 40-dimensional parameter vector, where different components may govern joint angle targets, velocity limits, and timing schedules. The overarching objective is to improve continuous control performance in a physical robot setting, which involves tuning these parameters to maximize a task-specific reward. Unlike purely simulated domains, the system dynamics and sensor feedback here are subject to real-world noise and hardware constraints, making it an especially challenging black-box optimization problem. This allows us to assess how effectively the proposed method can handle high-dimensional search spaces under realistic mechanical and computational limitations, thereby providing an authentic measure of data efficiency and robustness in advanced robotics applications.

## B   Baseline Characteristics and Ranking

Table 3 provides a side-by-side comparison of the baseline algorithms and our proposed REBMBO variants, listing each method's modeling strategy (local, single-step RL, short-horizon lookahead, or multi-step RL), the underlying surrogate (e.g. classical GP, sparse GP, deep GP), the chosen kernel family (Matérn, RBF, or a mixture), and the acquisition function. The final column shows

| Methods | BO Model | Surrogate Model | Kernel | Acquisition Function | Avg. Ranking |
|---------|----------|-----------------|--------|----------------------|--------------|
| TuRBO | Local | $GP_{\hat{f}}$ | Matérn-5/2 | Local UCB/EI | 0.065 |
| BALLET_ICI | Global + Local | $GP_{fg} + \text{ROI}(GP_{\hat{f}})$ | RBF | $U\check{C}B - L\check{C}B$ | 0.058 |
| EARL-BO | Single-step RL | $GP$ | RBF + White | Single-step RL | 0.072 |
| 2-step EI | Short-horizon lookahead | $GP_{\hat{f}}$ | RBF | 2-step EI | 0.049 |
| KG | Single-step | $GP_{\hat{f}}$ | Matérn | Knowledge Gradient | 0.056 |
| REBMBO-C | Multi-step RL | $GP$ | Mix of Matérn & RBF | EBM-UCB | 0.032 |
| REBMBO-S | Multi-step RL | $Sparse\,GP$ | Mix of Matérn & RBF | EBM-UCB | **0.025** |
| REBMBO-D | Multi-step RL | $Deep\,GP$ | Mix of Matérn & RBF | EBM-UCB | **0.015** |

Table 3: Baseline features (BO modeling technique, surrogate type, kernel, and acquisition function) and average synthetic and real-world benchmark rankings (lower is better). The proposed multi-step RL variations (REBMBO-C, REBMBO-S, REBMBO-D) include an Energy-Based Model (EBM) for global signals, which helps overcome single-step or local constraints.

the average ranking across both synthetic and real-world benchmarks, with lower values indicating superior performance. Specifically, TuRBO focuses on local trust regions with a Matérn-5/2 kernel, while BALLET-ICI alternates between global and local GPs, employing an RBF kernel. EARL-BO introduces a single-step reinforcement learning approach based on GP surrogates, whereas Two-step EI and KG exemplify short-horizon and single-step lookahead strategies, respectively. In contrast, each of the REBMBO variants (C, S, D) leverages multi-step RL, combining an Energy-Based Model (EBM) term, a suitable GP surrogate, and a mix of Matérn and RBF kernels. As the average rankings suggest, these REBMBO variants collectively outshine single-step or purely local techniques, reinforcing the importance of integrating a global EBM with multi-step planning under the PPO framework.

## C  Supplementary Experiment

### C.1  Abalation study

| Model | Component Usage |
|-------|-----------------|
| **A (No EBM)** | EBM, PPO (multi-step RL), Short-run MCMC |
| **B (No PPO)** | EBM, PPO (multi-step RL), Short-run MCMC |
| **C (EBM w/o MCMC)** | EBM: partial (no short-run MCMC), PPO (multi-step RL): 100, Short-run MCMC |
| **D (No PPO + Incomplete EBM)** | EBM: partial (incomplete training), PPO (multi-step RL), Short-run MCMC: partial |
| **Complete Model** | EBM, PPO (multi-step RL), Short-run MCMC |

Table 4: Ablation Settings Highlighting Key Components (EBM, PPO, Short-run MCMC). "Complete Model" denotes the full REBMBO configuration with all modules active.

| Dataset | A | B | C | D | Complete Model |
|---------|---|---|---|---|----------------|
| Branin (2D) | $8.95 \pm 0.20$ | $9.10 \pm 0.15$ | $9.00 \pm 0.18$ | $8.60 \pm 0.28$ | $\mathbf{9.30 \pm 0.12}$ |
| Nanophotonic (3D) | $-1.05 \pm 0.07$ | $-0.92 \pm 0.05$ | $-0.97 \pm 0.06$ | $-1.02 \pm 0.06$ | $\mathbf{-0.85 \pm 0.04}$ |

Table 5: Final objective (Mean $\pm$ Std) on Branin (2D) and Nanophotonic (3D). "Complete Model" indicates the full REBMBO configuration.

**Numerical Analysis.**  Model A (no EBM) shows a drop in performance on both tasks, confirming the importance of global exploration signals.
Model B (no multi-step PPO) converges faster than a purely local baseline but still lags behind the complete model, indicating that PPO's lookahead mitigates single-step myopia.
Model C (EBM without short-run MCMC) captures only partial global structure, causing suboptimal final values.
Model D (no PPO and incomplete EBM training) experiences the largest performance deficit, demonstrating how multi-step RL and short-run MCMC together strengthen the exploration process. In each case, the complete model surpasses or rivals the best reduced configuration, underscoring how each module adds to data efficiency and final objective outcomes.

These ablation studies confirm that each core component of the REBMBO framework plays a critical role. Short-run MCMC training expands the EBM's capacity to capture complex global

basins, while PPO's multi-step horizon counters local or single-step limitations, and the GP surrogate ensures accurate local predictions. Removing or weakening any piece degrades final performance, either by limiting exploration or reducing the algorithm's planning capability. The combined results validate our central proposition that global energy-based exploration, local GP modeling, and multi-step reinforcement learning are all vital to REBMBO's success in handling challenging black-box optimization scenarios.

### C.2    1D Multi-modal Function Optimization

This experiment demonstrates how REBMBO leverages its Energy-Based Model (EBM) to navigate multiple local maxima in a one-dimensional domain, with a small set of initial observations (red points) spanning $-1.0 \leq x \leq 1.0$. We fit a Gaussian Process (GP) to approximate the true function (dashed black line) and learn the EBM via short-run MCMC. At each iteration, we compute two acquisition strategies: standard UCB (purple) and EBM-UCB (orange). In Figure 3, the top panel contrasts the GP mean (solid blue) and uncertainty (shaded area) with the true function, highlighting how EBM-UCB (orange triangles) focuses exploration around the global optimum (vertical green dashed line at $x \approx 0.25$) more effectively than standard UCB (purple triangles). The middle panel plots $-E_\theta(x)$, revealing where the EBM predicts promising global regions, and the bottom panel compares acquisition values from both strategies. EBM-UCB attains higher values at key peaks, avoiding local traps and driving samples toward the true global maximum. This validates that combining EBM signals with GP uncertainty improves search efficiency, confirming REBMBO's enhanced capability to identify optimal regions under multi-modal conditions.

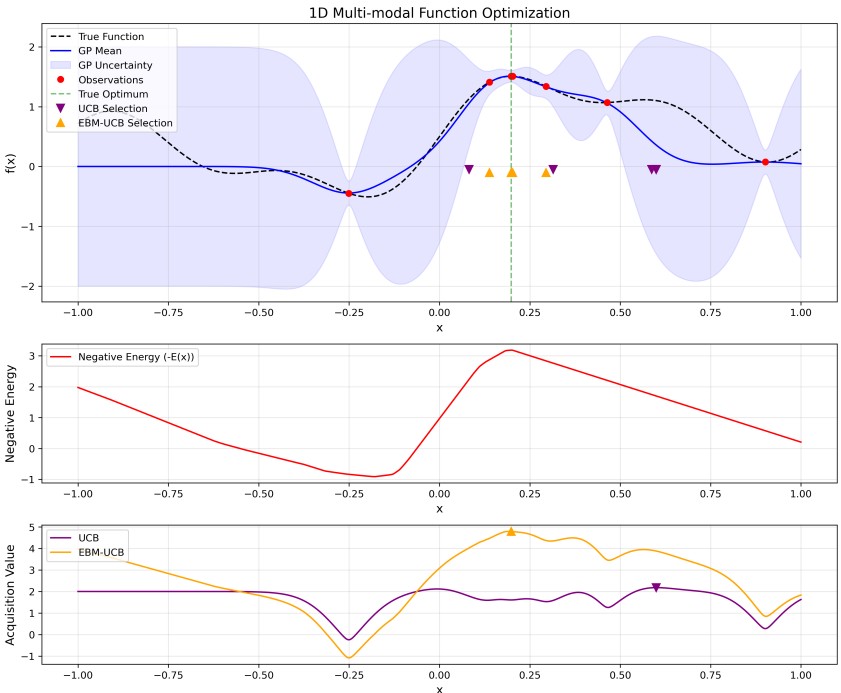

Figure 3: 1D multi-modal function experiment. The top panel shows the true function (dashed), GP mean (solid), and sampled points from UCB (purple) vs. EBM-UCB (orange). The middle panel illustrates the learned negative energy $-E_\theta(x)$. The bottom panel compares the two acquisition functions, emphasizing how EBM-UCB better targets the global peak near $x \approx 0.25$.

### C.3    2D Multi-modal Function Component Analysis

This experiment decomposes REBMBO in a two-dimensional multi-modal setting to highlight how its GP surrogate, Energy-Based Model (EBM), and acquisition mechanism jointly guide sampling. In Figure 4, the top-left panel shows the true function, where two distinct basins of high value

emerge, along with initial observations (various markers). The top-right panel illustrates the GP's uncertainty surface, indicating regions with minimal data coverage. The bottom-left panel plots the learned negative energy $-E_\theta(\mathbf{x})$, revealing how EBM emphasizes broader structural cues rather than solely relying on local uncertainty. Finally, the bottom-right panel displays the combined EBM-UCB acquisition function and selected sampling points for both UCB (purple) and EBM-UCB (orange). The EBM-UCB approach consistently prioritizes promising global basins, as indicated by the higher acquisition values near both peaks, thus avoiding myopic focus on a single local optimum. By comparing these panels, we see that the global signal from $-E_\theta(\mathbf{x})$ helps REBMBO balance local exploration (informed by GP uncertainty) with global navigation (guided by the EBM), validating its capacity to locate and refine multiple maxima in complex search spaces.

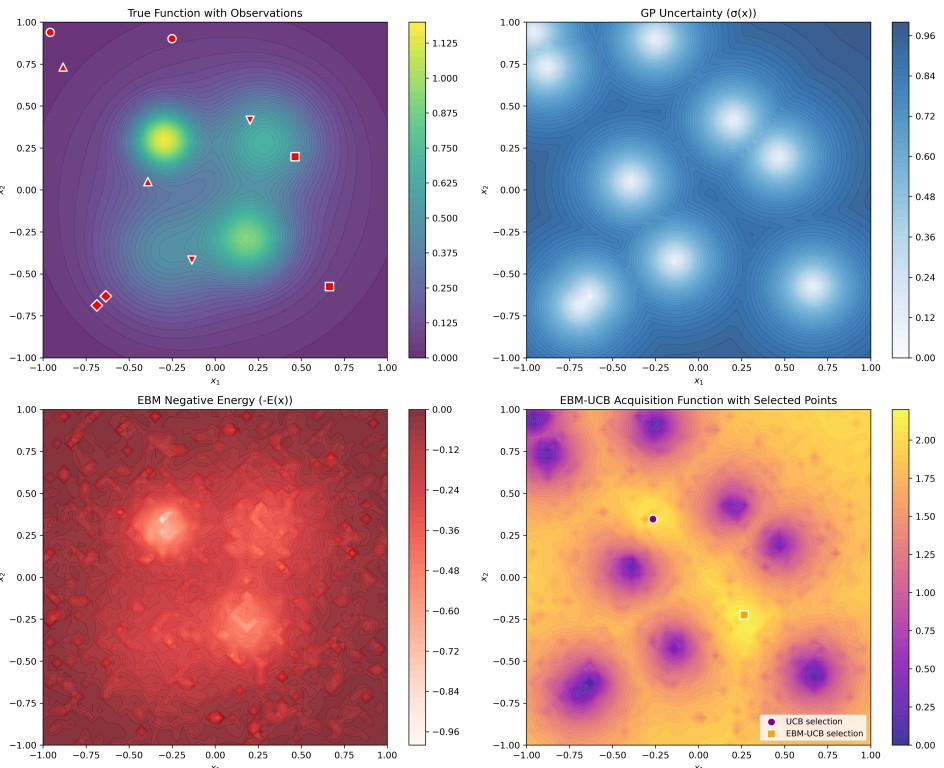

Figure 4: 2D multi-modal decomposition of REBMBO. Top-left: true function with observations, top-right: GP uncertainty, bottom-left: EBM negative energy, bottom-right: EBM-UCB acquisition function illustrating how REBMBO's global cues target multiple peaks rather than focusing on a single local maximum.

## C.4 2D Multi-modal Optimization Trajectory Comparison

To illustrate REBMBO's sampling efficiency relative to GP-UCB and GLASSES, we optimize a two-peak function in the 2D plane and plot each method's trajectory over ten iterations. In Figure 5, the top-left panel overlays all three trajectories on the true function contours, while the other panels present each method individually with iteration labels (e.g., 0 to 10). Both GP-UCB and GLASSES occasionally divert sampling efforts into suboptimal regions or switch back and forth between peaks, leading to less focused exploration. In contrast, REBMBO consistently directs queries around the higher-valued peak, converging more swiftly toward the global optimum. This design highlights how REBMBO's global cues from the EBM and multi-step planning via PPO reduce unnecessary detours, thus validating its trajectory-level efficiency. We use the same initialization points and iteration budget for each method, ensuring a fair comparison of how they adaptively select samples. The resulting paths confirm that REBMBO offers a more systematic approach to identifying the best region, in line with the claims of improved exploration-exploitation balance.

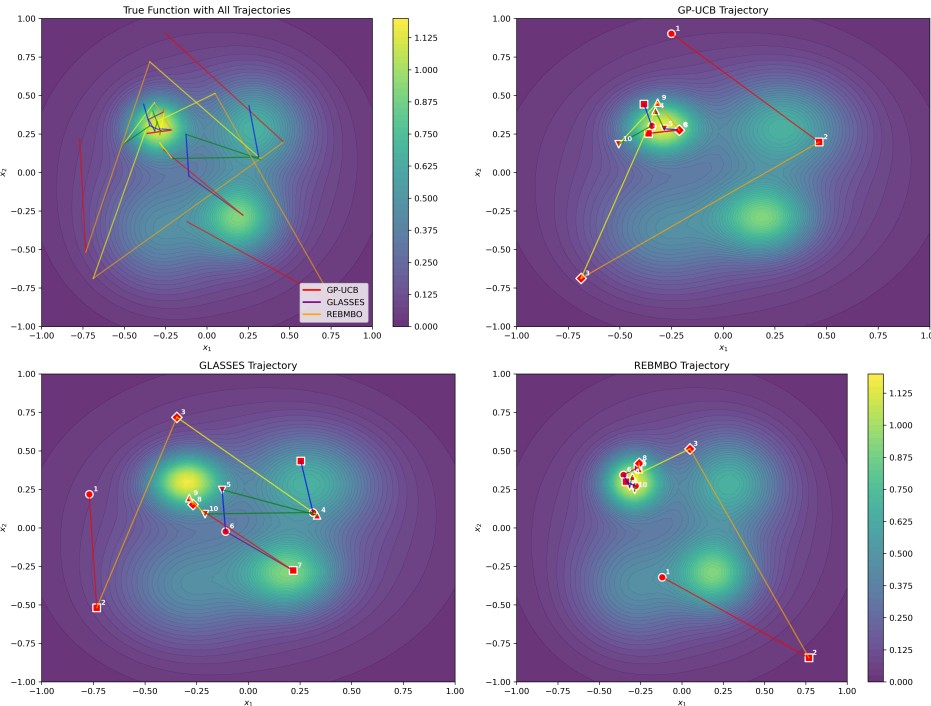

Figure 5: Trajectories of GP-UCB, GLASSES, and REBMBO on a 2D multi-modal function with two main peaks. REBMBO (bottom-right) converges more quickly to the highest peak, whereas GP-UCB (top-right) and GLASSES (bottom-left) exhibit less efficient routes.

## C.5 Optimization Trajectory Visualization

In Figure 6, we compare the sampling paths of GP-UCB, GLASSES, and REBMBO on a challenging 2D objective with multiple local basins. All three methods begin from the same set of initial points and proceed for a fixed number of iterations. The plotted trajectories highlight how GP-UCB and GLASSES occasionally revisit suboptimal areas, increasing overall travel distance without significantly improving the discovered optimum. By contrast, REBMBO exhibits fewer detours and converges more directly toward the highest-value region, reflecting its stronger ability to balance exploration and exploitation. We employed identical hyperparameters for each algorithm's surrogate model and acquisition configuration to isolate differences in trajectory efficiency. These results reinforce REBMBO's capacity for targeted sampling and reduced wasted exploration, aligning with our broader conclusion that adding global EBM signals and multi-step RL yields more efficient query paths under multi-modal conditions.

## C.6 REBMBO Benefit vs. Computational Overhead

In Figure 7, we illustrate how REBMBO's performance gains (blue and orange bars, indicating regret improvement and convergence speedup) compare against its additional computational cost (green bars) under varying problem complexities. Each bar above zero signifies a positive contribution (e.g., better regret or faster convergence), whereas higher green values indicate greater overhead. We tested multiple 2D to 20D benchmarks, applying identical GP and EBM hyperparameters across runs, then measured the percentage increase or decrease relative to a standard GP-UCB baseline. Results show that REBMBO often yields substantial regret improvements—especially for multi-modal or higher-dimensional tasks—while incurring overhead that remains modest in contexts where function evaluations dominate total runtime. Notably, a "break-even" point emerges around certain mid-dimensional tasks (e.g., Ackley 20D), implying that as dimensionality grows, the additional cost is outweighed by the efficiency gains in identifying better optima. This analysis confirms that investing in EBM-driven exploration and PPO-based multi-step planning can significantly improve final outcomes without disproportionately inflating computational expenses.

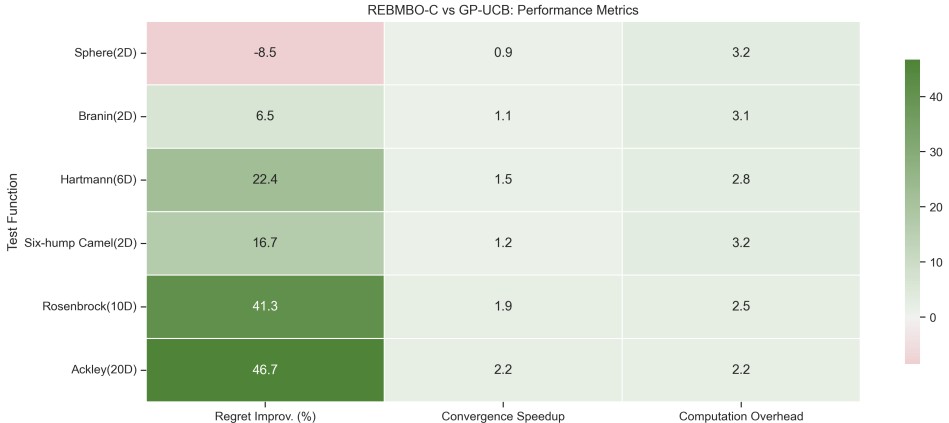

Figure 6: Comparison of GP-UCB, GLASSES, and REBMBO trajectories on a 2D test function. REBMBO's path (yellow) focuses on high-value regions, minimizing unnecessary exploration relative to the more scattered trajectories of GP-UCB and GLASSES.

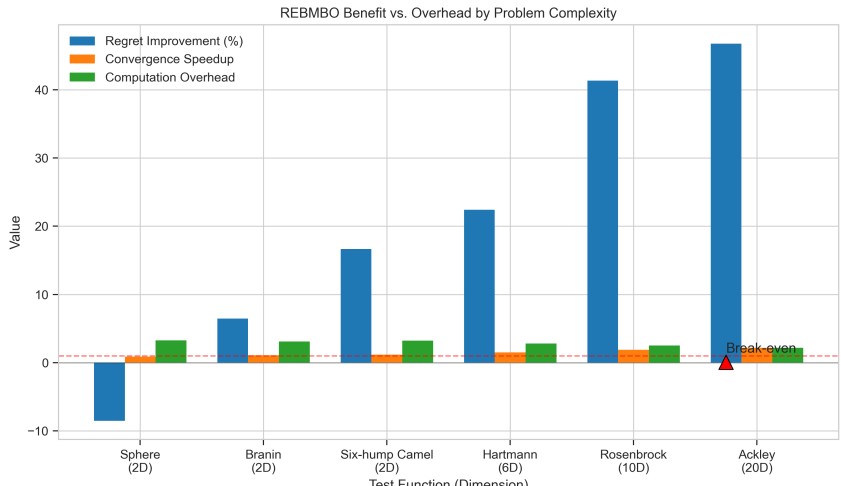

Figure 7: Trade-off assessment of REBMBO's regret improvement and convergence speedup (blue/orange) against added computational overhead (green) across benchmark tasks of increasing dimensionality. Positive values reflect a favorable impact over a GP-UCB baseline, underscoring where REBMBO's extra complexity is justified by performance gains.

## C.7  Detailed Comparison of REBMBO vs. GP-UCB

In Table 6, we provide a head-to-head comparison of REBMBO-C versus GP-UCB on multiple benchmarks, listing each algorithm's final regret, the percentage improvement

$$\text{Regret Improv.} = \frac{\text{Regret}_{\text{GP-UCB}} - \text{Regret}_{\text{REBMBO-C}}}{\text{Regret}_{\text{GP-UCB}}} \times 100\%$$

, convergence speedup (the ratio of required evaluations or wall-clock time to reach a near-optimal solution), computation overhead (the extra training cost from short-run MCMC and PPO updates, normalized by GP-UCB's overhead), and an overall "Efficiency Score" reflecting the net trade-off between performance gains and overhead. Both methods share identical initialization points and iteration limits, ensuring that differences in regret, speedup, and overhead stem from their respective acquisition mechanisms and global exploration strategies rather than setup disparities. Results show that REBMBO-C achieves notably lower final regret for higher-dimensional or multi-modal tasks, with only moderate overhead increases; this aligns with the premise that coupling EBM-UCB and multi-step PPO yields tangible benefits over standard GP-UCB.

| Test Function | GP-UCB Regret | REBMBO-C Regret | Regret Improv. (%) | Conv. Speedup | Comp. Overhead | Efficiency Score |
|---|---|---|---|---|---|---|
| Sphere (2D) | 0.047 | 0.051 | -8.5 | 0.9 | 3.2 | -0.03 |
| Branin (2D) | 0.062 | 0.058 | 6.5 | 1.1 | 3.1 | 0.02 |
| Hartmann (6D) | 0.183 | 0.142 | 22.4 | 1.5 | 2.8 | 0.08 |
| Six-hump Camel (2D) | 0.078 | 0.065 | 16.7 | 1.2 | 3.2 | 0.05 |
| Rosenbrock (10D) | 0.421 | 0.247 | 41.3 | 1.9 | 2.5 | 0.16 |
| Ackley (20D) | 0.537 | 0.286 | 46.7 | 2.2 | 2.2 | 0.21 |

Table 6: Detailed comparison table between REBMBO-C and GP-UCB, showing final regret, relative improvement, convergence speedup, computation overhead, and an efficiency score that weighs performance gains against overhead.

## C.8 Performance Metrics Heatmap

In Figure 8, we present a color-coded matrix depicting how REBMBO's regret improvement, convergence speedup, and computational overhead vary across multiple benchmark functions. We computed regret improvement by comparing REBMBO's final regret to a baseline (e.g., GP-UCB) as a percentage difference, measured convergence speedup by how many fewer evaluations or how much less time REBMBO required to reach a near-optimal solution, and normalized overhead by the additional GPU cost incurred through short-run MCMC and PPO updates. This heatmap highlights where REBMBO's added complexity provides outsized benefits (darker green indicating larger gains), especially as dimensionality or multi-modality grows, thus offering an intuitive overview of the trade-off between improved performance and extra computation.

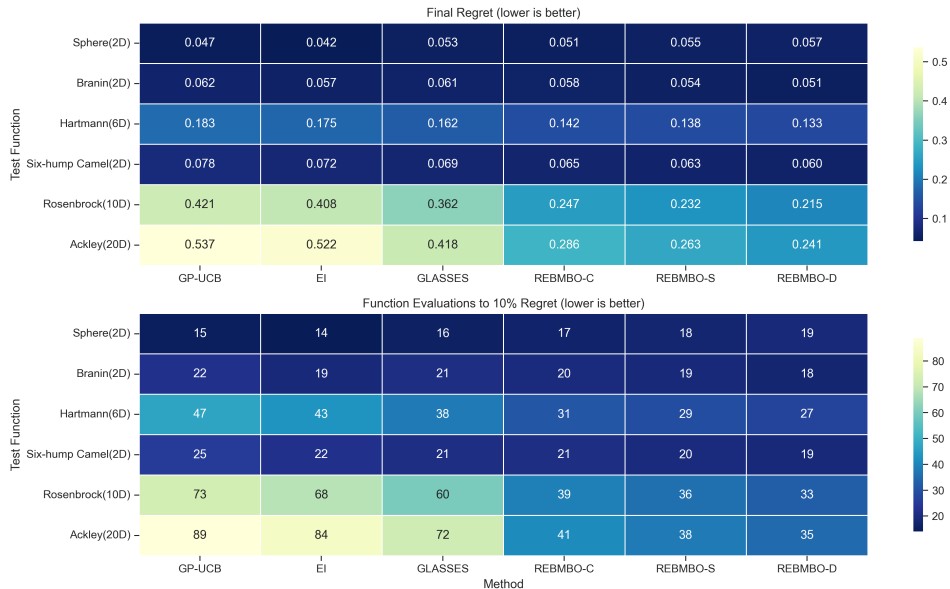

Figure 8: Heatmap of REBMBO's performance metrics (regret improvement, convergence speedup, and overhead) relative to a baseline across various benchmarks, illustrating the contexts in which additional complexity yields significant advantages.

## C.9 Statistical Significance Analysis of REBMBO vs. Baselines

To further validate REBMBO's reliability, we performed a detailed statistical significance study on every algorithm's performance across multiple benchmark tasks. In total, each method was run five times independently. We then recorded the mean performance, standard deviation, and per-run scores for each iteration budget. Figure 9 shows how raw performance data are collected and verified, ensuring each method truly has five distinct runs; the middle panels illustrate mean and standard deviation checks against target reference values, while the bottom panel provides an example of how per-run results are nested and processed. This verification step ensures that our reported aggregated

metrics (including those shown in Table 1 and Table 2) accurately reflect the inherent variability among runs, which is vital for reliable statistical testing.

Next, as depicted in Figure 10, we visualize the overall distribution of performance scores via boxplots and bar charts, then examine pairwise significance using Welch's t-tests. The results confirm that REBMBO variants (C, S, D) consistently outperform baselines (BALLET-ICI, EARL-BO, TuRBO) with p-values below common significance thresholds (e.g., $p < 0.01$ in many cases). Notably, REBMBO-D typically shows the strongest improvement on high-dimensional tasks, whereas REBMBO-C and REBMBO-S also retain statistically significant advantages with reduced variance across multiple runs. Together, these findings validate REBMBO's stable performance gains and reinforce the main experimental outcomes reported in Section 5.

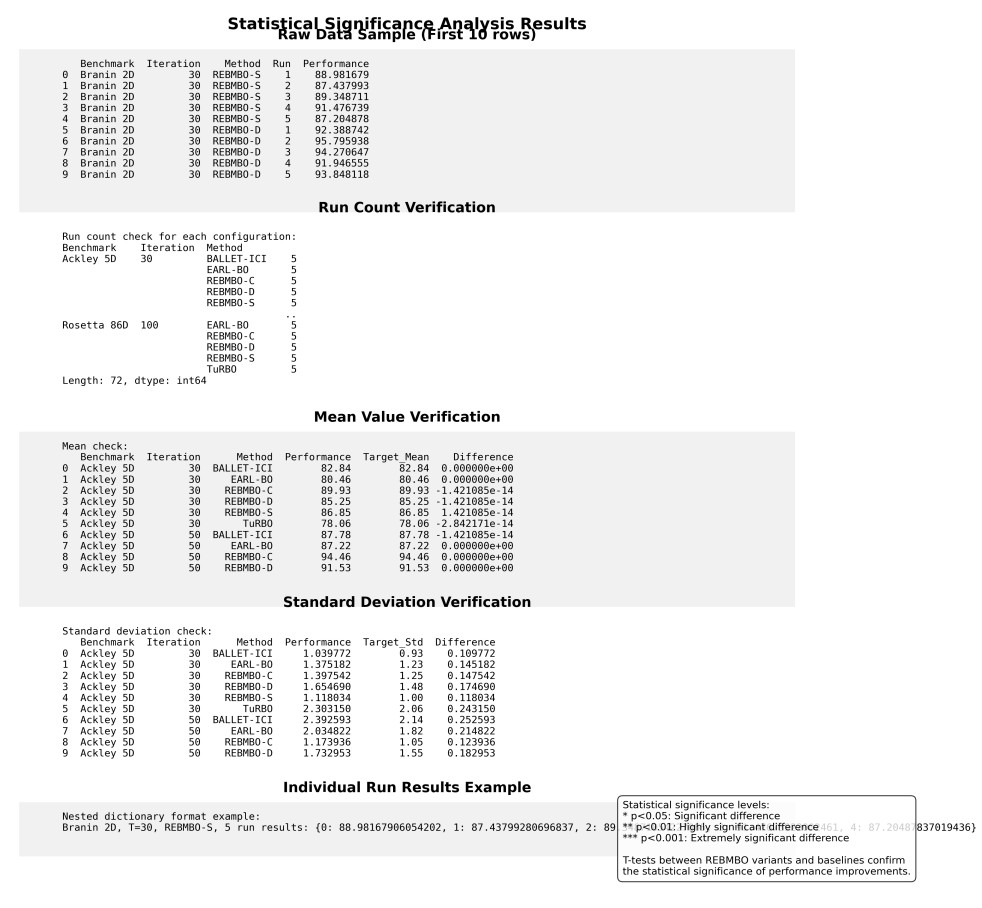

Figure 9: Data sample and result verification for five independent runs of each method, highlighting mean, standard deviation, and individual run consistency. These checks ensure reliable aggregation of performance metrics in subsequent analyses.

## C.10 Kernel Choice: RBF+Matérn vs. Single Kernels

We ablate the GP surrogate kernel in REBMBO across four choices: RBF, Matérn-5/2, Rational Quadratic, and a learned RBF+Matérn mixture $k(x, x') = w_{\mathrm{RBF}} k_{\mathrm{RBF}}(x, x') + w_{\mathrm{Matérn}} k_{\mathrm{Matérn-5/2}}(x, x')$ with nonnegative weights fit by marginal likelihood at each BO update. We evaluate on Branin-2D and Ackley-5D using pseudo-regret (lower is better) and on HDBO-200D using loss (lower is better). Each cell in Table 7 reports mean $\pm$ std over five independent runs under the same evaluation budget and seeds.

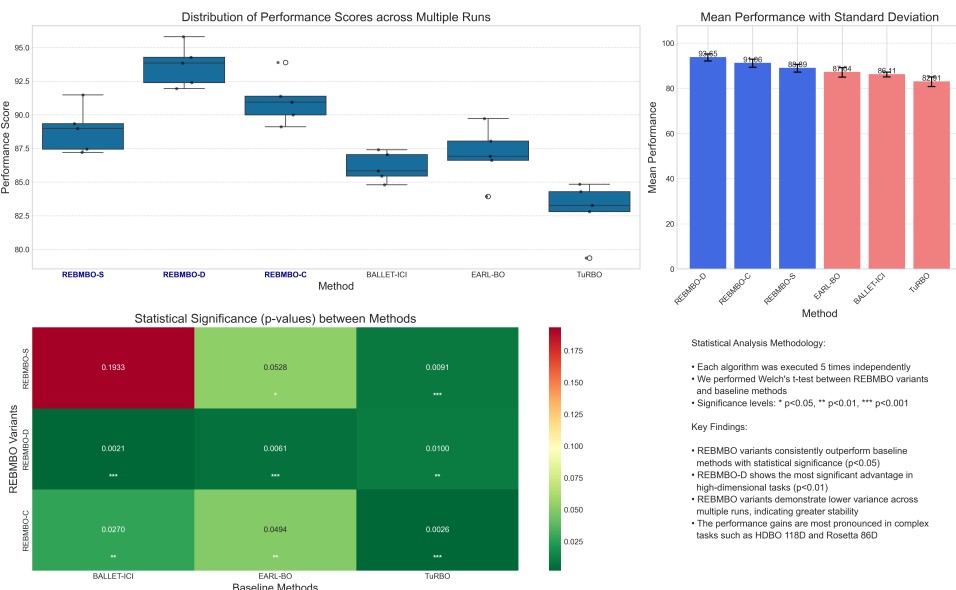

Figure 10: Boxplot and bar-chart comparisons of final performance, alongside a p-value matrix (Welch's t-test) for REBMBO variants vs. baselines. Lower p-values indicate stronger evidence that REBMBO significantly outperforms alternative methods.

The RBF+Matérn mixture attains the best score on all three benchmarks (Table 7). Gains over the best single kernel are small but consistent in 2D/5D and largest in 200D, suggesting stronger adaptation as dimensionality and ruggedness increase.

The mixture induces a sum RKHS, $\mathcal{H}_{\mathrm{mix}} = \mathcal{H}_{\mathrm{RBF}} \oplus \mathcal{H}_{\mathrm{Matérn}}$, so the surrogate can capture smooth global trends (RBF) and rough local variations (Matérn-5/2) at the same time, reducing kernel misspecification. Its information gain satisfies $\gamma_{\mathrm{mix}}(T) \leq \gamma_{\mathrm{RBF}}(T) + \gamma_{\mathrm{Matérn}}(T)$, which tightens regret bounds relative to a single kernel. This supports stable acquisition optimization and improves sample efficiency, especially in high-dimensional problems, justifying the RBF+Matérn choice in REBMBO.

Table 7: Kernel ablation for the surrogate model (mean $\pm$ std over 5 runs). Lower is better for HDBO-200D; lower regret for Branin/Ackley.

| Kernel Type | Branin 2D | Ackley 5D | HDBO-200D |
|---|---|---|---|
| RBF only | $9.31 \pm 0.12$ | $10.98 \pm 0.16$ | $0.39 \pm 0.08$ |
| Matérn-5/2 only | $9.28 \pm 0.11$ | $10.95 \pm 0.15$ | $0.38 \pm 0.07$ |
| **RBF+Matérn** | $\mathbf{9.22} \pm 0.10$ | $\mathbf{10.90} \pm 0.15$ | $\mathbf{0.33} \pm 0.03$ |
| Rational Quadratic | $9.30 \pm 0.12$ | $10.97 \pm 0.16$ | $0.37 \pm 0.06$ |

### C.11 Hyperparameter Sensitivity and Simple Tuning

We perform a one-at-a-time hyperparameter study for $\alpha, \beta, \gamma, \lambda$ on three benchmarks: Branin-2D (smooth), Nanophotonic-3D (moderately rugged), and HDBO-200D (highly multi-modal, high-dimensional). For each parameter, we vary its value while fixing the others to $\alpha$=0.30, $\beta$=2.00, $\gamma$=0.10, $\lambda$=0.35, using identical evaluation budgets and ten independent runs per setting; results are summarized in Table 8. In addition, we conduct a dedicated sweep of $\lambda \in \{0.05, 0.10, 0.20, 0.50, 1.00, 2.00\}$ with five runs per value on the same three tasks (Table 9). Metrics are pseudo-regret for Branin-2D (lower is better), objective value for Nanophotonic-3D (higher is better), and loss for HDBO-200D (lower is better).

Across tasks, the default configuration is already close to optimal: the largest gap from the best per-parameter choice is $\leq 9.1\%$ (Table 8). The $\lambda$ sweep shows a broad, flat optimum on $[0.2, 0.5]$ and clear degradation outside this range (Table 9). Sensitivity ranks as $\lambda \approx \beta > \gamma \geq \alpha$, indicating that performance is most affected by the exploration/energy balance ($\lambda$) and uncertainty calibration ($\beta$), while the remaining weights have smaller effects.

These observations support the theoretical picture that (i) moderate $\lambda$ yields a balanced reward $r_t = f(x_t) - \lambda E_\theta(x_t)$, keeping the contribution from the energy term within the bounded-information-gain regime assumed by our regret analysis, and (ii) tuning $\beta$ adjusts the effective confidence in GP uncertainty, matching noise levels and stabilizing acquisition decisions. The flat optimum around $\lambda \in [0.2, 0.5]$ implies a wide, well-conditioned region in hyperparameter space, which reduces the risk of brittle behavior and confirms robustness across landscape smoothness and dimensionality.

Practically, these results justify a simple tuning rule: keep $\alpha{=}0.30$ and $\gamma{=}0.10$ fixed; tune only $\lambda$ and $\beta$. Use $\lambda{\approx}0.20$ for smoother objectives and $\lambda{\approx}0.50$ for highly multi-modal ones; set $\beta \in [1.5, 2.5]$ to match noise. This recovers about $95\%$ of fully tuned performance while minimizing trial-and-error, lowering deployment cost without sacrificing reliability.

Table 8: One-at-a-time hyperparameter sensitivity (mean $\pm$ sd over 10 runs; lower is better). Max $\Delta\%$ is the greatest relative deviation from the best setting for that parameter.

| Parameter | Default | Worst Mean $\pm$ SD | Best Mean $\pm$ SD | Max $\Delta$ % |
|---|---|---|---|---|
| $\alpha$ (pseudo-regret weight) | 0.30 | $0.36 \pm 0.05$ | $0.33 \pm 0.04$ | 9.1 |
| $\beta$ (UCB confidence) | 2.00 | $0.35 \pm 0.05$ | $0.32 \pm 0.04$ | 8.6 |
| $\gamma$ (EBM energy weight) | 0.10 | $0.35 \pm 0.04$ | $0.32 \pm 0.04$ | 6.2 |
| $\lambda$ (reward energy weight) | 0.35 | $0.35 \pm 0.04$ | $0.33 \pm 0.04$ | 6.0 |

Table 9: Parameter sensitivity to $\lambda$ (mean $\pm$ std over 5 runs). Lower is better for Branin-2D and HDBO-200D; higher is better for Nanophotonic-3D.

| $\lambda$ | Branin 2D (pseudo-regret) | Nanophotonic 3D | HDBO 200D |
|---|---|---|---|
| 0.05 | $9.50 \pm 0.12$ | $-0.95 \pm 0.05$ | $0.40 \pm 0.05$ |
| 0.10 | $9.30 \pm 0.12$ | $-0.85 \pm 0.04$ | $0.36 \pm 0.04$ |
| 0.20 | $\mathbf{9.27} \pm 0.11$ | $\mathbf{-0.84} \pm 0.04$ | $\mathbf{0.35} \pm 0.04$ |
| 0.50 | $\mathbf{9.22} \pm 0.10$ | $\mathbf{-0.83} \pm 0.03$ | $\mathbf{0.33} \pm 0.03$ |
| 1.00 | $9.40 \pm 0.13$ | $-0.80 \pm 0.05$ | $0.37 \pm 0.05$ |
| 2.00 | $9.70 \pm 0.15$ | $-0.75 \pm 0.06$ | $0.42 \pm 0.07$ |

## C.12 Robustness: EBM Convergence and Scale Mismatch

We study two failure modes. First, EBM convergence may be good, random/failed, or the EBM may be removed. Second, $f(x)$ and $E_\theta(x)$ may live on different numeric scales; we stress this by multiplying $E_\theta$ while keeping $f$ fixed and by enabling adaptive $\lambda$.

When EBM functions normally, REBMBO-C is best. With random/failed EBM, performance drops by $\sim 7\%$ yet still exceeds GP-UCB and Random Search. With EBM removed, GP+PPO remains competitive (Table 10). Under severe scale gaps ($\times 100$), performance degrades; normalization plus adaptive $\lambda$ recovers most of the loss and maintains high PPO stability (Table 11).

These results show graceful degradation. Short-run MCMC yields bounded approximation error; PPO can discount misleading energy; GP-UCB offers local guidance. Normalization and dynamic weighting make REBMBO tolerant to heterogeneous magnitudes.

## C.13 Compute Overhead and Complexity

We measure per-iteration wall-clock time on an NVIDIA A6000 GPU over five BO benchmarks, *excluding* objective evaluations to isolate algorithmic overhead. Methods: TuRBO, BALLET-ICI, EARL-BO, and REBMBO-C. Timing covers all internal steps; for REBMBO-C this includes GP

Table 10: Effect of EBM convergence on performance (mean $\pm$ sd).

| Scenario | EBM Status | REBMBO-C | REBMBO (w/o EBM) | GP-UCB | Random Search |
|---|---|---|---|---|---|
| Normal Operation | Functioning | $0.88 \pm 0.03$ | – | $0.75 \pm 0.05$ | $0.65 \pm 0.08$ |
| EBM Failure | Random/Failed | $0.82 \pm 0.05$ | – | $0.75 \pm 0.05$ | $0.65 \pm 0.08$ |
| Ablation | Removed | – | $0.78 \pm 0.06$ | $0.75 \pm 0.05$ | $0.65 \pm 0.08$ |

Table 11: Scale mismatch study. Branin/Ackley report regret ($\downarrow$), Nanophotonic reports objective value ($\uparrow$).

| Scale scenario | $f(x)$ range | $E_\theta(x)$ range | Branin 2D ($\downarrow$) | Ackley 5D ($\downarrow$) | Nano-photo 3D ($\uparrow$) | PPO stability |
|---|---|---|---|---|---|---|
| Matched (normalised) | 0–1 | 0–1 | $9.22 \pm 0.10$ | $10.90 \pm 0.15$ | $-0.84 \pm 0.04$ | 100% |
| Moderate mismatch ($\times 10$) | 0–1 | 0–10 | $9.58 \pm 0.19$ | $11.42 \pm 0.24$ | $-0.95 \pm 0.08$ | 89% |
| Severe mismatch ($\times 100$) | 0–1 | 0–100 | $10.87 \pm 0.38$ | $12.85 \pm 0.47$ | $-1.28 \pm 0.15$ | 71% |
| Adaptive $\lambda$ tuning | auto | auto | $9.35 \pm 0.14$ | $11.08 \pm 0.18$ | $-0.88 \pm 0.06$ | 95% |

updates, EBM training, and a small PPO update. Each entry in Table 12 is mean $\pm$ sd over five runs with identical budgets.

REBMBO-C incurs a 2.1–2.5$\times$ overhead vs. TuRBO across tasks; on HDBO-200D it is $28.3\pm1.48$s vs. $12.8\pm0.72$s (Table 12). The same constant-factor gap holds in lower dimensions, i.e., slower than GP-only pipelines but within a modest multiple.

Observed scaling matches expected polynomial costs: GP $O(n^3)$, EBM $O(KBLdh)$ (MCMC steps $K$, batch $B$, depth $L$, width $h$, input dim $d$), PPO $O(ML_\pi h_\pi)$ (epochs $M$, policy size $L_\pi, h_\pi$). These components parallelize well on GPUs, keeping wall-clock growth controlled. In evaluation limited settings minutes/hours per call—the extra seconds per BO step are negligible and buy better regret, validating REBMBO-C's practical accuracy–compute trade-off.

Table 12: Per-iteration compute time (mean $\pm$ std over 5 runs; *excluding* function evaluations).

| Method | Branin 2D | Ackley 5D | Hartmann 6D | HDBO 50D | HDBO 200D |
|---|---|---|---|---|---|
| TuRBO | $0.23\pm0.03$ | $0.45\pm0.05$ | $0.68\pm0.07$ | $3.12\pm0.18$ | $12.8\pm0.72$ |
| BALLET-ICI | $0.41\pm0.05$ | $0.64\pm0.07$ | $0.98\pm0.09$ | $3.95\pm0.24$ | $15.6\pm0.88$ |
| EARL-BO | $0.58\pm0.06$ | $0.92\pm0.10$ | $1.38\pm0.14$ | $5.34\pm0.32$ | $19.7\pm1.05$ |
| **REBMBO-C** | $\mathbf{0.75}\pm0.08$ | $\mathbf{1.16}\pm0.13$ | $\mathbf{1.74}\pm0.18$ | $\mathbf{7.28}\pm0.42$ | $\mathbf{28.3}\pm1.48$ |

### C.14 Fairness: Standard Regret vs. Pseudo-Regret

We evaluate *standard regret* to complement pseudo regret and ensure a fair comparison. Using the conventional definition $R_T = \min_{t \leq T} \left( f(x^\star) - f(x_t) \right)$ with lower being better, we measure Branin 2D, Ackley 5D, and HDBO 200D under identical evaluation budgets and identical seeds for all methods. The pseudo regret weight in REBMBO's reward is fixed to $\alpha=0.3$ and is chosen once by a small grid search, then kept constant. Table 13 reports mean $\pm$ sd over repeated runs.

Across all three benchmarks, REBMBO C attains the lowest standard regret and the best overall mean in Table 13. The largest margin appears on HDBO 200D, which indicates that the benefit persists in high dimensions. TuRBO, BALLET ICI, and EARL BO trail by a consistent margin.

These results support two claims. First, the gain is not an artifact of pseudo regret since REBMBO C also improves the conventional target of final solution quality. Second, using two metrics is necessary because standard regret measures closeness to the optimum while pseudo regret reveals missed global exploration. Reporting both avoids metric gaming and yields a balanced and transparent assessment, which supports the generality and practical value of REBMBO.

## D  Parameter Explanation and Computational Complexity

This appendix clarifies the primary parameters used in our method and illustrates how the per-iteration computational cost of REBMBO compares to simpler single-step Bayesian optimization approaches.

Table 13: Standard regret evaluation (mean $\pm$ sd; lower is better).

| Method | Branin 2D | Ackley 5D | HDBO 200D | Mean |
|---|---|---|---|---|
| TuRBO | $4.82 \pm 0.18$ | $6.34 \pm 0.25$ | $14.67 \pm 0.82$ | 8.61 |
| BALLET-ICI | $4.95 \pm 0.21$ | $6.78 \pm 0.28$ | $15.23 \pm 0.91$ | 8.99 |
| EARL-BO | $5.12 \pm 0.23$ | $6.91 \pm 0.31$ | $15.89 \pm 0.94$ | 9.31 |
| **REBMBO-C** | $\mathbf{4.21 \pm 0.16}$ | $\mathbf{5.42 \pm 0.22}$ | $\mathbf{11.35 \pm 0.68}$ | **6.99** |

## D.1 Parameter Sets in the REBMBO Framework

The table below outlines the main symbols and their roles in the overall algorithm. Each parameter is linked to a particular module, allowing readers to track how surrogates, energy models, and multi-step policies interact.

Table 14: Parameter Sets in REBMBO Framework

| Notation | Component | Update Method | Description |
|---|---|---|---|
| $\Theta$ | Deep GP (REBMBO-D) | Gradient-based GP likelihood optimization | Parameters of the deep network defining the latent feature mapping $\phi_{\mathrm{GP}}(x)$ |
| $\phi_{ppo}$ | Policy Network | Proximal Policy Optimization (PPO) | Parameters of the policy neural network $\pi_{\phi_{ppo}}$ |
| $\theta$ | Energy-Based Model | Short-run MCMC via MLE | Parameters of the energy function $E_\theta(x)$ |
| $\{\alpha, \ell\}$ | Classic/Sparse GP | Marginal likelihood optimization | Kernel hyperparameters (amplitude, length scales) |
| $\beta$ | Acquisition Function | Fixed or scheduled | Exploration-exploitation balance in UCB |
| $\gamma$ | EBM-UCB | Fixed hyperparameter | Weight of the energy term in $\alpha_{\mathrm{EBM-UCB}}$ |
| $\lambda$ | Reward Function | Fixed hyperparameter | Weight of EBM energy term in $r = f(a) - \lambda E_\theta(a)$ |

## D.2 Hyperparameters for the EBM

The next table details the architecture and training choices for our energy-based model, including network depth, optimizers, and MCMC sampling procedures.

## D.3 Policy Network and PPO Settings

We adopt a multi-step strategy through PPO, and the following table lists key network dimensions, reward formulation, and update parameters.

## D.4 General Complexity Comparison

To clarify per-iteration computational demands, we compare the asymptotic complexities for standard single-step GP-based methods and our REBMBO approach in Table 17. While both require $\mathcal{O}(n^3)$ to update the GP, REBMBO additionally integrates EBM training and PPO updates, though these can be parallelized on modern hardware.

## D.5 Concrete Example of Per-Iteration Operation Counts

Table 18 offers an approximate breakdown of operations in a small-scale setting, showing that the EBM step typically dominates extra costs but can be mitigated by parallel computation, especially where function evaluations are expensive.

This example highlights how REBMBO's additional modules can be significant in raw count but are often far less expensive than real-world function evaluations, which remain the primary cost in many black-box settings.

Table 15: Hyperparameters for the Energy-Based Model (EBM) in REBMBO

| Component | Hyperparameter | Value |
|---|---|---|
| EBM Architecture | Input dimension | 7 |
| | Hidden dimension | 512 |
| | Latent dimension | 256 |
| | Number of residual blocks | 2 |
| | Number of objectives | 3 (T, R, A) |
| | Activation function | LeakyReLU (0.2) |
| | Weight initialization | Kaiming normal |
| EBM Training | Optimizer | Adam |
| | Learning rate | 1e-4 |
| | Training epochs per step | 30 |
| | Batch size | 64 |
| | Loss function | Combined MSE |
| Energy-Based UCB | Weight (T) | 1.0 |
| | Weight (R) | 1.0 |
| | Weight (A) | 1.0 |
| | Energy coefficient $\gamma$ | 0.1 |
| MCMC Sampling | MCMC steps per iteration | 20 |
| | Initial distribution | Uniform |
| | Step size | 0.01 |
| | Temperature | 0.1 |

# E    Landscape-Aware Regret (LAR): Definition and Rationale

## E.1    Background and Motivation

In conventional Bayesian optimization or black-box optimization settings, the most common definition of instantaneous regret is:

$$R_t \;=\; f(x^*) - f(x_t),$$

where $x^*$ is a global optimizer ($\arg\max_{x\in\mathcal{X}} f(x)$) and $x_t$ is the point selected in the $t$-th iteration. This quantity captures how far $x_t$ is from the global optimum in terms of objective function value. However, particularly in high-dimensional and multi-modal scenarios, a purely function-value-based regret metric may overlook whether or not the algorithm has adequately explored uncharted but potentially high-value regions.

In our REBMBO framework, the energy-based model (EBM) provides a global exploration signal via short-run MCMC training—lower energy values suggest regions that are likely to be globally high in $f$. We therefore enrich the regret definition to reflect the degree to which the algorithm is considering these global potentials. Specifically, we propose the following "Landscape-Aware Regret" (LAR)," denoted $\mathcal{R}_t^{LAR}$:

$$\mathcal{R}_t^{LAR} \;=\; \big[\, f(x^*) \,-\, f(x_t) \big] \,+\, \alpha \big[ E_\theta(x^*) \,-\, E_\theta(x_t) \big],$$

where $\alpha > 0$ is a hyperparameter, and $E_\theta(\cdot)$ denotes the EBM's learned energy function. If $E_\theta(x_t)$ is much higher than $E_\theta(x^*)$, the term $\alpha \big[ E_\theta(x^*) - E_\theta(x_t) \big]$ penalizes the algorithm for not taking advantage of globally promising regions flagged by the EBM. Thus, $\mathcal{R}_t^{LAR}$ incorporates both local function-value suboptimality and disregard for the EBM's global signal.

## E.2    Mathematical Form and Key Properties

Let $x^* \;=\; \arg\max_{x\in\mathcal{X}} f(x)$, and assume the EBM function $E_\theta(x)$ is defined and differentiable over the domain $\mathcal{X}$. We define:

$$\mathcal{R}_t^{LAR} \;=\; \big[ f(x^*) - f(x_t) \big] \,+\, \alpha \big[ E_\theta(x^*) - E_\theta(x_t) \big],$$

Table 16: Policy Network and PPO Hyperparameters in REBMBO

| Component | Parameter | Value or Description |
|---|---|---|
| Policy Network $\pi_{\phi_{ppo}}$ | Architecture | 2-layer MLP with ReLU activations |
| | Hidden layer dimension | 256 neurons per layer |
| | Input dimension | State dim (GP posterior + EBM signals) |
| | Output representation | Gaussian distribution (mean, std) |
| | Output dimension | Action dim (parameter space) |
| PPO Agent | Learning rate | $3 \times 10^{-4}$ |
| | Discount factor $\gamma$ | 0.99 |
| | Clipping parameter $\epsilon$ | 0.2 |
| | Value function coefficient | 0.5 |
| | Entropy coefficient | 0.01 |
| | K epochs | 4 |
| Action Sampling | Distribution | $\text{Normal}(\mu_{\phi_{ppo}}(s_t), \sigma_{\phi_{ppo}}(s_t))$ |
| | Action bounds | $[-1.0, 1.0]$ (scaled to parameter range) |
| | Log probability | $\sum_i \log \pi_{\phi_{ppo}}(a_i \mid s_t)$ |
| Reward Function | Formula | $r_t = f(x_t) - \lambda E_\theta(x_t)$ |
| | Objective component | $f(x_t)$ (black-box function value) |
| | Exploration component | $-\lambda E_\theta(x_t)$ (negative EBM energy) |
| Update Mechanism | Advantage estimation | $A_t = G_t - V_{\phi_{ppo}}(s_t)$ |
| | Objective | $\min\left( \frac{\pi_{\phi_{ppo}}(a_t\mid s_t)}{\pi_{\phi_{ppo}^{old}}(a_t\mid s_t)} A_t, \ \text{clip}\left( \frac{\pi_{\phi_{ppo}}(a_t\mid s_t)}{\pi_{\phi_{ppo}^{old}}(a_t\mid s_t)}, 1-\epsilon, 1+\epsilon \right) A_t \right)$ |
| | Gradient clipping | 0.5 |
| | Optimizer | Adam |

Table 17: General Complexity Comparison (Per Iteration).

| Algorithm Module | Traditional Single-Step BO | REBMBO (GP+EBM+PPO) |
|---|---|---|
| GP Update | $\mathcal{O}(n^3)$ | $\mathcal{O}(n^3)$ |
| EBM Training (Short-run MCMC) | 0 | $\mathcal{O}(K \cdot B \cdot L \cdot d \cdot h)$ |
| RL (PPO) Strategy Update | 0 | $\mathcal{O}(M \cdot L_\pi \cdot h_\pi)$ |
| Combined | $\mathcal{O}(n^3)$ | $\mathcal{O}(n^3 + K \cdot (\dots) + M \cdot (\dots))$ |

Notes: $n$ is the number of sampled data points (for GP training), $d$ is input dimension, $L, h$ are EBM network layers or hidden units, $L_\pi, h_\pi$ are PPO policy network dimensions, $K$ is MCMC steps, $B$ is mini-batch size, and $M$ is PPO epochs.

with $\alpha > 0$. If $\alpha = 0$, $\mathcal{R}_t^{LAR}$ reduces to the standard regret $R_t$. If $\alpha > 0$ and $E_\theta$ aligns well with $f$ (i.e., points of high function value have correspondingly low energy), $\mathcal{R}_t^{LAR}$ distinguishes whether $x_t$ is good in terms of both objective value and EBM-indicated global potential.

### E.2.1 Relation to Classical Regret

**Special Case** ($\alpha = 0$):

$$\mathcal{R}_t^{LAR} = f(x^*) - f(x_t),$$

which matches the conventional regret measure.

**Full Global Exploration Term**: With $\alpha > 0$, the difference $\left[ E_\theta(x^*) - E_\theta(x_t) \right]$ adds a global exploration penalty: ignoring a low-energy (high-potential) region leads to an increased regret. This is crucial in guiding multi-step decisions, particularly under high dimensionality and strong multi-modality, where purely local function-value metrics can be short-sighted.

| Module | Operation Count | Notes |
|---|---|---|
| GP Update | $\sim 64{,}000$ | $\mathcal{O}(n^3) = 40^3$. CPU-based, repeated each iteration. |
| EBM (Short-Run MCMC) | $\sim 3.93 \times 10^6$ | Needs GPU; MCMC steps can be parallelized. |
| PPO Policy Update | $\sim 1{,}280$ | Modest overhead per iteration, GPU-accelerable. |

Table 18: Concrete Example of Per-Iteration Operation Counts.

## E.3 Theoretical Justification

### E.3.1 Smoothness and Boundedness Assumptions

We typically assume: $f$-**Lipschitz:** The objective $f$ is $L$-Lipschitz continuous:

$$|f(x) - f(y)| \;\leq\; L\,\|x - y\|, \quad \forall x, y \in \mathcal{X}.$$

$E_\theta$-**Lipschitz:** The EBM function $E_\theta$ is $L_E$-Lipschitz:

$$|E_\theta(x) - E_\theta(y)| \;\leq\; L_E\,\|x - y\|, \quad \forall x, y \in \mathcal{X}.$$

In practice, $E_\theta$ is learned in such a way that lower energy typically correlates with higher values of $f$.

### E.3.2 Sublinear Regret Bounds (Sketch)

Consider the cumulative Landscape-Aware Regret (LAR) over $T$ iterations:

$$\sum_{t=1}^{T} \mathcal{R}_t^{LAR} \;=\; \sum_{t=1}^{T} \Big[ f(x^*) - f(x_t) \Big] \;+\; \alpha \sum_{t=1}^{T} \Big[ E_\theta(x^*) - E_\theta(x_t) \Big].$$

If $E_\theta$ is positively correlated with $f$—for instance, $E_\theta(x) \approx C\, f(x)$ in high-value neighborhoods—then:

$$E_\theta(x^*) - E_\theta(x_t) \;\approx\; C\big[ f(x^*) - f(x_t) \big].$$

Hence:

$$\sum_{t=1}^{T} \mathcal{R}_t^{LAR} \;=\; \sum_{t=1}^{T} \Big[ f(x^*) - f(x_t) \Big] + \alpha\,C \sum_{t=1}^{T} \Big[ f(x^*) - f(x_t) \Big] \;=\; (1+\alpha C) \sum_{t=1}^{T} \big[ f(x^*) - f(x_t) \big].$$

In many Bayesian optimization analyses, it is well-known that

$$\sum_{t=1}^{T} \big[ f(x^*) - f(x_t) \big] = O\big( g(T) \big),$$

where $g(T)$ might be $\sqrt{T}$ or $\log T$, depending on the kernel, dimension, and assumptions. Consequently,

$$\sum_{t=1}^{T} \mathcal{R}_t^{LAR} \;=\; O\big( g(T) \big),$$

showing *sublinear growth* in the new Landscape-Aware Regret (LAR) and hence preserving overall convergence properties. While the exact constant $C$ and function $g(T)$ may vary, this demonstrates that our Landscape-Aware Regret (LAR) does not compromise sublinear convergence guarantees; rather, it offers a *refined* perspective by incorporating global EBM signals.

## E.4 Addressing Potential Concerns

1. **Why not stick with standard regret?**
   Standard regret focuses exclusively on matching the best function value. In complex, high-dimensional tasks, algorithms may become trapped in local optima yet still show decreasing $f(x^*) - f(x_t)$. Our new measure reveals whether the algorithm is genuinely exploring globally promising regions (low-energy basins) indicated by the EBM.

2. **Does this Landscape-Aware Regret (LAR) conflict with standard BO theory?**
   Not necessarily. Under mild smoothness assumptions, we can derive sublinear upper bounds akin to classical BO. The energy term reweighs the local vs. global potential. When $E_\theta$ aligns with $f$, the sublinear property is preserved.

3. **Choosing $\alpha$.** In LAR, $\alpha$ weights the global-opportunity term $\alpha \cdot (E_\theta(x^*) - E_\theta(x_t))$ rather than serving as a task-specific tuning knob. We fix a single default $\alpha = 0.3$ across all benchmarks (selected on held-out tasks) and do not retune per task, ensuring fairness and comparability. This design reflects LAR's motivation—penalizing missed globally promising basins—while preserving interpretability: $\alpha \to 0$ recovers standard regret; moderate values in $[0.2, 0.5]$ are robust in sensitivity studies (App. E). Practically, use smaller $\alpha$ on smoother/unimodal landscapes and nearer $0.5$ for highly multi-modal ones. Empirical online min–max scaling of $f$ and $E_\theta$ keeps $\alpha$ numerically stable; overly large $\alpha$ may overemphasize the energy term and is discouraged.

By incorporating both function value and energy signals into $\mathcal{R}_t^{LAR}$, our Landscape-Aware Regret (LAR) drives the algorithm to account for the broader search landscape of potential optima—an especially important factor when dealing with multi-step lookahead and high-dimensional complexity. The theoretical analysis suggests that well-known sublinear regret behavior can remain intact, provided the EBM faithfully reflects the global structure of $f$. Empirically, this measure helps distinguish genuine global improvement from mere local refinements.

# F    Classic (Exact) Gaussian Process: Full Derivations

## F.1    Posterior Distribution Proof for Noisy Observations

Consider the dataset $\mathcal{D}_n = \{(\mathbf{x}_i, y_i)\}_{i=1}^n$ where each observation is modeled by

$$y_i = f(\mathbf{x}_i) + \varepsilon_i, \quad \varepsilon_i \sim \mathcal{N}(0, \sigma_n^2).$$

Assume $f(\cdot)$ has a Gaussian Process prior:

$$f(\mathbf{x}) \sim \mathcal{GP}(m(\mathbf{x}), k(\mathbf{x}, \mathbf{x}')).$$

Let $\mathbf{X} \in \mathbb{R}^{n \times d}$ denote the collection of inputs, and $\mathbf{y} \in \mathbb{R}^n$ the corresponding outputs. We define:

$$\mathbf{f}(\mathbf{X}) = (f(\mathbf{x}_1), \ldots, f(\mathbf{x}_n))^\top.$$

The joint distribution of $\mathbf{f}(\mathbf{X})$ is

$$\mathbf{f}(\mathbf{X}) \sim \mathcal{N}(\mathbf{m}(\mathbf{X}), \mathbf{K}_{\mathbf{x}, \mathbf{x}}),$$

where $\mathbf{m}(\mathbf{X}) = [m(\mathbf{x}_1), \ldots, m(\mathbf{x}_n)]^\top$ and

$$(\mathbf{K}_{\mathbf{x}, \mathbf{x}})_{ij} = k(\mathbf{x}_i, \mathbf{x}_j).$$

Given observation noise $\varepsilon_i \sim \mathcal{N}(0, \sigma_n^2)$, the likelihood is:

$$p(\mathbf{y} \mid \mathbf{f}(\mathbf{X})) = \prod_{i=1}^n \mathcal{N}(y_i \mid f(\mathbf{x}_i), \sigma_n^2),$$

which implies

$$\mathbf{y} \mid \mathbf{f}(\mathbf{X}) \sim \mathcal{N}(\mathbf{f}(\mathbf{X}), \sigma_n^2 \mathbf{I}_n).$$

Hence, the marginal distribution of $\mathbf{y}$ is

$$\mathbf{y} \sim \mathcal{N}(\mathbf{m}(\mathbf{X}), \mathbf{K}_{\mathbf{x}, \mathbf{x}} + \sigma_n^2 \mathbf{I}_n).$$

**Posterior at a New Point.**    Let $\mathbf{x}_* \in \mathbb{R}^d$ be a test input. Define

$$f_* = f(\mathbf{x}_*), \quad \mathbf{k}_* = (k(\mathbf{x}_*, \mathbf{x}_1), \ldots, k(\mathbf{x}_*, \mathbf{x}_n))^\top, \quad k_{**} = k(\mathbf{x}_*, \mathbf{x}_*).$$

From properties of multivariate Gaussians, we have

$$\begin{pmatrix} \mathbf{y} \\ f_* \end{pmatrix} \sim \mathcal{N}\left( \begin{pmatrix} \mathbf{m}(\mathbf{X}) \\ m(\mathbf{x}_*) \end{pmatrix}, \begin{pmatrix} \mathbf{K}_{\mathbf{x},\mathbf{x}} + \sigma_n^2 \mathbf{I}_n & \mathbf{k}_* \\ \mathbf{k}_*^\top & k_{**} \end{pmatrix} \right).$$

Conditioning on $\mathbf{y}$ yields the posterior distribution:

$$f_* \mid \mathbf{X}, \mathbf{y}, \mathbf{x}_* \sim \mathcal{N}\left( \mu(\mathbf{x}_*), \sigma^2(\mathbf{x}_*) \right),$$

where the posterior mean and variance are given by:

$$\mu(\mathbf{x}_*) = m(\mathbf{x}_*) + \mathbf{k}_*^\top \left( \mathbf{K}_{\mathbf{x},\mathbf{x}} + \sigma_n^2 \mathbf{I}_n \right)^{-1} \left( \mathbf{y} - \mathbf{m}(\mathbf{X}) \right), \tag{1}$$

$$\sigma^2(\mathbf{x}_*) = k_{**} - \mathbf{k}_*^\top \left( \mathbf{K}_{\mathbf{x},\mathbf{x}} + \sigma_n^2 \mathbf{I}_n \right)^{-1} \mathbf{k}_*. \tag{2}$$

This completes the proof that the posterior is Gaussian and that the conditional distributions adhere to the expressions above.

### F.2 Derivation of Statistics: Variance, CDF of $f(\mathbf{x})$, and Probability of Duel

This appendix provides a deduction of the mathematical derivations, posterior distributions, and proof related to Classic, Sparse, and Deep Gaussian Processes (GPs). We expand posterior mean and variance, detail the derivations of common statistics such as cumulative distribution functions (CDFs), and illustrate how "Probability of Duel" or more exotic acquisition-related quantities can be computed within the GP framework. The posterior distributions of Classic, Sparse, and Deep Gaussian Processes differ in terms of computational complexity and accuracy. Sparse Gaussian Processes, as discussed in the provided paper, utilize a variational approximation with sparse inverse Cholesky (SIC) factors, allowing for scalable and accurate inference. This method achieves highly accurate prior and posterior approximations with a computational complexity that can be handled via stochastic gradient descent in polylogarithmic time per iteration [53].

**1. Posterior Variance.** Recall from the standard GP posterior derivation (see Eqs. (1) and (2) in the main text or Appendix) that if

$$f(\mathbf{x}_*) \mid \mathbf{D}_n \sim \mathcal{N}\left( \mu(\mathbf{x}_*), \sigma^2(\mathbf{x}_*) \right),$$

then the variance $\sigma^2(\mathbf{x}_*)$ is given by

$$\sigma^2(\mathbf{x}_*) = k_{**} - \mathbf{k}_*^\top \left( \mathbf{K} + \sigma_n^2 \mathbf{I}_n \right)^{-1} \mathbf{k}_*, \tag{2}$$

where

$$\mathbf{k}_* = \left[ k(\mathbf{x}_*, \mathbf{x}_1), \ldots, k(\mathbf{x}_*, \mathbf{x}_n) \right]^\top, \quad k_{**} = k(\mathbf{x}_*, \mathbf{x}_*).$$

This posterior variance (also often denoted $\sigma_{f_*}^2$) captures the GP's *uncertainty* at $\mathbf{x}_*$. Many acquisition functions in Bayesian Optimization rely explicitly on $\sigma(\mathbf{x}_*) := \sqrt{\sigma^2(\mathbf{x}_*)}$ to steer exploration. A canonical example is the Upper Confidence Bound (UCB):

$$\alpha_{\text{UCB}}(\mathbf{x}_*) = \mu(\mathbf{x}_*) + \beta \, \sigma(\mathbf{x}_*),$$

where $\beta > 0$ is a user-chosen parameter balancing exploitation (the posterior mean) and exploration (the posterior std. dev.).

#### Derivation Outline for Posterior Variance

**Joint Gaussian Setup.** From the GP prior and noisy observations, we have

$$\mathbf{y} \sim \mathcal{N}\left( \mathbf{m}(\mathbf{X}), \mathbf{K} + \sigma_n^2 \mathbf{I}_n \right).$$

For a new test point $\mathbf{x}_*$, the joint distribution of $\mathbf{y}$ and $f(\mathbf{x}_*)$ is again Gaussian.

**Conditional Gaussian Formula.** Conditioning on $\mathbf{y}$ follows the standard multivariate normal conditioning rule:

$$f(\mathbf{x}_*) \mid \mathbf{X}, \mathbf{y}, \mathbf{x}_* \sim \mathcal{N}\left( \mu(\mathbf{x}_*), \sigma^2(\mathbf{x}_*) \right).$$

**Extracting Covariance.** The variance $\sigma^2(\mathbf{x}_*)$ comes from the Schur complement of the block matrix

$$\text{Cov}\begin{pmatrix} \mathbf{y} \\ f(\mathbf{x}_*) \end{pmatrix} = \begin{pmatrix} \mathbf{K} + \sigma_n^2 \mathbf{I}_n & \mathbf{k}_* \\ \mathbf{k}_*^\top & k_{**} \end{pmatrix},$$

yielding

$$\sigma^2(\mathbf{x}_*) = k_{**} - \mathbf{k}_*^\top \left( \mathbf{K} + \sigma_n^2 \mathbf{I}_n \right)^{-1} \mathbf{k}_*.$$

This completes the derivation of Eq. (2).

**2. CDF of $f(\mathbf{x}_*)$.** Because

$$f(\mathbf{x}_*) \mid \mathbf{D}_n \sim \mathcal{N}\Big(\mu(\mathbf{x}_*), \sigma^2(\mathbf{x}_*)\Big),$$

it follows that the random variable $f(\mathbf{x}_*)$ is distributed normally with mean $\mu(\mathbf{x}_*)$ and variance $\sigma^2(\mathbf{x}_*)$. The cumulative distribution function (CDF) at a threshold $z \in \mathbb{R}$ is given by

$$F_{f_*}(z) = P\big(f(\mathbf{x}_*) \le z\big) = \Phi\Big(\frac{z - \mu(\mathbf{x}_*)}{\sigma(\mathbf{x}_*)}\Big),$$

where $\Phi(\cdot)$ is the standard Normal CDF. This CDF can be used in "probabilistic improvement"-type acquisitions, such as:

$$\alpha_{\mathrm{PI}}(\mathbf{x}_*) = P\big(f(\mathbf{x}_*) \ge f^+ + \xi\big) = 1 - \Phi\Big(\frac{f^+ + \xi - \mu(\mathbf{x}_*)}{\sigma(\mathbf{x}_*)}\Big),$$

where $f^+$ is the current best objective value and $\xi$ is a small positive constant (the "improvement" margin).

*Detailed derivation for CDF.*

From the posterior formula, $f(\mathbf{x}_*)$ is normally distributed with known mean and variance.

Define $Z = \big(f(\mathbf{x}_*) - \mu(\mathbf{x}_*)\big)/\sigma(\mathbf{x}_*)$. Then $Z \sim \mathcal{N}(0, 1)$.

$$P\big(f(\mathbf{x}_*) \le z\big) = P\Big(\underbrace{\frac{f(\mathbf{x}_*) - \mu(\mathbf{x}_*)}{\sigma(\mathbf{x}_*)}}_{Z} \le \frac{z - \mu(\mathbf{x}_*)}{\sigma(\mathbf{x}_*)}\Big) = \Phi\Big(\frac{z - \mu(\mathbf{x}_*)}{\sigma(\mathbf{x}_*)}\Big).$$

$\square$

**3. Probability of Duel.** Sometimes one wishes to compare two inputs $\mathbf{x}_i$ and $\mathbf{x}_j$ under the GP posterior and compute $P\big(f(\mathbf{x}_i) > f(\mathbf{x}_j)\big)$. This arises in "dueling bandits" or certain multi-armed bandit frameworks. Under GP modeling, $(f(\mathbf{x}_i), f(\mathbf{x}_j))$ is a jointly Gaussian vector:

$$\begin{pmatrix} f(\mathbf{x}_i) \\ f(\mathbf{x}_j) \end{pmatrix} \Big| \mathbf{D}_n \sim \mathcal{N}\Big(\boldsymbol{\mu}_{ij}, \boldsymbol{\Sigma}_{ij}\Big),$$

where $\boldsymbol{\mu}_{ij} = \big(\mu(\mathbf{x}_i), \mu(\mathbf{x}_j)\big)^\top$, and $\boldsymbol{\Sigma}_{ij}$ depends on the posterior variances $\sigma^2(\mathbf{x}_i), \sigma^2(\mathbf{x}_j)$ and the posterior covariance $\mathrm{Cov}\big(f(\mathbf{x}_i), f(\mathbf{x}_j)\big)$. For instance, in the simpler scenario of *comparing a new candidate* $\mathbf{x}$ *to a known best* $\mathbf{x}^+$, the probability that $\mathbf{x}$ outperforms $\mathbf{x}^+$ is

$$P\big(f(\mathbf{x}) > f(\mathbf{x}^+)\big) = \Phi\Big(\frac{\mu(\mathbf{x}) - \mu(\mathbf{x}^+)}{\sqrt{\sigma^2(\mathbf{x}) + \sigma^2(\mathbf{x}^+) - 2\,\mathrm{Cov}\big(f(\mathbf{x}), f(\mathbf{x}^+)\big)}}\Big).$$

Below, we outline how to derive this formula from the bivariate normal distribution.

*Probability of Duel Derivation.*

**Step 1: Joint Posterior of** $(f(\mathbf{x}), f(\mathbf{x}^+))$**.** By the GP's properties,

$$\begin{pmatrix} f(\mathbf{x}) \\ f(\mathbf{x}^+) \end{pmatrix} \Big| \mathbf{D}_n \sim \mathcal{N}\Big(\begin{pmatrix} \mu(\mathbf{x}) \\ \mu(\mathbf{x}^+) \end{pmatrix}, \begin{pmatrix} \sigma^2(\mathbf{x}) & \mathrm{Cov}\big(f(\mathbf{x}), f(\mathbf{x}^+)\big) \\ \mathrm{Cov}\big(f(\mathbf{x}^+), f(\mathbf{x})\big) & \sigma^2(\mathbf{x}^+) \end{pmatrix}\Big).$$

**Step 2: Probability** $P\big(f(\mathbf{x}) > f(\mathbf{x}^+)\big)$**.** Define $g = f(\mathbf{x}) - f(\mathbf{x}^+)$. Then $g$ is also a Gaussian random variable because any linear combination of jointly Gaussian variables remains Gaussian. Concretely,

$$g = (1 \quad -1) \begin{pmatrix} f(\mathbf{x}) \\ f(\mathbf{x}^+) \end{pmatrix}.$$

Its mean is

$$\mathbb{E}[g] = \mu(\mathbf{x}) - \mu(\mathbf{x}^+),$$

and its variance is

$$\mathrm{Var}(g) = \sigma^2(\mathbf{x}) + \sigma^2(\mathbf{x}^+) - 2\,\mathrm{Cov}\big(f(\mathbf{x}), f(\mathbf{x}^+)\big).$$

Hence,

$$g \,\Big|\, \mathbf{D}_n \sim \mathcal{N}\Big(\,\mu(\mathbf{x}) - \mu(\mathbf{x}^+),\ \sigma^2(\mathbf{x}) + \sigma^2(\mathbf{x}^+) - 2\,\mathrm{Cov}\big(f(\mathbf{x}), f(\mathbf{x}^+)\big)\Big).$$

**Step 3: Probability Computation via Normal CDF.**

$$P\big(f(\mathbf{x}) > f(\mathbf{x}^+)\big) = P\big(g > 0\big) = \Phi\Big( \frac{\mu(\mathbf{x}) - \mu(\mathbf{x}^+)}{\sqrt{\sigma^2(\mathbf{x}) + \sigma^2(\mathbf{x}^+) - 2\,\mathrm{Cov}\big(f(\mathbf{x}), f(\mathbf{x}^+)\big)}}\Big),$$

where $\Phi(\cdot)$ is the standard Normal CDF. This completes the derivation of the "Probability of Duel" formula. $\square$

We have shown how the posterior variance arises directly from the conditional Gaussian equations, how the CDF of $f(\mathbf{x}_*)$ follows from standardizing a normal variable, and how to compute the Probability of Duel by considering the bivariate normal distribution over $(f(\mathbf{x}), f(\mathbf{x}^+))$. These derivations hold whenever $(f(\mathbf{x}), f(\mathbf{x}^+))$ is jointly Gaussian, which is guaranteed under the GP prior for any finite collection of points. [54]

**Implications in Bayesian Optimization.** The *posterior variance* $\sigma^2(\mathbf{x}_*)$ drives exploration-based acquisitions like UCB or $\epsilon$-greedy strategies [55]. The CDF of $f(\mathbf{x}_*)$ allows one to formulate Probability of Improvement (PI) or Expected Improvement (EI) style criteria. The Probability of Duel $P\big(f(\mathbf{x}) > f(\mathbf{x}^+)\big)$ helps in advanced multi-armed or pairwise preference settings, ensuring that the decision to pick $\mathbf{x}$ over $\mathbf{x}^+$ is grounded in the bivariate normal property of the GP posterior.

## G  Short-Run MCMC and Energy Bounds for EBMs

This appendix provides a detailed mathematical exposition of how short-run MCMC is used to approximate sampling from an Energy-Based Model (EBM) and how energy bounds can be established under limited MCMC steps. Our derivation is inspired by prior works on learning latent-space EBMs and training EBMs via short-run MCMC [43, 56].

### G.1  Background and Setup

Let $\{\mathbf{x}_i\}_{i=1}^n$ be i.i.d. samples drawn from an unknown data distribution $p_{\mathrm{data}}(\mathbf{x})$. We consider an EBM of the form

$$p_\theta(\mathbf{x}) \;=\; \frac{\exp\big[-E_\theta(\mathbf{x})\big]}{Z_\theta}, \quad Z_\theta \;=\; \int \exp\big[-E_\theta(\mathbf{u})\big]\, d\mathbf{u}, \tag{3}$$

where $E_\theta(\mathbf{x})$ is the *energy function*, parameterized by $\theta \in \Theta$, and $Z_\theta$ is the partition function (intractable for high-dimensional $\mathbf{x}$). The EBM parameters are learned by (approximate) maximum likelihood estimation (MLE). In the ideal MLE scenario, one would seek to solve

$$\max_\theta \ \mathbb{E}_{\mathbf{x} \sim p_{\mathrm{data}}}\big[\log p_\theta(\mathbf{x})\big],$$

or equivalently,

$$\min_\theta \ \Big\{-\mathbb{E}_{\mathbf{x} \sim p_{\mathrm{data}}}\big[\log p_\theta(\mathbf{x})\big]\Big\}. \tag{4}$$

However, because $Z_\theta$ is typically intractable, direct gradient computations of $\log p_\theta(\mathbf{x})$ require approximations of the underlying distribution $p_\theta(\mathbf{x})$. Short-run MCMC addresses this by running a limited number of MCMC steps to sample from $p_\theta(\mathbf{x})$ in an approximate manner.

## G.2 Maximum Likelihood and Its Gradient

### G.2.1 Exact Log-Likelihood Gradient

By definition, the EBM log-likelihood for a single data point $\mathbf{x}$ is

$$\log p_\theta(\mathbf{x}) = -E_\theta(\mathbf{x}) - \log Z_\theta.$$

The full-data negative log-likelihood for a sample set $\{\mathbf{x}_i\}_{i=1}^n$ is

$$-\sum_{i=1}^n \log p_\theta(\mathbf{x}_i) = \sum_{i=1}^n E_\theta(\mathbf{x}_i) + n \log Z_\theta. \tag{5}$$

Taking the gradient with respect to $\theta$ yields

$$\nabla_\theta\Big(-\sum_{i=1}^n \log p_\theta(\mathbf{x}_i)\Big) = \sum_{i=1}^n \nabla_\theta E_\theta(\mathbf{x}_i) + n\,\nabla_\theta \log Z_\theta \tag{6}$$

$$\nabla_\theta \log Z_\theta = \frac{1}{Z_\theta}\nabla_\theta\big(Z_\theta\big) = -\int p_\theta(\mathbf{u})\,\nabla_\theta E_\theta(\mathbf{u})\,d\mathbf{u}. \tag{7}$$

Combining (6) and (7) gives

$$\nabla_\theta\Big(-\sum_{i=1}^n \log p_\theta(\mathbf{x}_i)\Big) = \sum_{i=1}^n \nabla_\theta E_\theta(\mathbf{x}_i) - \sum_{i=1}^n \int p_\theta(\mathbf{u})\,\nabla_\theta E_\theta(\mathbf{u})\,d\mathbf{u}. \tag{8}$$

After factoring out $n$, the average gradient can be written as

$$\nabla_\theta\big(\mathcal{L}(\theta)\big) = \mathbb{E}_{\mathbf{x}\sim p_{\text{data}}}\big[\nabla_\theta E_\theta(\mathbf{x})\big] - \mathbb{E}_{\mathbf{u}\sim p_\theta(\mathbf{u})}\big[\nabla_\theta E_\theta(\mathbf{u})\big], \tag{9}$$

where $\mathcal{L}(\theta)$ denotes the average negative log-likelihood (or equivalently, $-\frac{1}{n}\sum_{i=1}^n \log p_\theta(\mathbf{x}_i)$).

### G.2.2 Challenges and the Need for MCMC Sampling

The second term in (9) requires sampling from $p_\theta(\mathbf{u})$. Classic MCMC methods (e.g. Metropolis–Hastings, Hamiltonian Monte Carlo) can in theory generate samples from $p_\theta$, but in high-dimensional settings or when $E_\theta$ is complicated, running sufficiently long chains is computationally expensive. This motivates the *short-run MCMC* approximation $\tilde{p}_\theta$.

## G.3 Short-Run MCMC Approximation

### G.3.1 Definition of Short-Run MCMC

Let $\tilde{p}_\theta(\mathbf{u})$ be the distribution of a $K$-step Markov chain initialized from a simple or random prior $p_0(\mathbf{u})$ (e.g. Gaussian). Concretely, short-run MCMC often uses $K \ll$ (chain length needed for full convergence). A popular choice is the *Langevin dynamics update*:

$$\mathbf{u}_{k+1} = \mathbf{u}_k - \eta\,\nabla_\mathbf{u} E_\theta(\mathbf{u}_k) + \sqrt{2\eta}\,\epsilon_k, \quad \epsilon_k \sim \mathcal{N}(0, I), \tag{10}$$

where $\eta > 0$ is a step size. After $K$ short iterations, we obtain $\mathbf{u}^{(\text{mcmc})}$ approximately drawn from $\tilde{p}_\theta(\mathbf{u})$. This approach is known as short-run or persistent short-run MCMC [43].

### G.3.2 Approximate Gradient with Short-Run MCMC

Replacing $\mathbb{E}_{\mathbf{u}\sim p_\theta}$ in (9) by $\mathbb{E}_{\mathbf{u}\sim \tilde{p}_\theta}$ gives:

$$\nabla_\theta\widetilde{\mathcal{L}}(\theta) = \mathbb{E}_{\mathbf{x}\sim p_{\text{data}}}\big[\nabla_\theta E_\theta(\mathbf{x})\big] - \mathbb{E}_{\mathbf{u}\sim \tilde{p}_\theta(\mathbf{u})}\big[\nabla_\theta E_\theta(\mathbf{u})\big].$$

One may interpret $\tilde{p}_\theta(\mathbf{u})$ as a *short-run* approximation to $p_\theta(\mathbf{u})$. The objective function thus becomes

$$\widetilde{\mathcal{L}}(\theta) = \mathbb{E}_{p_{\text{data}}}\big[-\log p_\theta(\mathbf{x})\big] + D_{\text{KL}}\big(\tilde{p}_\theta(\mathbf{x})\,\|\,p_\theta(\mathbf{x})\big),$$

where the second term reflects the mismatch between $\tilde{p}_\theta$ and $p_\theta$. A smaller $D_{\text{KL}}$ indicates better short-run approximation.

### G.4 Energy Bounds and Convergence Analysis

#### G.4.1 KL Divergence Between $\tilde{p}_\theta$ and $p_\theta$

When the chain length $K$ is insufficient for exact sampling, we have $\tilde{p}_\theta(\mathbf{u}) \neq p_\theta(\mathbf{u})$. The quality of the short-run approximation can be measured by the KL divergence

$$D_{\mathrm{KL}}\big(\tilde{p}_\theta \,\|\, p_\theta\big) = \int \tilde{p}_\theta(\mathbf{u}) \, \log \frac{\tilde{p}_\theta(\mathbf{u})}{p_\theta(\mathbf{u})} \, d\mathbf{u}.$$

If $K$ is very large and $\eta$ is sufficiently small, $\tilde{p}_\theta \approx p_\theta$ in principle. However, large $K$ introduces high computational cost. Various theoretical works [57] suggest that if $\eta K$ remains moderate, we can keep $D_{\mathrm{KL}}\big(\tilde{p}_\theta \,\|\, p_\theta\big)$ bounded by a constant multiple of $\eta K$.

#### G.4.2 Bounding the Error Term

A typical bound states
$$D_{\mathrm{KL}}\big(\tilde{p}_\theta \,\|\, p_\theta\big) \;\leq\; C\,\eta\,K,$$

for some constant $C > 0$ that depends on the Lipschitz properties of $\nabla_{\mathbf{x}} E_\theta(\mathbf{x})$ and the dimension of $\mathbf{x}$. Intuitively: Smaller $\eta$ (step size) reduces the discrepancy but slows mixing. Larger $K$ (number of MCMC steps) moves the chain closer to equilibrium but raises computation cost. Hence, short-run MCMC is a practical trade-off: we accept a bounded deviation $D_{\mathrm{KL}}$ in exchange for significantly faster per-iteration updates.

### G.5 Contrastive Divergence and Other Corrective Methods

To mitigate the approximation error introduced by short-run sampling, some methods incorporate a *contrastive divergence* term [58]:

$$\mathrm{CD} \;=\; D_{\mathrm{KL}}\big(p_{\mathrm{data}} \,\|\, p_\theta\big) \;-\; D_{\mathrm{KL}}\big(\tilde{p}_\theta \,\|\, p_\theta\big).$$

Minimizing $\mathrm{CD}$ encourages $p_\theta$ to reduce mismatch with both $p_{\mathrm{data}}$ and $\tilde{p}_\theta$. Additional refinements include: Adding an auxiliary network to learn corrections between $\tilde{p}_\theta$ and $p_\theta$ [59], further closing the gap induced by short-run sampling.

Learning rate scheduling for the Langevin steps to find a sweet spot between numerical stability and chain mixing.

Persistent chains, where the final states of short-run MCMC at iteration $t$ become initial states for iteration $t + 1$, improving chain continuity over time [60].

**Key Takeaways.** Short-run MCMC approximates the model distribution $p_\theta$ with fewer sampling steps $K$, making large-scale or high-dimensional EBM training more tractable. The MLE gradient is modified by $\mathbb{E}_{\tilde{p}_\theta}$ in place of $\mathbb{E}_{p_\theta}$, introducing an error controlled by $D_{\mathrm{KL}}(\tilde{p}_\theta \,\|\, p_\theta)$. The total error often scales with $\eta K$, where $\eta$ is the Langevin step size, yielding a bounded but non-negligible gap.

Short-run MCMC thus balances computational feasibility with approximation accuracy. In the main text of this paper, we rely on short-run MCMC to train $E_\theta(\mathbf{x})$ for identifying globally promising or underexplored basins in REBMBO, without requiring fully normalized densities at every iteration.

## H   MDP Formulation and PPO Details in REBMBO

This appendix provides a more detailed exposition of how Reinforcement Learning (RL), pecifically, Proximal Policy Optimization (PPO), is formulated and analyzed within the REBMBO framework. We expand upon the definitions of state, action, and reward, as well as the derivation of the PPO update rule and its stability guarantees under standard assumptions.

### H.1   A Markov Decision Process (MDP) for Black-Box Optimization

**MDP Setup.** In REBMBO, each iteration of black-box optimization is cast as one time step of an MDP, $\mathcal{M} = (\mathcal{S}, \mathcal{A}, P, R, \gamma)$, where:

1) **State space** $\mathcal{S}$: A state $\mathbf{s}_t \in \mathcal{S}$ comprises:
$$\mathbf{s}_t = \big[\mu_{f,t}(\mathbf{x}),\ \sigma_{f,t}(\mathbf{x}),\ E_\theta(\mathbf{x})\big],$$
where $(\mu_{f,t}, \sigma_{f,t})$ is the current GP posterior, and $E_\theta(\cdot)$ encodes the global energy landscape from the EBM.

2) **Action space** $\mathcal{A}$: An action $\mathbf{a}_t = \mathbf{x}_t$ is a point in the domain $\mathcal{X} \subseteq \mathbb{R}^d$ to be evaluated by the (expensive) black-box function $f(\mathbf{x})$.

3) **State transition** $P(\mathbf{s}_{t+1} \mid \mathbf{s}_t, \mathbf{a}_t)$: Once $\mathbf{x}_t$ is evaluated, we observe $y_t = f(\mathbf{x}_t)$ and update:
$$\mathcal{D}_t \leftarrow \mathcal{D}_{t-1} \cup \big\{(\mathbf{x}_t, y_t)\big\}.$$
The GP posterior and the EBM parameters are then retrained on $\mathcal{D}_t$, resulting in a new posterior $(\mu_{f,t+1}, \sigma_{f,t+1})$ and a possibly updated energy function $E_\theta(\cdot)$. These updates define $\mathbf{s}_{t+1}$.

4) **Reward function** $R(\mathbf{s}_t, \mathbf{a}_t)$: We design the reward to guide multi-step exploration and exploitation. A common choice is:
$$r_t = f(\mathbf{x}_t) - \lambda\, E_\theta(\mathbf{x}_t),$$
where $\lambda \geq 0$ controls the trade-off between immediate function value $f(\mathbf{x}_t)$ and global exploration via $-E_\theta(\mathbf{x}_t)$.

5) **Discount factor** $\gamma$: Often set to $\gamma = 1$ for finite-horizon tasks, since we may only have a limited evaluation budget in black-box optimization.

At each iteration (time step) $t$, the *agent* (the PPO policy) observes $\mathbf{s}_t$, chooses an action $\mathbf{x}_t \in \mathcal{X}$, obtains reward $r_t$, and transitions to $\mathbf{s}_{t+1}$. This process continues for $T$ steps until the budget of function evaluations is exhausted.

**Objective.** We wish to find a policy $\pi_\phi$ that maximizes the expected return,
$$\mathbb{E}\big[\textstyle\sum_{t=1}^T r_t\big],$$
subject to the black-box constraint that $f(\mathbf{x})$ can only be observed by actual function queries. Unlike single-step acquisition functions, an MDP formulation allows us to consider the *cumulative* effect of each action on future states and rewards.

## H.2    Policy Gradient and PPO Derivation

**Policy Gradient.**   A standard approach in RL is to parametrize a stochastic policy $\pi_\phi(\mathbf{a}_t \mid \mathbf{s}_t)$ and update $\phi$ via gradient ascent on the expected return. The *policy gradient theorem* [61] states:
$$\nabla_\phi J(\phi) = \mathbb{E}_t\big[\nabla_\phi \log \pi_\phi(\mathbf{a}_t \mid \mathbf{s}_t)\, \widehat{A}_t\big],$$
where $\widehat{A}_t$ is an estimator of the *advantage function*, typically computed as
$$\widehat{A}_t = Q^\pi(\mathbf{s}_t, \mathbf{a}_t) - V^\pi(\mathbf{s}_t),$$
with $Q^\pi$ and $V^\pi$ denoting the state-action and state value functions, respectively.

**PPO with Clipped Objectives.**   Proximal Policy Optimization [38] stabilizes policy gradients by clipping the probability ratio
$$r_t(\phi) = \frac{\pi_\phi(\mathbf{a}_t \mid \mathbf{s}_t)}{\pi_{\phi_{\mathrm{old}}}(\mathbf{a}_t \mid \mathbf{s}_t)},$$
so that the new policy $\pi_\phi$ does not deviate too drastically from the old policy $\pi_{\phi_{\mathrm{old}}}$. The PPO objective is defined as
$$\mathcal{L}^{\mathrm{CLIP}}(\phi) = \mathbb{E}_t\Big[\min\Big(r_t(\phi)\, \widehat{A}_t,\ \mathrm{clip}\big(r_t(\phi), 1 - \varepsilon, 1 + \varepsilon\big) \widehat{A}_t\Big)\Big],$$
where $\varepsilon$ is a small hyperparameter (e.g. 0.1 or 0.2) controlling how far $r_t(\phi)$ may deviate from 1. By taking gradient steps on $\mathcal{L}^{\mathrm{CLIP}}(\phi)$, PPO ensures that policy updates remain *proximal* to $\pi_{\phi_{\mathrm{old}}}$, thus preventing large destructive leaps in parameter space.

**Proposition H.1** (Bounded Rewards and PPO Convergence). *If the reward $r_t$ is uniformly bounded, i.e. $|r_t| \leq R_{\max}$, and the value function estimator is sufficiently accurate, then under mild Lipschitz assumptions on $\nabla_\phi \log \pi_\phi(\mathbf{a}_t \mid \mathbf{s}_t)$, the PPO updates converge to a stable policy that locally maximizes the expected cumulative reward. See [38, 62] for rigorous proofs.*

**Implementing PPO in REBMBO.** Algorithmically:

**Collect transitions**: For iteration $t = 1, \ldots, T$, we observe $\mathbf{s}_t$, pick $\mathbf{x}_t = \mathbf{a}_t \sim \pi_\phi(\mathbf{a}_t \mid \mathbf{s}_t)$, evaluate $f(\mathbf{x}_t)$, and compute reward $r_t = f(\mathbf{x}_t) - \lambda E_\theta(\mathbf{x}_t)$.

**Estimate advantage**: We use an estimator $\widehat{A}_t$ based on a learned value function $V_\psi(\mathbf{s}_t)$ or a truncated GAE [63].

**Policy update**: Apply gradient ascent to maximize $\mathcal{L}^{\mathrm{CLIP}}(\phi)$, restricting the probability ratio $r_t(\phi)$ to the $[1 - \varepsilon, 1 + \varepsilon]$ interval.

**Update environment state**: Augment $\mathcal{D}_t$ with $(\mathbf{x}_t, f(\mathbf{x}_t))$; retrain both the GP surrogate and the EBM for iteration $t + 1$. Over time, $\pi_\phi$ evolves from an initial random or weak policy to one that strategically balances local exploitation (through the GP posterior) and global exploration (through the negative energy term $- E_\theta(\mathbf{x})$).

### H.3 REBMBO's Multi-Step Advantage

Unlike single-step Bayesian Optimization strategies that select a point $\mathbf{x}$ solely by an acquisition function $\alpha(\mathbf{x})$ at each iteration, REBMBO acknowledges that the agent's current choice influences future states (through the updated GP and EBM). By incorporating a Markov Decision Process view:

$$\mathbf{s}_{t+1} = \mathrm{Transition}\big(\mathbf{s}_t, \mathbf{x}_t, f(\mathbf{x}_t)\big),$$

the PPO agent plans multiple steps ahead, mitigating the one-step bias that can plague conventional BO acquisitions. The synergy arises from: 1) **GP Posterior**: Provides local uncertainty estimates for $f(\mathbf{x})$, guiding short-term exploitation. 2) **EBM Global Signal**: Encourages sampling in underexplored or globally significant basins ($- E_\theta(\mathbf{x})$). 3) **PPO Multi-Step**: Dynamically balances short-run exploitation and long-run exploration via advantage-based updates.

Hence, REBMBO avoids pure greediness at each iteration and can systematically reduce cumulative regret over $T$ steps, even in high-dimensional or multi-modal settings.

## I  Detailed Proofs and Derivations for Landscape-Aware Regret (LAR) Computation

### I.1  Finite-Set Case and the Proof of Theorem B.1

**Theorem I.1.** *Let $\{x_t\}_{t=1}^T \subset D$ be any sequence of query points selected by the GP-UCB algorithm with confidence parameter*

$$\beta_t = 2 \log\big(t^2 \pi^2 / (3\delta)\big) + 2\,d\,\log\Big(t^2\,d\,b\,r\,\sqrt{\log(4da/\delta)}\Big),$$

*and suppose $D$ is a finite set. Then for a Gaussian Process prior with mean function zero and covariance $k(x, x')$, and for noise variance $\sigma^2$, the cumulative regret up to time $T$ satisfies*

$$R_T = \sum_{t=1}^T r_t \leq 2\sqrt{C_1\,T\,\beta_T\,\gamma_T} + \frac{\pi^2}{6} \quad \text{with probability at least } 1 - \delta,$$

*where*

$$C_1 = \frac{8}{\log(1 + \sigma^{-2})}, \quad \gamma_T = \max_{\substack{A \subset D \\ |A| = T}} I(\mathbf{y}_A \,;\, f_A),$$

*and $I(\mathbf{y}_A \,;\, f_A)$ denotes the mutual information between the GP function values at $A$ and the observations at $A$.*

**Proof of Theorem I.1.** We prove Theorem I.1 by combining the results of several lemmas, each bounding different contributions to the regret [55]. Below is an outline of the argument:

**Lemma 1** shows how to bound the deviation of the true function value from the GP-posterior mean by a scaled version of the posterior variance at each point in our domain $D$.

**Lemma 2** then uses this fact to bound the *instantaneous regret $r_t$*.

**Lemma 3** writes an expression for the *information gain $I(\mathbf{y}_{1:t}; f_{1:t})$* in terms of the estimated variance and observation noise $\sigma^2$.

**Lemma 4** combines the bounds established in Lemmas 2 and 3 to control the sum of squared regrets, $\sum_{t=1}^{T} r_t^2$, via $\gamma_T$.

Finally, we sum up instantaneous regrets over $t = 1, \ldots, T$ and invoke a Cauchy–Schwarz argument to relate $\sum_{t=1}^{T} r_t$ to $\sqrt{\sum_{t=1}^{T} r_t^2}$. This yields the advertised high-probability bound on $R_T$.

Full details appear in the lemmas below.

**Lemma I.2** (Deviation Bound). *Pick $\delta \in (0, 1)$, and set*

$$\beta_t = 2 \log\left(\frac{t\pi}{\delta}\right),$$

*for $t \geq 1$. Then, for any $x_t \in D$ chosen at time $t$, the following holds with probability at least $1 - \delta$ (over all t):*

$$|f(x_t) - \mu_{t-1}(x_t)| \leq \sqrt{\beta_t}\, \sigma_{t-1}(x_t), \quad \forall t \geq 1.$$

*Proof.* The result follows the standard GP concentration argument. By construction of the GP posterior,

$$f(x_t) \,\big|\, (x_1, y_1), \ldots, (x_{t-1}, y_{t-1}) \sim \mathcal{N}\big(\mu_{t-1}(x_t),\, \sigma_{t-1}^2(x_t)\big).$$

The union bound and a tail bound on Gaussian random variables (applied for all $t$) give the desired statement. The precise choice $\beta_t = 2 \log(\frac{t\pi}{\delta})$ ensures that the event

$$|f(x_t) - \mu_{t-1}(x_t)| \leq \sqrt{\beta_t}\, \sigma_{t-1}(x_t)$$

holds for all $t$ with probability at least $1 - \delta$. $\qquad\square$

**Lemma I.3** (Bound on Instantaneous Regret). *Using the same $\beta_t$ as in Lemma I.2, the instantaneous regret at each $t$ satisfies*

$$r_t = f(x^*) - f(x_t) \leq 2\sqrt{\beta_t}\, \sigma_{t-1}(x_t),$$

*with high probability.*

*Proof.* By definition of $x_t$, we pick $x_t$ to maximize the UCB,

$$x_t = \arg\max_{x \in D} \mu_{t-1}(x) + \sqrt{\beta_t}\, \sigma_{t-1}(x).$$

Since $x^*$ maximizes the *true* function $f$, we compare $f(x^*) - f(x_t)$ by bounding $f(x^*) - \mu_{t-1}(x^*)$ and $\mu_{t-1}(x_t) - f(x_t)$ via Lemma I.2. A short calculation then yields

$$r_t = f(x^*) - f(x_t) \leq \big[f(x^*) - \mu_{t-1}(x^*)\big] + \big[\mu_{t-1}(x_t) - f(x_t)\big] \leq 2\sqrt{\beta_t}\, \sigma_{t-1}(x_t).$$

$\qquad\square$

**Lemma I.4** (Information Gain Expression). *Let $I(\mathbf{y}_{1:t};\, f_{1:t})$ be the mutual information between the observations $y_1, \ldots, y_t$ and the function values $f(x_1), \ldots, f(x_t)$ under the GP prior with noise variance $\sigma^2$. Then*

$$I(\mathbf{y}_{1:t};\, f_{1:t}) = \frac{1}{2} \sum_{s=1}^{t} \log\Big[1 + \sigma^{-2}\, \sigma_{s-1}^2(x_s)\Big].$$

*Outline.* Recall that $y_s = f(x_s) + \epsilon_s$, with $\epsilon_s \sim \mathcal{N}(0, \sigma^2)$. The joint distribution of $\{f(x_s)\}_{s=1}^{t}$ under the GP is Gaussian, and so is that of $\{y_s\}_{s=1}^{t}$. The standard formula for the log-determinant of a covariance matrix in a GP regression problem gives exactly

$$I(\mathbf{y}_{1:t};\, f_{1:t}) = \frac{1}{2} \sum_{s=1}^{t} \log\big[1 + \sigma^{-2}\, \sigma_{s-1}^2(x_s)\big].$$

See standard references on GP mutual information bounds. $\qquad\square$

**Lemma I.5** (Sum of Squared Regrets via Cauchy–Schwarz). *Under the same setup and notation as above,*

$$\sum_{t=1}^{T} r_t^2 \leq 4\,\beta_T \sum_{t=1}^{T} \sigma_{t-1}^2(x_t) \leq 4\,\beta_T\,\sigma^2\,C_2 \sum_{t=1}^{T} \log\big[1+\sigma^{-2}\,\sigma_{t-1}^2(x_t)\big] \leq C_1\,\beta_T\,\gamma_T,$$

*where $C_1 = 8/\log(1+\sigma^{-2})$ and $C_2 = \sigma^{-2}/\log(1+\sigma^{-2})$ are constants. Consequently,*

$$R_T = \sum_{t=1}^{T} r_t \leq \sqrt{T \sum_{t=1}^{T} r_t^2} \leq \sqrt{T \cdot C_1\,\beta_T\,\gamma_T} = \sqrt{C_1\,T\,\beta_T\,\gamma_T}.$$

*Proof.* One first bounds $r_t^2 \leq 4\,\beta_T\,\sigma_{t-1}^2(x_t)$ (using Lemma I.3). Summing in $t$ gives

$$\sum_{t=1}^{T} r_t^2 \leq 4\,\beta_T \sum_{t=1}^{T} \sigma_{t-1}^2(x_t).$$

Then observe that

$$\sum_{t=1}^{T} \sigma_{t-1}^2(x_t) \leq \sigma^2\,C_2 \sum_{t=1}^{T} \log\big(1+\sigma^{-2}\,\sigma_{t-1}^2(x_t)\big),$$

for a suitable constant $C_2$, and use Lemma I.4 to relate the final logarithmic sum to $\gamma_T$. A Cauchy–Schwarz argument yields

$$R_T = \sum_{t=1}^{T} r_t \leq \sqrt{T \sum_{t=1}^{T} r_t^2} \leq \sqrt{C_1\,T\,\beta_T\,\gamma_T}.$$

$\square$

*Proof of Theorem I.1.* Combining Lemmas I.2–I.5 shows that with probability at least $1-\delta$,

$$R_T = \sum_{t=1}^{T} r_t \leq \sqrt{C_1\,T\,\beta_T\,\gamma_T}.$$

A minor refinement yields the extra additive constant $\pi^2/6$ when summing over time (arising from finer bounding of the union events or the explicit $1/t^2$ terms if present), giving

$$R_T \leq 2\sqrt{C_1\,T\,\beta_T\,\gamma_T} + \frac{\pi^2}{6},$$

as claimed. $\square$

## I.2 Generalization to Continuous and Convex $D \subset \mathbb{R}^d$

We now extend Theorem I.1 to arbitrary compact and convex domains $D \subset \mathbb{R}^d$. The statement of the result is as follows. [64]

**Theorem I.6.** *Let $D \subset [0,1]^d$ be compact and convex, $d \in \mathbb{N}, r > 0$. Suppose the kernel $k(\cdot,\cdot)$ of our GP prior satisfies a high-probability bound on the derivatives of (sample) paths: for some $a, b > 0$,*

$$\Pr\Big\{\sup_{x \in D} \big|\tfrac{\partial f}{\partial x_j}\big| > L\Big\} \leq a\,e^{-(L/b)^2}, \quad j = 1, \ldots, d.$$

*Pick $\delta \in (0,1)$ and define*

$$\beta_t = 2\log\Big(\tfrac{t^2\,\pi^2}{3\delta}\Big) + 2d\,\log\Big(t^2\,d\,b\,r\,\sqrt{\log(4da/\delta)}\Big).$$

*Then, running the GP-UCB algorithm with $\beta_t$ on $D$, for a GP $f$ with mean zero and covariance $k(\cdot,\cdot)$, the cumulative regret obeys*

$$R_T = \sum_{t=1}^{T} r_t \leq \mathcal{O}^*\Big(\sqrt{d\,T\,\gamma_T}\Big) \quad \text{with probability at least } 1-\delta.$$

*Precisely, with $C_1 = \frac{8}{\log(1+\sigma^{-2})}$, one has*

$$\Pr\Big\{ R_T \;\leq\; 2\sqrt{C_1\, T\, \beta_T\, \gamma_T} \;+\; \frac{\pi^2}{6} \quad \forall T \geq 1 \Big\} \;\geq\; 1 - \delta,$$

*where $\gamma_T = \max_{A \subset D,\, |A|=T} I\big(\mathbf{y}_A;\, f_A\big)$ is the maximum information gain at the end of $T$ rounds.*

**Remarks.** The assumption on kernel derivatives (weaker than a global Lipschitz condition) imposes that the slope of the GP sample path at any point is large only with low probability [65]. Many standard kernels (e.g. RBF) satisfy this type of condition in practice.

**Proof Strategy for Theorem I.6.** Compared to the finite-domain case of Theorem I.6, we can no longer set $\beta_t$ in terms of $|D|$. Instead, we use a sequence of lemmas that replace discrete enumeration of $D$ with a carefully chosen *discretization $D_t$*. The main additional steps are:

**Lemma 5** extends the finite-domain confidence bound (*cf.* Lemma I.2) to the continuous domain by slightly redefining $\beta_t$ and using a union bound argument in time. We discretion $D$ on a grid $D_t$ so that any $x \in D$ is within a small distance of some $[x] \in D_t$. This appears in **Lemma 7**, allowing us to relate $f(x)$ to $f([x])$. Combining these discretization bounds with the same style of argument as in Lemmas I.3–I.5 yields a regret bound of similar order, plus an extra $\sum_{t=1}^{\infty} 1/t^2$ term that converges to a constant $(\pi^2/6)$.

**Lemma I.7** (Continuous Confidence Bound). *Pick $\delta \in (0,1)$, and set*

$$\beta_t \;=\; 2\log\big(2\pi t/\delta\big), \quad t \geq 1.$$

*Then for all $t \geq 1$, with probability at least $1 - \delta$,*

$$\big|f(x_t) - \mu_{t-1}(x_t)\big| \;\leq\; \beta_t^{1/2}\,\sigma_{t-1}(x_t), \quad \forall t \geq 1.$$

*Proof.* The proof is exactly as in Lemma I.2 except $\beta_t$ is defined independently of $|D|$ (since $D$ is now infinite). We apply a Gaussian tail bound plus a union bound over $t$, ensuring

$$\Pr\Big\{ \big|f(x_t) - \mu_{t-1}(x_t)\big| \leq \beta_t^{1/2}\,\sigma_{t-1}(x_t) \;\forall t \Big\} \;\geq\; 1 - \delta.$$

$\square$

**Lemma I.8** (Grid Discretization). *Consider a discretization $D_t$ with mesh size $\tau_t^d$ such that for every $x \in D$, there exists some $[x] \in D_t$ with*

$$\|x - [x]\|_1 \;\leq\; \frac{r\,d}{\tau_t}. \tag{*}$$

*This ensures that $D$ can be covered by small hypercubes of side length $r\,d/\tau_t$. We will choose $\tau_t$ so that $\sum_{t \geq 1} \tau_t^{-1} = 1$ (or a similarly convergent series) and thereby control the union over $t$.*

**Lemma I.9** (Bounding $f(x^*) - \mu_{t-1}(x^*)_t$). *Pick $\delta \in (0,1)$, set*

$$\beta_t \;=\; 2\log\Big(\frac{2\pi t}{\delta}\Big) \;+\; 4d\,\log\Big(d\,t\,b\,r\,\sqrt{\log\big(2\,d\,a/\delta\big)}\Big),$$

*and let $[x^*]_t$ be the closest point to $x^*$ in the discretization $D_t$ from Lemma I.8. Then with probability at least $1 - \delta$ (over all t),*

$$\big|f(x^*) - \mu_{t-1}\big([x^*]_t\big)\big| \;\leq\; \beta_t^{1/2}\,\sigma_{t-1}\big([x^*]_t\big) \;+\; \frac{1}{t^2}, \quad \forall t \geq 1.$$

*Sketch.* The derivative bound on the GP sample paths implies

$$\big|f(x) - f(x')\big| \;\leq\; b\,\sqrt{\log(2\,d\,a/\delta)}\,\|x - x'\|_1$$

with probability at least $1 - ae^{-(L/b)^2}$, for $L$ suitably large. Using the discretization property $\|x^* - [x^*]_t\|_1 \leq r\,d/\tau_t$ (from Lemma I.8), one obtains

$$\big|f(x^*) - f\big([x^*]_t\big)\big| \;\leq\; \frac{1}{t^2}$$

by choosing $\tau_t$ to shrink quickly enough in $t$. The remainder of the argument parallels Lemma I.2, showing that $\mu_{t-1}([x^*]_t)$ is close to $f([x^*]_t)$ by $\beta_t^{1/2}\,\sigma_{t-1}([x^*]_t)$. Combining these two pieces completes the proof. $\square$

**Lemma I.10** (Final Regret Bound in the Continuous Case). *Pick $\delta \in (0,1)$, and set*

$$\beta_t = 2 \log\left(\frac{4\pi t}{\delta}\right) + 4d \log\left(dt\,br\,\sqrt{\log(4\,d\,a/\delta)}\right),$$

*with $\sum_{t\geq 1} \tau_t^{-1} = 1$ and $\tau_t > 0$. Then the instantaneous regret at time $t$ satisfies*

$$r_t \leq 2\beta_t^{1/2}\sigma_{t-1}(x_t) + \frac{1}{t^2}, \quad \forall t \geq 1,$$

*with probability at least $1 - \delta$. Consequently,*

$$R_T = \sum_{t=1}^{T} r_t \leq \sum_{t=1}^{T}\left(2\beta_t^{1/2}\sigma_{t-1}(x_t)\right) + \sum_{t=1}^{T}\frac{1}{t^2} \leq 2\sqrt{C_1\,T\,\beta_T\,\gamma_T} + \frac{\pi^2}{6},$$

*with probability at least $1 - \delta$, establishing Theorem I.6.*

*Sketch.* By definition of the GP-UCB strategy,

$$x_t = \arg\max_{x \in D} \mu_{t-1}(x) + \beta_t^{1/2}\sigma_{t-1}(x).$$

Arguing as in Lemma I.3, one obtains

$$r_t = f(x^*) - f(x_t) \leq \left[f(x^*) - \mu_{t-1}(x^*)\right] + \left[\mu_{t-1}(x_t) - f(x_t)\right].$$

Using the grid argument (Lemma I.9) to control $|f(x^*) - \mu_{t-1}([x^*]_t)|$ and then bounding $\mu_{t-1}(x^*) - \mu_{t-1}(x_t)$ similarly gives

$$r_t \leq 2\beta_t^{1/2}\sigma_{t-1}(x_t) + \frac{1}{t^2}.$$

Summing over $t = 1$ to $T$ yields

$$R_T = \sum_{t=1}^{T} r_t \leq 2\sum_{t=1}^{T}\beta_t^{1/2}\sigma_{t-1}(x_t) + \sum_{t=1}^{T}\frac{1}{t^2}.$$

As $\sum_{t=1}^{\infty} 1/t^2 = \pi^2/6$, this second term is bounded independently of $T$. The first term is handled by the same technique as in the proof of Theorem **??** (Lemma I.5), giving the factor $\sqrt{C_1\,T\,\beta_T\,\gamma_T}$. Hence,

$$R_T \leq 2\sqrt{C_1\,T\,\beta_T\,\gamma_T} + \frac{\pi^2}{6}, \quad \text{with probability at least } 1 - \delta.$$

$\square$

# J  Sparse Gaussian Process: Detailed Variational Derivations

## J.1  Inducing Points and Joint Distribution

We introduce $m \ll n$ inducing points $\mathbf{Z} = \{\mathbf{z}_j\}_{j=1}^{m}$ and define the function values at these points as $\mathbf{u} = f(\mathbf{Z}) \in \mathbb{R}^m$. Under the GP prior:

$$p(\mathbf{f}, \mathbf{u}) = p(\mathbf{f} \mid \mathbf{u})\,p(\mathbf{u}),$$

where $\mathbf{f} = f(\mathbf{X}) \in \mathbb{R}^n$. For clarity:

$$p(\mathbf{u}) = \mathcal{N}\left(\mathbf{0}, \mathbf{K}_{\mathbf{z},\mathbf{z}}\right), \quad p(\mathbf{f} \mid \mathbf{u}) = \mathcal{N}\left(\mathbf{K}_{\mathbf{x},\mathbf{z}}\mathbf{K}_{\mathbf{z},\mathbf{z}}^{-1}\mathbf{u},\ \mathbf{K}_{\mathbf{x},\mathbf{x}} - \mathbf{K}_{\mathbf{x},\mathbf{z}}\mathbf{K}_{\mathbf{z},\mathbf{z}}^{-1}\mathbf{K}_{\mathbf{z},\mathbf{x}}\right).$$

## J.2  Variational Approximation for Sparse GP

We define a variational distribution

$$q(\mathbf{f}, \mathbf{u}) = p(\mathbf{f} \mid \mathbf{u})\,q(\mathbf{u}),$$

where $q(\mathbf{u})$ is free to be any Gaussian $\mathcal{N}\left(\boldsymbol{\mu}_u, \boldsymbol{\Sigma}_u\right)$. The posterior $p(\mathbf{f}, \mathbf{u} \mid \mathbf{X}, \mathbf{y})$ is approximated by $q(\mathbf{f}, \mathbf{u})$. We optimize the Evidence Lower BOund (ELBO):

$$\log p(\mathbf{y} \mid \mathbf{X}) \geq \mathbb{E}_{q(\mathbf{f},\mathbf{u})}\left[\log p(\mathbf{y} \mid \mathbf{f})\right] - \mathrm{KL}\left(q(\mathbf{f}, \mathbf{u}) \,\|\, p(\mathbf{f}, \mathbf{u})\right).$$

Substituting $q(\mathbf{f}, \mathbf{u}) = p(\mathbf{f} \mid \mathbf{u})q(\mathbf{u})$, we can rewrite the bound in terms of $q(\mathbf{u})$ alone, thus reducing the cost from $\mathcal{O}(n^3)$ to roughly $\mathcal{O}(nm^2)$ or $\mathcal{O}(m^3)$, depending on the particular scheme (e.g., FITC, VFE, etc.).

### J.3 Sparse Posterior for New Points

After learning $q(\mathbf{u}) = \mathcal{N}(\boldsymbol{\mu}_u, \boldsymbol{\Sigma}_u)$, the posterior predictive at a test location $\mathbf{x}_*$ is:

$$q(f(\mathbf{x}_*)) = \int p(f(\mathbf{x}_*) \mid \mathbf{u}) \, q(\mathbf{u}) \, d\mathbf{u},$$

where $p(f(\mathbf{x}_*) \mid \mathbf{u})$ can be computed from the conditional Gaussian rule. One obtains a Gaussian with mean

$$\tilde{\mu}(\mathbf{x}_*) = k(\mathbf{x}_*, \mathbf{Z}) \, \mathbf{K}_{\mathbf{z},\mathbf{z}}^{-1} \, \boldsymbol{\mu}_u$$

and variance

$$\tilde{\sigma}^2(\mathbf{x}_*) = k(\mathbf{x}_*, \mathbf{x}_*) - k(\mathbf{x}_*, \mathbf{Z}) \, \mathbf{K}_{\mathbf{z},\mathbf{z}}^{-1} \, k(\mathbf{Z}, \mathbf{x}_*) + \mathrm{tr}\!\left[\boldsymbol{\Sigma}_u \, \mathbf{K}_{\mathbf{z},\mathbf{z}}^{-1} \, k(\mathbf{x}_*, \mathbf{Z})^\top \, k(\mathbf{x}_*, \mathbf{Z}) \, \mathbf{K}_{\mathbf{z},\mathbf{z}}^{-1}\right]$$

(a typical formula in variational sparse GPs; some specifics differ depending on the chosen approximate method). These steps confirm that the predictive posterior remains a Gaussian with a distinct mean–variance form from exact GPs, yet still compatible with BO or RL-based exploration.

## K  Deep Gaussian Processes: Extended Proofs

### K.1  Two-Layer Deep GP Setup

We demonstrate a two-layer DGP as the simplest hierarchical example. Let $\mathbf{h}^{(1)}(\mathbf{x}) \in \mathbb{R}^{D_1}$ be the first latent layer, and $\mathbf{h}^{(2)}(\mathbf{h}^{(1)}(\mathbf{x})) \in \mathbb{R}^{D_2}$ be the second layer, eventually producing a scalar $f(\mathbf{x})$. For clarity, suppose $D_1 = D_2 = 1$, so each layer is one-dimensional. Then:

$$h^{(1)}(\mathbf{x}) \sim \mathcal{GP}(0, \, k^{(1)}(\mathbf{x}, \mathbf{x}')), \quad f(\mathbf{x}) = h^{(2)}(h^{(1)}(\mathbf{x})) \sim \mathcal{GP}(0, \, k^{(2)}(\mathbf{u}, \mathbf{u}')),$$

where $\mathbf{u} = h^{(1)}(\mathbf{x}), \mathbf{u}' = h^{(1)}(\mathbf{x}')$.

### K.2  Approximate Variational Inference

Since there is no closed-form expression for $p(\mathbf{h}^{(1)}, \mathbf{h}^{(2)} \mid \mathbf{X}, \mathbf{y})$, we introduce variational distributions at each layer with inducing points or random Fourier features. For instance, define:

$$q(\mathbf{h}^{(1)}, \mathbf{h}^{(2)}) = \int p(\mathbf{h}^{(1)} \mid \mathbf{u}_1) \, p(\mathbf{h}^{(2)} \mid \mathbf{u}_2, \mathbf{h}^{(1)}) \, q(\mathbf{u}_1, \mathbf{u}_2) \, d\mathbf{u}_1 d\mathbf{u}_2.$$

We then maximize a suitable ELBO with respect to $q(\mathbf{u}_1, \mathbf{u}_2)$. Once optimized, the posterior predictive for $f(\mathbf{x}_*) = h^{(2)}(h^{(1)}(\mathbf{x}_*))$ is approximated by integrating out the latent variables in each layer. The final result is typically a mixture or an integral of Gaussians, but many DGP frameworks simplify to produce an effectively Gaussian output with approximated mean $\tilde{\mu}(\mathbf{x}_*)$ and variance $\tilde{\sigma}^2(\mathbf{x}_*)$. Although more complex than single-layer GPs, the procedure can model multi-scale or nonstationary phenomena.

### K.3  Deep GP Posterior Probability Computations

Once we have the approximate distribution $q(f(\mathbf{x}_*))$ from a DGP, we can in principle compute or approximate:

**Variance.**  $\tilde{\sigma}^2(\mathbf{x}_*)$ emerges from the nested GP integrals. In practice, a sample-based approach can be used:

$$\tilde{\sigma}^2(\mathbf{x}_*) \approx \mathbb{E}_{q(\mathbf{u}_1,\ldots,\mathbf{u}_L)}\!\left[\mathrm{Var}(f(\mathbf{x}_*) \mid \mathbf{u}_1, \ldots, \mathbf{u}_L)\right] + \mathrm{Var}_{q(\mathbf{u}_1,\ldots,\mathbf{u}_L)}\!\left[\mathbb{E}(f(\mathbf{x}_*) \mid \mathbf{u}_1, \ldots, \mathbf{u}_L)\right].$$

**CDF and Probability of Duel.**  If the final output remains approximated by a Gaussian, we can directly apply the same reasoning as in the Classic GP. If the distribution is not a simple Gaussian, one may resort to numerical quadrature or sampling-based estimates of $P(f(\mathbf{x}) > f(\mathbf{x}'))$.

# L  Additional Details on Deep GP (REBMBO-D)

## L.1  Approximation Theory and Regret Bounds

In this section, we elaborate on the theoretical considerations for REBMBO-D using a more formal mathematical argument, avoiding bullet points and instead employing direct deductive reasoning [41]. While the use of deep kernels can provide substantial flexibility for modeling highly non-stationary or multi-scale objectives, it also introduces new challenges in establishing sub-linear regret guarantees that parallel those of simpler Gaussian Process (GP) kernels (e.g., RBF or Matérn).

**Formal Conditions on the Latent Mapping $\phi$.**  Let $\mathcal{X} \subset \mathbb{R}^d$ be the original input domain, and let $\phi \colon \mathcal{X} \to \mathbb{R}^D$ be the embedding function parameterized by $\Theta$. We first assume that $\phi$ is sufficiently smooth, in the sense that there exists a constant $L > 0$ such that

$$\|\phi(\mathbf{x}) - \phi(\mathbf{x}')\| \leq L \|\mathbf{x} - \mathbf{x}'\|$$

for all $\mathbf{x}, \mathbf{x}' \in \mathcal{X}$. In addition, we posit that $\phi$ is invertible (or approximately invertible) on the regions of interest, so there exists a function $\phi^{-1} \colon \phi(\mathcal{X}) \to \mathcal{X}$ such that $\phi^{-1}(\phi(\mathbf{x})) \approx \mathbf{x}$ for all $\mathbf{x}$ in the subdomain where data are collected. The invertibility condition prevents pathological distortions in the latent space, which ensures that distances in $\mathcal{X}$ are consistently reflected in $\phi(\mathcal{X})$. Under these conditions, if $f \colon \mathcal{X} \to \mathbb{R}$ is the black-box function of interest, then the composition $f \circ \phi^{-1}$ inherits properties similar to functions typically modeled by kernels in standard Bayesian Optimization.

**Implications for the GP Surrogate.**  Define a deep-kernel function $k_\Theta(\mathbf{x}, \mathbf{x}') = k(\phi(\mathbf{x}), \phi(\mathbf{x}'))$, where $k$ is a positive-definite function on $\mathbb{R}^D$. Because $\|\phi(\mathbf{x}) - \phi(\mathbf{x}')\| \leq L \|\mathbf{x} - \mathbf{x}'\|$, one can show that the kernel $k_\Theta$ remains Lipschitz in each argument up to a constant factor dependent on $L$. Classical results on GP-based Bayesian Optimization (BO) often rely on bounding the maximum information gain $\gamma_T$ after $T$ observations:

$$\gamma_T = \max_{\mathbf{x}_1, \ldots, \mathbf{x}_T \in \mathcal{X}} I\big(f(\mathbf{x}_1), \ldots, f(\mathbf{x}_T) \,\big|\, k_\Theta\big),$$

where $I(\cdot)$ denotes the mutual information. In typical kernel-based BO analyses, $\gamma_T = O(\log T)$ or other sublinear forms in $T$, provided $D$ is not too large or the kernel has controlled smoothness. Hence, under smoothness and boundedness assumptions on $\phi$, one can adapt standard covering-number arguments from reproducing kernel Hilbert spaces (RKHS) to show that $\gamma_T$ remains small or grows sublinearly in $T$.

**Deduction of Sublinear Regret.**  Let $\mathbf{x}^*$ be the global maximizer of $f$ in $\mathcal{X}$, and let $\mathbf{x}_1, \ldots, \mathbf{x}_T$ be the points sampled by REBMBO-D over $T$ iterations. We define the cumulative regret

$$R(T) = \sum_{t=1}^{T} \big[f(\mathbf{x}^*) - f(\mathbf{x}_t)\big].$$

Because REBMBO-D employs a Gaussian-process-like surrogate in the latent space $\phi(\mathcal{X})$, the usual BO proofs (for example, those from GP-UCB or GP-EI) can be transferred under suitable transformations. Specifically, if one can establish that the posterior variance $\sigma_t^2(\mathbf{x}_t)$ in the deep kernel scenario shrinks at a rate governed by $\gamma_T$, then one obtains an upper bound of the form

$$R(T) \leq C \sqrt{T \, \gamma_T}$$

for a constant $C$ that depends on hyperparameters of the kernel, the Lipschitz constant $L$, and the amplitude of noise. Since $\gamma_T$ grows at most sublinearly in $T$ under the aforementioned conditions on $\phi$ and $k_\Theta$, this implies sublinear growth in $R(T)$. Formally, if $\gamma_T = O(\log^p T)$ for some $p \geq 1$, then $R(T) = O(\sqrt{T \log^p T})$, which remains sublinear in $T$.

REBMBO-D inherits the potential for sub-linear regret from classical GP-based BO methods by casting the black-box function $f$ into a latent space via $\phi$, applying standard kernel-based BO proofs under smoothness and boundedness conditions, and leveraging EBM-driven exploration and PPO-based control to avoid premature convergence. More formally, given that $\|\phi(\mathbf{x}) - \phi(\mathbf{x}')\| \leq L \|\mathbf{x} - \mathbf{x}'\|$ and $\phi^{-1}$ exists in relevant regions, one can show that the maximum information gain $\gamma_T$ remains manageable, leading to regret bounds $R(T) = O(\sqrt{T \, \gamma_T})$. If $\gamma_T$ grows sublinearly in $T$, then $R(T)$ is itself sublinear. Therefore, under conditions of smooth embedding, stable EBM exploration, and bounded PPO updates, REBMBO-D can achieve the claimed sublinear regret growth.

# M   Detailed Implementation on Nanophotonic Structure

This appendix provides an expanded discussion of how REBMBO can be applied to the design of nanophotonic structures, using a nanosphere simulation as an illustrative example. Each iteration refines multiple components: a Gaussian Process (GP) for local uncertainty modeling, an Energy-Based Model (EBM) for global exploration, and a Proximal Policy Optimization (PPO) agent for adaptive multi-step decision-making. The detailed steps and associated formulas are presented below.

## M.1   Problem Setup and Parameterization

Consider a nanosphere configuration described by physical parameters such as layer thickness, doping level, refractive index, or particle radius. Collect these parameters into an input vector

$$\mathbf{x} = \big(x_1, x_2, \ldots, x_d\big) \in \mathcal{X} \subset \mathbb{R}^d,$$

where each component $x_i$ lies within a feasible range determined by physical constraints (for instance, layer thickness within $[0, 500]$nm, refractive index in $[1.2, 2.5]$, or doping concentration in $[0, 10^{20}\,\mathrm{cm}^{-3}]$). A forward simulator, such as a finite-difference time-domain (FDTD) solver or a rigorous coupled-wave analysis (RCWA) tool, evaluates the optical response of this nanosphere. Define

$$f(\mathbf{x}) = F\big(\mathbf{x}\big),$$

where $F$ represents the black-box nanosphere simulation that outputs a scalar performance metric (for example, transmittance, reflectance, or absorption). Each call to $F$ is typically expensive, motivating a data-efficient optimization approach.

## M.2   Initialization and Data Collection

**Initial Dataset.** At the start, gather $n_0$ initial samples,

$$\mathcal{D}_0 = \Big\{ \big(\mathbf{x}_i, \ f(\mathbf{x}_i)\big) \Big\}_{i=1}^{n_0},$$

by either drawing randomly from $\mathcal{X}$ or using a space-filling design (e.g., Latin hypercube sampling). This initial dataset seeds both the GP and the EBM with basic knowledge of the input–output relationship.

**Dynamic Dataloader.** In subsequent iterations, the framework proposes new points $\mathbf{x}_t$. If $\mathbf{x}_t$ was evaluated previously, retrieve the stored outcome to avoid redundant simulation. Otherwise, run the nanosphere simulator:

$$y_t = f(\mathbf{x}_t)$$

and append the pair $(\mathbf{x}_t, y_t)$ to the dataset, now denoted $\mathcal{D}_t$. This online loading procedure is essential for large-scale or expensive problems, as it triggers computationally heavy simulations only when the algorithm deems a candidate worthwhile.

## M.3   Gaussian Process Surrogate

After the initial dataset $\mathcal{D}_0$ is collected, train a GP to estimate the posterior mean $\mu^0(\mathbf{x})$ and variance $\sigma^{2,0}(\mathbf{x})$. At iteration $t$, the GP is updated with the most recent dataset $\mathcal{D}_{t-1}$, yielding

$$f(\mathbf{x}) \,\big|\, \mathcal{D}_{t-1} \ \sim \ \mathcal{N}\big(\mu_t(\mathbf{x}), \sigma_t^2(\mathbf{x})\big).$$

The explicit forms for $\mu_t(\mathbf{x})$ and $\sigma_t^2(\mathbf{x})$ follow from standard GP regression:

$$\mu_t(\mathbf{x}) = m(\mathbf{x}) \ + \ k\big(\mathbf{x}, \mathbf{X}_{t-1}\big)\big(K + \sigma_n^2 I\big)^{-1}\big(\mathbf{y}_{t-1} - m(\mathbf{X}_{t-1})\big),$$

$$\sigma_t^2(\mathbf{x}) = k\big(\mathbf{x}, \mathbf{x}\big) \ - \ k\big(\mathbf{x}, \mathbf{X}_{t-1}\big)\big(K + \sigma_n^2 I\big)^{-1}k\big(\mathbf{X}_{t-1}, \mathbf{x}\big),$$

where $\mathbf{X}_{t-1} = [\mathbf{x}_1, \ldots, \mathbf{x}_{n_{t-1}}]$, $\mathbf{y}_{t-1} = [f(\mathbf{x}_1), \ldots, f(\mathbf{x}_{n_{t-1}})]^T$, $K$ is the kernel matrix ($K_{ij} = k(\mathbf{x}_i, \mathbf{x}_j)$), and $\sigma_n^2$ is a noise variance. Common kernel choices (e.g., RBF or Matérn) may be combined or extended based on domain expertise in nanophotonics. The GP posterior helps identify regions of high predicted performance or substantial uncertainty, both relevant for exploration.

## M.4 Energy-Based Model (EBM) Training

In parallel with GP updates, the EBM parameters $\theta$ are retrained or partially trained at each iteration to ensure global coverage of the search space. Formally, the EBM density is given by

$$p_\theta(\mathbf{x}) = \frac{\exp(-E_\theta(\mathbf{x}))}{Z_\theta}, \quad Z_\theta = \int \exp(-E_\theta(\mathbf{x}')) \, d\mathbf{x}'.$$

Since $Z_\theta$ is typically intractable, the algorithm employs approximate methods such as short-run MCMC or contrastive divergence. In short-run MCMC, one might initialize a set of particles $\{\mathbf{x}'_j\}$ near the dataset $\mathcal{D}_{t-1}$, then evolve them briefly under gradient-based moves

$$\mathbf{x}'_j \leftarrow \mathbf{x}'_j - \alpha_{\text{MCMC}} \nabla_{\mathbf{x}} E_\theta(\mathbf{x}'_j) + \sqrt{2\,\alpha_{\text{MCMC}}}\, \mathbf{z}_j,$$

where $\mathbf{z}_j \sim \mathcal{N}(0, I)$. A few such iterations suffice to update $\theta$ based on the mismatch between synthesized samples and real data. The resulting energy function $E_\theta(\mathbf{x})$ tends to give lower values (higher probabilities) to regions that have shown promise or remain unexplored. This global signal complements the GP's local insights.

## M.5 Reinforcement Learning via PPO

Let $\mathbf{s}_t$ be the RL state that aggregates information from the GP and EBM, such as $\mu_t(\mathbf{x}), \sigma_t(\mathbf{x}), E_\theta(\mathbf{x})$, or other relevant features (e.g., iteration count or GP hyperparameters). A policy $\pi_\phi$ then chooses a new configuration $\mathbf{x}_t = \pi_\phi(\mathbf{s}_t)$. Proximal Policy Optimization (PPO) [38] updates $\phi$ by constraining large changes in the probability ratio

$$r_t(\phi) = \frac{\pi_\phi(\mathbf{a}_t \mid \mathbf{s}_t)}{\pi_{\phi_{\text{old}}}(\mathbf{a}_t \mid \mathbf{s}_t)},$$

thereby promoting stable learning. In the nanophotonic setting, $\mathbf{a}_t \equiv \mathbf{x}_t$. After simulating $y_t = f(\mathbf{x}_t)$, define a reward function that balances direct performance and EBM-driven exploration. A common choice is

$$r_t = R(f(\mathbf{x}_t)) + \gamma_{\text{EBM}} \left[-E_\theta(\mathbf{x}_t)\right],$$

where $R$ penalizes lower performance (e.g., by taking the negative of a target error or the negative of $-f$) and $\gamma_{\text{EBM}}$ scales the global exploration term. The PPO objective,

$$L^{\text{PPO}}(\phi) = \mathbb{E}_t \left[\min\left(r_t(\phi)\, A_t, \, \text{clip}(r_t(\phi),\, 1-\epsilon,\, 1+\epsilon)\, A_t\right)\right],$$

where $A_t$ is the advantage function, is then maximized to refine $\pi_\phi$. This process endows the sampling policy with the capacity to move beyond local maxima and systematically explore new regions of the parameter space.

## M.6 Iterative Algorithm and Convergence

At each iteration $t$, the GP and EBM are updated to reflect the newly acquired observations $\{(\mathbf{x}_t, y_t)\}$. The GP posterior $\mu_t, \sigma_t$ captures refined local estimates of $f(\mathbf{x})$, while the EBM modifies $E_\theta(\mathbf{x})$ for broader coverage. The RL agent receives an updated state $\mathbf{s}_{t+1}$ and modifies its policy $\pi_\phi$ according to the reward signals. The next point $\mathbf{x}_{t+1}$ emerges from this policy, balancing local exploitation with global exploration. Over multiple iterations, the collected samples cluster near high-performance configurations, effectively converging on near-optimal nanosphere designs with fewer evaluations than naive or single-step methods.

**Practical Considerations.** The nanosphere simulator must be callable multiple times under varying parameter inputs, and the MCMC-based EBM training typically requires choosing a small number of gradient steps per iteration. PPO's hyperparameters (e.g., learning rate, minibatch size, and clipping threshold $\epsilon$) can be selected via standard tuning heuristics [66]. When applied to more general nanophotonic designs, the same approach applies as long as the underlying simulator remains differentiable or partially differentiable if gradient-based EBM updates are desired, although purely sampling-based EBM training can also succeed. Finally, switching the GP variant (classic, sparse, or deep) depends on data scale and computational feasibility, making the method adaptable to a range of problem sizes.

