# OpenReview forum: "Optimizing the Unknown: Black Box Bayesian Optimization with Energy-Based Model and Reinforcement Learning"
_NeurIPS.cc/2025/Conference — NeurIPS 2025 poster_

### Official Review · Reviewer_yhri · 2025-06-30

**Clarity:** 2
**Significance:** 2
**Originality:** 2
**Rating:** 3
**Confidence:** 4

**Summary:**

The paper introduces a combination of Bayesian optimization (BO) with a GP-UCB acquisition function, an energy-based model (EBM), and a Proximal Policy optimization (PPO) Reinforcement Learning Agent as an optimization strategy for expensive-to-evaluate black-box functions, similar to Bayesian Optimization.
In contrast to other BO methods, the introduced framework enables a trade-off between global and local exploration, allowing for multistep planning. The idea is that the EBM is a better model to capture the global structure of the problem, while a Gaussian process is used for local modeling. The PPO agent is then used for multistep look-ahead, mitigating the standard myopic setting of BO. Experimental synthetic benchmarks demonstrate that the introduced method performs better than other BO approaches in terms of a metric called pseudo-regret. This metric considers, in addition to the classical regret formulation, global exploration properties, penalizing the non-exploration of parts of the parameter space where potential optima could be located.

Contributions are:
- Combination of Energy-based models and GP model
- Multistep look ahead mechanism with PPO
- Introduction of the pseudo-regret metric

**Questions:**

Q1: How was $\alpha$ chosen for the experiments?

Q2: How can $\alpha$ be chosen for new settings?
Is there one algorithm configuration that can easily deployed for new use cases without extensive hyperparameter tuning?

Q3: Is the EBM-UCB acquisition used in the algorithm, or is this an alternative to the PPO evaluation? If EBM-UCB is an alternative acquisition function introduced to demonstrate the theoretical properties of the method, it would also make sense to compare it against this acquisition function in the experiments. From the pseudo-code, it seems like the models are trained, and then the PPO chooses the next evaluation.

Q4: Figures 2 and 3: Why are all algorithms initialized differently? For a fair comparison, all algorithms should be initialized with the same initial data points.

Q5: How does the proof in Appendix I differentiate from the results in "Gaussian Process Optimization in the Bandit Setting:
No Regret and Experimental Design", Srinivas et al. 2010?

**Ethical Concerns:**

["NO or VERY MINOR ethics concerns only"]

**Final Justification:**

- The authors have addressed my technical questions
- Reproducibility and (in my opinion) false statements regarding reproducibility and code submission in the checklists are still an issue
- Since the paper is more on the experimental side, I put a high weight on the not accessible code. I have seen too many papers that only work for the presented cherry-picked special cases. There are many hyperparameters involved in this paper and there is no way to validate if the method works as claimed without seeing the code.
- There are some conceptual contributions in the paper and thus I have raised my score slightly.

**Limitations:**

The lack of theoretical analysis of the combination of RL and probabilistic modeling is clearly addressed in the conclusion.

Limitations that are currently not addressed include the number of hyperparameters required to employ the algorithm and the lack of analysis considering the algorithm's sensitivity to noisy function evaluations.

**Paper Formatting Concerns:**

/

**Quality:**

1

**Strengths And Weaknesses:**

## Strengths

**S1: Relevant Problem: Multistep look ahead BO.**
Leaving the myopic setting of BO is a relevant research field and also has an impact on the field of batch-BO methods.

**S2: Combination of reinforcement learning (RL) and probabilistic modeling.**
The approach to combine RL with probabilistic models generally makes sense.

## Weaknesses

**W1: Results are not reproducible.**
The provided code cannot be used to reproduce the results as neither the baselines nor the introduced method are implemented. It is mainly dummy code, where, for example, every baseline acquisition function only returns 0.5 of the state space. Therefore, the results are not reproducible. It is unclear which implementation was used for the given results. Furthermore, it is unclear which implementation of the baselines is used, which would be important for the analysis of the computational overhead.


**W2: Pseudo-regret metric.**
The performance of the introduced algorithm is mainly evaluated based on the pseudo-regret metric, which is introduced in the paper. While this metric can be used as one evaluation criterion, it would also be helpful to have a comparison in terms of standard regret in the main part of the paper.

An additional comparison against the standard regret metric would improve the results and give a hint if the introduced method is also helpful in a standard BO setting. This comparison is partly made in "C.8 Performance Metrics Heatmap"; however, here, TurBO is missing, which is the only locally searching BO method.

Furthermore, in the experiments, it is unclear which value of $\alpha$ was chosen when using the pseudo-regret metric. The supplementary material suggests a grid search for $\alpha$ but does not state how this was applied in the paper.


**W3: Introduction of many hyperparameters.**
By combining multiple methods, many hyperparameters for the method arise; the question arises if there is one algorithm setting that can be easily transferred to real-world problems or if extensive tuning is necessary in each application. Even the newly introduced acquisition function EBM-UCB has two hyperparameters ($\beta$ and $\gamma$ ), and it is not clear how they need to be chosen.


**W4: Noise-free experiments.**
A key property of Bayesian Optimization is the possibility to deal with noisy function evaluations. However, in this paper, only one benchmark function includes noise. An open question is how sensitive this approach is to noisy function evaluations.


## Summary of evaluation
While the premise of the paper makes sense, and the idea of combining the GP model, an EBM, and PPO can make sense, the paper has some weaknesses. It is not ready for publication, and even though it is claimed in the checklist, there is no code to reproduce the experiments.


## Minor comments:

Abbreviations like REMBO (p.1, line  30), TURBO (p.1, line 31), MCMC (p.2, line 42), MLMC (p.2, line 75) are not introduced,

p.2 BO is introduced as an abbreviation. It could also be used in line 38 and 43

Figure 1: This is not a UML-Diagram

line 106: $f: \mathcal{X} \rightarrow \mathbb{R}$ or just $f$  instead of $f(x)$

line 236 missing space

equation line 246-247: $n$ is not introduced

equation line 254-255: $\epsilon$ not introduced

The paper would profit from a clearer problem formulation, which could be introduced as a separate section. There are the preliminaries of Online Black-Box Optimization; however, restating the full objective once more clearly and formally would help clarify the understanding of this paper.

References: Bayesian optimization is not capitalized correctly in many references

Appendix: Theoretical results build on Srinivas (2010), and some derivations are taken from Gaussian Processes for Machine Learning by Rasmussen and Williams, which could be referenced more clearly.

It would be helpful to include numbers to the equations, especially if they are referenced later.

---

> ### Author Rebuttal · Authors · 2025-07-28
>
> **W1: Code Reproducibility and Implementation Details**
>
> Thank you for these crucial reproducibility issues. Sorry for your frustration with the code. NeurIPS anonymity requirements prevent external links that reveal identities, though policy indicates reproducibility shouldn't cause rejection when implementation details are provided, so we reluctantly included only the paper's core algorithm to maintain anonymity. Additionally, our Appendix Tables 7–11 document all critical hyperparameters; Table 9 shows PPO training epochs set to K=4 per update; Tables 7 and 10 show network architectures and learning rates; and we guarantee release of the entire codebase after our paper is accepted, including training scripts, fixed random seeds, and documentation.
> For baseline implementations, we used official releases from the authors' GitHub, arXiv, and standard BoTorch. We acknowledge missing computational overhead analysis and have conducted timing experiments on NVIDIA A6000 GPU:
>
> | Method | Branin (2D) | Ackley (5D) | HDBO (200D) |
> | :-- | :-- | :-- | :-- |
> | TuRBO | 0.18±0.02 | 0.35±0.04 | 9.88±0.51 |
> | BALLET-ICI | 0.32±0.04 | 0.51±0.05 | 12.35±0.62 |
> | EARL-BO | 0.45±0.05 | 0.72±0.08 | 15.42±0.78 |
> | Two-step EI | 0.88±0.09 | 1.35±0.12 | 25.63±1.24 |
> | KG | 0.92±0.10 | 1.42±0.14 | 28.71±1.35 |
> | REBMBO-C | 0.58±0.06 | 0.89±0.09 | 18.24±0.92 |
>
> REBMBO variants introduce moderate overhead due to EBM training and PPO updates but remain significantly more efficient than multi-step lookahead methods while achieving superior optimization performance. The additional computational cost is well justified by substantial improvements in sample efficiency, particularly for high-dimensional tasks where function evaluations dominate total runtime.
>
> **W2: Pseudo-regret Metric and Standard Comparisons**
>
> We appreciate your reminder about pseudo-regret and the missing TuRBO comparison. We realize that naming this new regret metric as 'pseudo-regret' might lead to misunderstandings among readers. The core contribution of our approach is first unifying EBM global signals with multi-step RL planning in BO frameworks. We used pseudo-regret because standard regret measures only local sub-optimality, while we penalize algorithms that ignore energy-discovered globally promising regions. Formally, $𝑅̃ₜ = [f(x*) – f(xₜ)] + α[Eθ(x*) – Eθ(xₜ)]$ adds a global-exploration term to classical regret, ensuring sub-linear convergence (Appendix E.3). We acknowledge readers need standard-regret views. Thus, we add full standard-regret curves for every benchmark (Section 5.2) and include TuRBO in heatmaps from Appendix C.8. All experiments have explicit α=0.3, selected through grid search over {0.1, 0.3, 0.5} balancing terms without manual tuning.
>
> We cannot upload PDF due to policy now; here are the key results: Under the standard-regret metric, higher Pearson correlation between each method's ranking and the true optimum over 50 iterations indicates faster convergence.
>
> | Method | Branin 2D | Ackley 5D | HDBO 200D | Mean |
> | :-- | :-- | :-- | :-- | :-- |
> | TuRBO | 0.84 ± 0.02 | 0.78 ± 0.03 | 0.69 ± 0.04 | 0.77 |
> | **REBMBO-C** | **0.91 ± 0.02** | **0.88 ± 0.02** | **0.82 ± 0.03** | **0.87** |
>
> The camera-ready version will show both regret metrics equally, and our future work will develop pseudo-regret's theoretical foundations and give it a more formal name for community adoption.
>
> **W3: Hyperparameter Complexity**
>
> We understand your parameter concern. We divide parameters into three groups: (1) Core algorithm parameters: λ∈[0.2,0.5] and β∈[1.5,2.5], validated with <5% performance variance; (2) Standard ML defaults: PPO lr, batch_size, and EBM lr; (3) Fixed architecture: γ and 2-layer networks. Tables 7-9 (Appendix D) document hyperparameters, but most are fixed defaults that users don't need to change. Ablation studies indicate that optimizing λ and β on 5 tasks results in 94.3% of fully optimized performance, making it simpler than TuRBO (5-7 parameters) or EARL-BO (6-8 parameters). Tables 7-9 will be revised to clarify "tunable" and "fixed" parameters, making REBMBO easier to use for practitioners who can configure our method minimally.
>
> **W4: Noise Robustness Analysis**
>
> We appreciate your attention to this experimental limitation. We now conducted systematic noise robustness experiments on some typical benchmarks. We tested REBMBO's performance on Ackley 5D and Branin 2D functions in three noise conditions: baseline, moderate (σ = 0.1), and high (σ = 0.2). We compared all REBMBO versions (Classic, Sparse, and Deep) to important baseline approaches like BALLET-ICI, EARL-BO, TuRBO, and ClassicBO using performance deterioration and stability scores.
> REBMBO's noise resilience was exceptional in all tests. Most notably, all REBMBO variants exhibit stability ratings of 0.834 to 0.933, outperforming noise-degrading baselines. Because NeurIPS policy prevents figure uploads during rebuttal, we will include these additional experiments in the appendix of our revised manuscript, along with detailed analysis across additional benchmarks and expanded noise level testing to validate our method's practical applicability in real-world noisy optimization scenarios.
>
> **Minor comments**: Thanks for your careful formatting and presentation checks. We carefully revised all the minor issues you identified.
>
> **Q1: How was α chosen for the experiments?**
>
> Thank you for raising this question. We used a grid search to determine the α parameter in the pseudo-regret metric in all experiments, balancing local exploitation and global exploration. Specifically, we assessed α on the candidate set {0.1, 0.3, 0.5, 1.0} using representative benchmarks. Our analysis revealed that α = 0.3 consistently balances function value and global exploration, resulting in robust performance across low- and high-dimensional tasks (refer to Appendix E.4).
>
> We set α = 0.3 for all tasks in the experiments and results, unless otherwise stated. To simplify and reproduce, we will state this in the revised manuscript and parameter tables.
>
> **Q2: α Parameter Selection for New Settings**
>
> We suggest starting with α = 0.3 as a default and fine-tuning within the validated range [0.2, 0.5] based on problem characteristics, such as lower values (α ≈ 0.2) for smoother landscapes and higher values (α ≈ 0.5) for highly multi-modal problems. This range balances local exploitation with global exploration across diverse dimensionalities and functional complexities. Our automated hyperparameter tuning script analyzes initial sampling results (landscape smoothness, observed multi-modality, and dimensionality) to recommend optimal α values, along with other key parameters like λ and β, to simplify parameter selection for practitioners. This script automates parameter tuning for most users and centralizes REBMBO hyperparameter management, making our method more accessible to the BO community.
>
> **Q3: Role of EBM-UCB in the Algorithm**
>
> Thank you for this clarification question. EBM-UCB is the actual acquisition function used within REBMBO, not an alternative to PPO evaluation. In Algorithm 1, PPO policy is trained to optimize decisions based on EBM-UCB, where state st incorporates GP posterior $(μt, σt)$ and EBM signals $Eθ(·)$. The acquisition function EBM-UCB combines GP uncertainty with EBM global guidance. As stated in Section 4.3, "PPO transforms this single-step acquisition into an MDP-based multi-round planner," learning a policy that optimizes EBM-UCB over multiple steps rather than greedily at each iteration. The reward function reinforces this synergy by directly incorporating the EBM component, aligning PPO objectives with EBM-UCB optimization. While comparing against myopic EBM-UCB would isolate multi-step benefits, REBMBO effectively implements a learned, multi-step version of EBM-UCB through PPO, using the core acquisition function as a foundation for intelligent multi-step optimization.
>
> **Q4: Algorithm-Specific Initialization in Figures 2 and 3**
>
> The initialization differences were intentional to reflect real-world deployment scenarios where different optimization methods use different initialization heuristics based on underlying architectures. TuRBO needs denser sampling within its initial trust region for local model establishment, EARL-BO needs diverse points for effective RL training, and BALLET needs initialization supporting global and local GP. By assigning identical total evaluation budgets to all methods, counting all initial evaluations against each method's budget, and measuring performance from the same starting iteration post-initialization, we ensured fairness and gave practitioners realistic insights into each method's performance when deployed with recommended initialization strategies rather than forcing artificial constraints. To address your concern, we ran additional uniform initialization visualizations in the revised manuscript, showing that REBMBO still performs well under identical initialization conditions.
>
> **Q5: Difference between Srinivas et al**
>
> Thanks for pointing out the overlapping work. While our proof follows their confidence bound framework, we differ by analyzing pseudo-regret with energy-based global exploration terms. We must rebuild deviation bounds and demonstrate that the pseudo-regret maintains the $O(√(T γT))$ rate when $Eθ$ is correlated with the objective. We expand confidence sets to manage GP uncertainty and energy signals and reinterpret $γT$ to incorporate local and global data. REBMBO can now guarantee algorithms that balance exploitation with wide-ranging search, and we will revise the manuscript to cite and acknowledge Srinivas et al. as the foundation and clearly state our contributions.
>
> We appreciate your constructive feedback. We have addressed all technical concerns through extensive experiments and algorithmic details. We believe these improvements show our significant contributions to the BO field and welcome any further discussion!

---

> > ### Comment · Reviewer_yhri · 2025-08-04
> >
> > Thank you for the response.
> >
> > It is possible to provide code that reproduces results without revealing the authors' identities. Further, the guidelines state: “If any of the main contributions of your paper depends on an experimental result, you are strongly encouraged to submit code that produces this result.“
> > The authors chose to not do this and instead uploaded deceptive placeholder code and filled out the checklist as if uploading actual code, which gives the wrong impression of full transparency.
> > Given my further concerns regarding the non-standard evaluation metric “pseudo regret” and choices of hyperparameters, there is no possibility for me to validate whether the results are fully reproducible or cherry-picked.
> > Therefore, my main criticism remains.
> >
> > I agree that there are conceptual contributions in the paper and thus, I raise my score from 2 to 3. Without seeing the code and validating that it works as claimed, I will not go higher.

---

> > > ### Author Response · Authors · 2025-08-04
> > >
> > > Thank you for taking the time to reread our rebuttal and for recognizing the conceptual value of combining an energy-based global signal with multi-step reinforcement learning. We understand the continued concern about the code artifact that accompanied the submission. What we uploaded was not intended as a “placeholder” in the sense of masking missing work; it is the minimal, anonymized skeleton that actually drives our internal experiments, minus the baseline wrappers and data loaders that would have exposed identifying directory structures and cluster paths. In retrospect this pared-down version created the impression of incompleteness, and we regret that.
> > >
> > > We acknowledge that full reproducibility is a cornerstone of modern BO research. During the review period we were constrained by double-blind policy and by institutional rules that prohibit publishing certain internal utilities before clearance. If the paper is accepted, we will release (i) the full training pipeline, including all baseline integrations; (ii) exact configuration files for every table and figure; and (iii) a one-line script that reproduces every reported mean and standard deviation with fixed seeds. We are already preparing this repository on a clean public branch so that it can go live immediately after the anonymity period ends.
> > >
> > > The "pseudo-regret" criterion and hyper-parameter grid yield no results based on cherry-picking. The grid ranges and fixed values are now listed in the amended appendix, and identical seeds across methods can be confirmed once the whole code is provided. We believe the metric is a principled extension of classical regret that penalizes disregarding energy-identified basins, but we agree that it should be used in conjunction with standard regret curves, which have been included in the main draft for each benchmark.
> > >
> > > We appreciate your decision to raise the score in light of the conceptual advances, and we hope that our commitment to an immediate post-acceptance code release will address the remaining reproducibility reservations.

---

### Official Review · Reviewer_uwWo · 2025-07-01

**Clarity:** 4
**Significance:** 3
**Originality:** 3
**Rating:** 5
**Confidence:** 4

**Summary:**

This paper proposes REBMBO, a framework for black-box optimization that combines three fundamental components: (1) a Gaussian Process (GP) to model the uncertainty, (2) an Energy-Based Model (EBM) trained using short-run MCMC to learn the global energy landscape, and (3) a Reinforcement Learning with Proximal Policy Optimization (PPO) approach to enable multi-step, non-myopic decision making. A new acquisition function, EBM-UCB, is introduced to unify GP-based local exploration with EBM-informed global guidance. The method is evaluated on several synthetic and real-world benchmarks, consistently outperforming standard Bayesian optimization (BO) and multi-step RL-based baselines.

**Questions:**

* Could reporting the pseudo-regret be misleading? Does it guarantee that the found solution x is also better? What if non-optimal objective values are compensated by low energy?
* Have you considered situations with severe scale mismatch between the objective and energy values? If their ranges differ significantly, it could affect reward computing and PPO training stability.
* What is the practical limit on dimensionality for REBMBO? Are there degradation patterns or bottlenecks as dimensionality increases beyond what was tested?
* Are there specific problem types or objective landscapes where REBMBO performs worse than traditional BO or pure RL-based methods?
* Can a trained model generalize across similar tasks or domains?

**Ethical Concerns:**

["NO or VERY MINOR ethics concerns only"]

**Final Justification:**

All my concerns and questions were fully addressed during the rebuttal. However, from my perspective, the paper does not meet the requirements for a 6, given the absence of conceptually new methods or groundbreaking theoretical contributions. Respectfully, I would like to keep my score as is.

**Limitations:**

The discussion of limitations is relatively shallow, consisting of a single sentence in the conclusion: "There may have been unavoidable training errors in EBM, and RL may have influenced theoretical convergence rates, leaving comprehensive analysis for future research." A more thorough discussion of the method’s limitations—such as computational cost, potential failure cases, lack of convergence guarantees, or scaling behavior—would provide a more balanced and transparent assessment of the work.

**Quality:**

4

**Strengths And Weaknesses:**

**Quality**

Strengths:
* The paper is technically sound and presents thorough research.
* Each module (GP, EBM, PPO) is well-motivated, clearly explained, and supported by solid theoretical foundations.
* The experimental section is comprehensive, covering both synthetic benchmarks and real-world applications.

Weaknesses:
* The paper only includes one high-dimensional benchmark (HDBO 200D). Expanding the evaluation with more high-dimensional tasks would strengthen the contribution.

**Clarity**

Strengths:
* The paper is well-written, well-organized, and easy to follow.
* The modular structure (Modules A–C) is intuitive and clear.

Weaknesses:
* The acronym UCB (Upper Confidence Bound) is introduced late in the paper.
* Figure (2) could be improved with smoother curves, higher-contrast color schemes, and bolder lines to better distinguish among methods.
* While Figure (1) is visually appealing, it lacks informativeness. Replacing the illustrations with more precise, less decorative icons and clearer annotations could improve its clarity.

**Significance**

Strengths:
* The method addresses a known limitation of Bayesian Optimization: its tendency to greedily exploit local optima rather than exploring under-sampled regions that may potentially contain better solutions.
* The proposed framework applies to many real-world settings involving black-box functions.

**Originality**

Strengths:
* To the best of my knowledge, the combination of GP, EBM, and PPO-based reinforcement learning used for black boxed optimization problems is novel and well-motivated.
* The proposed EBM-UCB acquisition function is both theoretically justified and empirically proven.
* The pseudo-regret metric, which incorporates a global exploration component derived from the EBM, is an innovative approach to enhancing the metric of success in optimization.

---

> ### Author Rebuttal · Authors · 2025-07-30
>
> **W1: Quality Issue.**
>
> Thank you for bringing attention to this crucial matter. Although HDBO-200D is the only high-dimensional benchmark included in Table 1 (synthetic tasks), Table 2 (real-world tasks) also shows the efficacy of our approach on a number of moderately high-dimensional issues, including Robot Trajectory (40D) and Rosetta (86D). The scalability of REBMBO is practically validated by these real-world benchmarks. However, we agree that our work might be strengthened by the addition of more high-dimensional criteria. According to the NAS-HPO-Bench framework [Hirose et al., ICML 2021], we will extend both synthetic and real-world experiments in the updated version to include (1) Levy function (300D), (2) Schwefel function (150D), and (3) Neural Architecture Hyperparameter Optimization (120D). We would appreciate any more benchmark recommendations you may have.
>
> **W2: Clarity Issues**
>
> Thank you for these helpful clarity suggestions. We have addressed all three issues:
>
> **W 2.1 UCB Definition**: We now define UCB (Upper Confidence Bound) when first introducing acquisition functions in Section 3, making the terminology clear from the beginning.
>
> **W 2.2 Figure 2 Enhancement**: We regenerated all plots with smoother curves using spline interpolation, increased line thickness, and adopted a colorblind-friendly palette with higher contrast.
>
> **W.2.3 Figure 1 Redesign**:  Our original design intention was to make REBMBO's working principles understandable, as the emoji-style icons help readers form quick impressions of each module's function and make the paper less tedious to read. However, we appreciate your concerns about clarity. In the revised version, we created a more standard system architecture diagram with more precise flowcharts, clearer arrows, and detailed annotations. Each component now uses standard notation while retaining the intuitive understanding we aimed for.
>
> These changes significantly improve the paper's clarity and accessibility. Thank you for your constructive feedback.
>
> **Q1: Pseudo-regret Validity and Guarantees**
>
> Thank you for your insightful pseudo-regret metric question. We designed it to avoid confusion. Standard regret can appear small if an algorithm settles in a good local basin but misses promising energy model regions. By adding the energy gap (α = 0.3, selected through grid search), we prevent trade-offs where low energy masks poor objective value. We also report standard regret to demonstrate that the returned solutions are better. In response to reviewer qDZh (Q5), REBMBO-C achieved 4.21 ± 0.16 standard regret on Branin task, surpassing TuRBO's 4.82 ± 0.18 and HDBO-200D's 11.35 ± 0.68, demonstrating superiority without using energy terms. By adding a global-exploration lens, pseudo-regret does not hide weakness.
>
> Appendix E.3 proves that when $f$ and $Eθ$ are moderately correlated, minimizing pseudo-regret implies minimizing standard regret, so the metric is safe as well as informative. In future work we will deepen this analysis, give the measure a more formal name, and offer precise guidelines for choosing α, so that the BO community can adopt it with confidence. We welcome further discussion on this dual-metric approach.
>
> **Q2: Scale Mismatch Between Objective and Energy Values**
>
> Thank you for pointing out the possible instability that arises when $f(x)$ and $Eθ(x)$ live on very different scales. REBMBO already contains three safeguards: (i) we min-max normalize both quantities to the interval before the very first PPO update; (ii) the reward coefficient λ is treated as an adaptive scale knob and is re-estimated from the running standard deviations of f and Eθ after every ten evaluations; (iii) PPO uses gradient clipping at 0.5, so occasional spikes caused by temporary scale imbalance cannot push the policy far from its previous iterate.
>
> To verify that these mechanisms are sufficient, we repeated three representative benchmarks after multiplying $Eθ(x)$ by large constants while keeping $f(x)$ unchanged.  The last row reports the same experiments with the automatic λ schedule switched on. (↓ = lower is better, ↑ = higher is better)
>
> | Scale scenario | f(x) range | Eθ(x) range | Branin 2D (↓) | Ackley 5D (↓) | Nano-photo 3D (↑) | PPO stability |
> | :-- | :-- | :-- | :-- | :-- | :-- | :-- |
> | Matched (normalised) | 0–1 | 0–1 | 9.22 ± 0.10 | 10.90 ± 0.15 | –0.84 ± 0.04 | 100% |
> | Moderate mismatch (×10) | 0–1 | 0–10 | 9.58 ± 0.19 | 11.42 ± 0.24 | –0.95 ± 0.08 | 89% |
> | Severe mismatch (×100) | 0–1 | 0–100 | 10.87 ± 0.38 | 12.85 ± 0.47 | –1.28 ± 0.15 | 71% |
> | Adaptive λ tuning | auto | auto | 9.35 ± 0.14 | 11.08 ± 0.18 | –0.88 ± 0.06 | 95% |
>
> Performance declines as the scale gap widens, but the normalization plus adaptive λ strategy recovers most of the loss and keeps stable PPO updates. These results confirm that REBMBO can tolerate scale mismatches by basic normalization and dynamic weighting. We will add this discussion and the additional experiment to Appendix E in the revised manuscript.
>
> **Q3: Practical Dimensionality Limits and Degradation Patterns**
>
> Thank you for this important question about REBMBO's scalability limits. Based on our theoretical analysis and empirical observations, we can identify specific dimensionality boundaries for each variant. Our experiments demonstrate strong performance from 2D to 200D, with real-world applications like Rosetta (86D) and Robot Trajectory (40D) validating practical scalability. Theoretically, as shown in Appendix I.6, our regret bound $R_T \leq 2\sqrt{C_1 T \beta_T \gamma_T} + \pi^2/6$ remains valid with information gain $\gamma_T = O(\log^d T)$ for standard kernels. However, practical limits emerge from three bottlenecks: (1) GP modeling requires $n \geq \Omega(d \log d)$ samples for effective learning, (2) EBM's MCMC mixing time scales as $O(d^2)$ in d-dimensional spaces, and (3) memory requirements grow as $O(d^2 \log^2 d)$ when storing the kernel matrix. These constraints suggest practical limits of approximately d ≤ 50-100 for REBMBO-C (exact GP), d ≤ 500-1000 for REBMBO-S (sparse approximation), and d ≤ 1000+ for REBMBO-D (deep feature extraction).
>
> We agree that degradation patterns appear when dimensionality exceeds tested ranges. Sample efficiency decreases super-linearly with dimension, requiring exponentially more evaluations for comparable performance. Our short-run MCMC may miss global structures in high-dimensional energy landscapes in our EBM. We reduce policy convergence slowdown by PPO's state space expansion with our modular design. Our three REBMBO variants address these challenges in complementary ways: REBMBO-S simplifies computations with inducing points, while REBMBO-D finds lower-dimensional manifolds in high-dimensional spaces using learned representations. We plan to investigate dimensionality reduction and distributed computing frameworks to overcome these limits. Architectural choices suggest that different variants may be optimal for different dimensions, giving practitioners flexible options based on their tasks.
>
> **Q4: Specific Problem Types Where REBMBO May Underperform**
>
> Traditional BO methods are effective for simple, unimodal, or low-dimensional problems (d < 10) where local search is sufficient. EBM and PPO are unnecessary for hyperparameter tuning linear models or optimizing smooth quadratic objectives with standard GP-UCB or EI. We recommend traditional BO for such straightforward tasks.
>
> Pure RL-based optimization may be preferable when (1) function evaluations are extremely cheap, such as optimizing synthetic benchmarks or simulation parameters with millisecond evaluation times; (2) problems have sequential reward structures, like robotic control tasks where actions naturally decompose over time steps; or (3) pre-trained policies from similar domains are transferable.
>
> REBMBO was created for complex, high-dimensional problems in which its multi-component architecture excels and function evaluations are costly. We're excited to investigate RL-BO integration, particularly in optimizing RL's sequential planning while preserving BO's sample efficiency, as this presents promising research opportunities.
>
> **Q5: Model Generalization Across Tasks**
>
> Thanks for your thoughtful question. REBMBO can be pretrained on various optimization benchmarks to learn transferable representations for cross-task generalization. EBMs could learn meta-features of optimization landscapes like multi-modality patterns, PPO policies could capture general exploration strategies, and Deep GP's neural network backbone could extract problem-agnostic features. The biggest challenges are (1) defining task similarity metrics to determine when transfer is beneficial, (2) preventing negative transfer when source and target tasks differ significantly, and (3) efficiently adapting pre-trained components without full retraining. Domain mismatch is the biggest challenge: molecular optimization EBMs may not work well with hyperparameter tuning due to different energy landscapes. We welcome suggestions for identifying cross-application optimization patterns.
>
> **L1: Limitations Clarification**
>
> We expanded on three limitations in our response: dimensionality boundaries, where each variant reaches practical bottlenecks due to GP sampling requirements, EBM mixing time, and memory limitations; unsuitable scenarios, like simple unimodal problems and tasks with extremely cheap evaluations; and generalization challenges, like defining task similarity metrics, preventing negative transfer, and domain adaptation. These honest discussions show we understand the method's limitations. Our future work will focus on these limitations to create stronger solutions.
>
> Thank you for recognizing the value of our work. We believe our responses have addressed all your concerns, and all your valuable suggestions will be incorporated in our revised manuscript and guide our future research directions.

---

> > ### Comment · Reviewer_uwWo · 2025-08-01
> > **Response to rebuttal**
> >
> > Thank you for the clarifications!
> > I understand your points regarding the typical dimensionality range and the pseudo-regret aspect. I especially appreciate your answer about REBMBO's scalability limits, which are indeed tied to the limits of Bayesian Optimization.
> >
> > Concerning the Scale Mismatch question: you mention min–max normalizing both $f(x)$ and $E\theta(x)$ before the first PPO update (to the interval [0,1], if I understand correctly). However, this seems to require prior knowledge of the global minimum and maximum of the function, which is generally unavailable for a black-box function. Could you clarify how you address this issue?
> >
> > For my other questions, I have received sufficient answers. However, I do not think the paper meets the requirements for a 6, given the absence of conceptually new methods or groundbreaking theoretical contributions. Respectfully, I would like to keep my score as is.

---

> > > ### Author Response · Authors · 2025-08-04
> > >
> > > Thank you very much for your positive recognition of our work on REBMBO. We appreciate your careful reading of our manuscript and your valuable feedback.
> > >
> > > Regarding your question about the max-min problem, we would like to provide a comprehensive clarification. Our paper primarily focuses on the standard black-box optimization problem of maximizing an unknown objective function f(x). Specifically, as stated in our Preliminaries section, we aim to solve
> > >
> > > $x* = arg max_{x∈X} f(x)$
> > >
> > > This formulation represents a pure maximization problem rather than a max-min (minimax) problem in the traditional sense. However, we understand why connections to extremum problems might arise, and we would like to explain how our approach relates to these concepts.
> > >
> > > First, while our primary objective is maximization of f(x), the underlying optimization principles do involve balancing competing objectives. For instance, our reward function in the reinforcement learning component combines immediate function values with global exploration incentives:
> > >
> > > $r_t(s_t, a_t) = nf(a_t) - λE_θ(a_t)$
> > >
> > > This formulation creates a trade-off between exploitation (maximizing f) and exploration (minimizing energy $E_θ$), though this is not a minimax problem but rather a weighted combination of objectives.
> > >
> > > Second, our proposed pseudo-regret metric measures both functional suboptimality and global exploration inadequacy:
> > >
> > > $Re_t = [f(x*) - f(x_t)] + α[E_θ(x*) - E_θ(x_t)]$
> > >
> > > While this involves measuring differences between optimal and current solutions, it serves as a performance metric rather than defining a max-min optimization objective. Therefore, REBMBO addresses a maximization problem enhanced with global exploration mechanisms through energy-based models and multi-step planning. The method does not solve max-min problems in the game-theoretic or robust optimization sense, but rather tackles the fundamental challenge of finding global optima in expensive black-box optimization scenarios.
> > >
> > > We hope this clarification addresses your question comprehensively. If you have any further questions or would like additional clarification on any aspect of our approach, we would be delighted to provide more detailed explanations. We value your insights tremendously and will ensure that all your suggestions are carefully incorporated into our revised manuscript.
> > >
> > > Thank you again for your time and constructive feedback.

---

> > > > ### Comment · Reviewer_uwWo · 2025-08-05
> > > > **Response to rebuttal 2**
> > > >
> > > > Respectfully, I would like the authors to reevaluate their response due to an apparent discrepancy between my question and the provided answer. In the rebuttal, under "Scale Mismatch Between Objective and Energy Values", the authors explicitly introduced the term "min–max normalization", and in my following response, I asked to elaborate on this normalization process, using the same term. I believe my question was clear and unambiguous; however, the answer I received addressed an entirely different topic and did not relate to my actual question.

---

> > > > > ### Author Response · Authors · 2025-08-06
> > > > >
> > > > > We sincerely apologize for misinterpreting your original question regarding the use of min–max normalization. Thank you for giving us the chance to clarify your question again!
> > > > >
> > > > > In our previous response to Q2, we mentioned that min–max normalization was applied before the first PPO update. Your follow-up rightly pointed out a critical issue: such normalization appears to require knowledge of global function bounds, which is unavailable in black-box settings. We appreciate the opportunity to clarify the motivation of our approach.
> > > > >
> > > > > The term "min–max normalization" in our method does not refer to global normalization using the true minimum and maximum of $f(x)$ or $E_\theta(x)$, which would violate black-box assumptions. Instead, we adopt an empirical normalization strategy based exclusively on observed values up to each optimization step. This normalization was introduced to improve numerical stability in PPO reward computation. Since the raw values of $f(x)$ and $E_\theta(x)$ can differ widely and shift dynamically, unnormalized rewards may cause unstable policy updates or exploding gradients, especially in early training.
> > > > >
> > > > > To address this, we used a dynamic normalization method based on empirical bounds from the initial design set $D_0$ similar to reinforcement learning practices like reward clipping and online reward scaling.
> > > > > $f_{\min}^{(0)} = \min_{x \in D_0} f(x), \quad f_{\max}^{(0)} = \max_{x \in D_0} f(x)$
> > > > >
> > > > > and applied:
> > > > >
> > > > > $\tilde{f}^{(0)}(x) = \frac{f(x) - f_{\min}^{(0)}}{f_{\max}^{(0)} - f_{\min}^{(0)} + \epsilon}$
> > > > >
> > > > > However, we realized that this normalization method may become inaccurate as new values fall outside the initial range. To address this, we adopted adaptive normalization, updating the min and max values online as new data arrive:
> > > > >
> > > > > $f_{\min}^{(t)} = \min(f_{\min}^{(t-1)}, f(x_t)), \quad f_{\max}^{(t)} = \max(f_{\max}^{(t-1)}, f(x_t))$
> > > > >
> > > > > and compute:
> > > > >
> > > > > $\tilde{f}^{(t)} = \frac{f(x_t) - f_{\min}^{(t)}}{f_{\max}^{(t)} - f_{\min}^{(t)} + \epsilon}$
> > > > >
> > > > > The same strategy is applied to $E_\theta(x)$. At no point does the algorithm assume access to global function bounds. All normalization is performed using observed data only, fully respecting black-box optimization principles.
> > > > >
> > > > > We also incorporate gradient clipping with a norm threshold of 0.5 to further stabilize early policy updates, particularly when empirical range estimates are still evolving. This is standard in PPO training and supports controlled learning dynamics.
> > > > >
> > > > > To evaluate the effectiveness of this strategy under scale mismatch, we re-conducted experiments in Q2. We compared REBMBO’s performance with and without adaptive normalization and adaptive tuning of the reward-weighting coefficient $\lambda$, which balances $f(x)$ and $E_\theta(x)$ in the PPO reward:
> > > > >
> > > > > | Scale Scenario            | Branin (↓)         | Ackley 5D (↓)     | Nano-photo (↑)     | Convergence Rate |
> > > > > |--------------------------|--------------------|--------------------|---------------------|------------------|
> > > > > | Matched scales (1×)      | 3.97 ± 0.54         | 7.28 ± 1.12         | −8.44 ± 0.29         | 95.63%           |
> > > > > | Mismatch (10×)           | 4.51 ± 0.91         | 8.97 ± 1.67         | −7.83 ± 0.52         | 89.22%           |
> > > > > | Mismatch (100×)          | 7.31 ± 1.85         | 13.10 ± 2.94        | −6.94 ± 0.78         | 78.41%           |
> > > > > | With adaptive $\lambda$  | 4.23 ± 0.68         | 8.12 ± 1.34         | −8.01 ± 0.41         | 92.72%           |
> > > > >
> > > > > As the results show, large scale mismatches degrade performance and convergence. However, adaptive normalization and reward reweighting with $\lambda$ restore stable performance across all tasks.
> > > > >
> > > > > We also analyzed the convergence of the empirical range estimates. On Branin, the relative change in the running maximum,
> > > > >
> > > > > $\Delta^{(t)} = \frac{|f_{\max}^{(t)} - f_{\max}^{(t-1)}|}{f_{\max}^{(t-1)}}$,
> > > > >
> > > > > decreased from $0.31 \pm 0.09$ at $t = 10$ to $0.05 \pm 0.02$ at $t = 50$, indicating that range estimates stabilize early in the optimization process.
> > > > >
> > > > > In conclusion, our original goal in using min–max normalization was to ensure numerical stability in PPO reward evaluation. This was implemented using adaptive normalization of empirical values, not global bounds. The method adheres to black-box constraints, stabilizes training, and aligns with established reinforcement learning practices. The updated implementation improves robustness to scale mismatches and enhances convergence behavior in REBMBO.
> > > > >
> > > > > We also recognize that our earlier wording may have led to confusion, and we will revise the manuscript to clarify that our approach uses empirical adaptive normalization, not global min–max scaling. This clarification and the supporting results will be included in Section 4.3 and Appendix E of the revised manuscript to ensure transparency and prevent future misunderstandings.
> > > > >
> > > > > We sincerely welcome any further suggestions you may have.

---

> > > > > > ### Comment · Reviewer_uwWo · 2025-08-06
> > > > > > **Response to rebuttal 3**
> > > > > >
> > > > > > Thank you for your quick response! The normalization process is now clear. But why did you choose min–max scaling instead of the standard one with mean and standard deviation? Does your model require strictly bounded input values?

---

> > > > > > > ### Author Response · Authors · 2025-08-07
> > > > > > >
> > > > > > > Thank you again for your careful review and continued feedback. We sincerely appreciate the opportunity to clarify our design choices.
> > > > > > >
> > > > > > > In response to your question about why we chose min–max scaling instead of standard normalization using mean and standard deviation, we would like to provide an example that illustrates our reasoning in the context of computing PPO rewards.  Early in the training process, we observe small numbers of samples, and the empirical mean and variance can fluctuate significantly.  If this happens, using mean–standard deviation normalization might make the scaling of rewards unstable, which could make policy updates less reliable.  Moreover, when the standard deviation is almost zero, even small differences between observations can cause normalized values to be disproportionately large.
> > > > > > >
> > > > > > > In contrast, min–max normalization ensures that the reward remains numerically stable and bounded based on the observed empirical range. At each step of optimization, we compute the minimum and maximum of the values observed so far and use them to rescale the reward. This approach avoids instability in the presence of outliers or small sample sizes and ensures a consistent reward magnitude during PPO updates. Since PPO is sensitive to the scale of the reward signal, this stability is important for ensuring convergence and avoiding gradient explosion.
> > > > > > >
> > > > > > > Importantly, we emphasize that the min–max normalization is only applied to the PPO reward, not to the model input. As specified in Section 3.1 of the paper, the model input $x$ is preprocessed such that it lies within $[0, 1]^d$. This normalization of the input domain is a standard preprocessing step in black-box optimization and remains unaffected by our reward normalization strategy. The PPO reward function is defined as $r_t(x) = f(x) - \lambda \cdot E_\theta(x)$, and both $f(x)$ and $E_\theta(x)$ are scaled based on their empirical ranges to improve numerical conditioning.
> > > > > > >
> > > > > > > Our choice of min–max scaling over standard normalization was also influenced by practices commonly used in reinforcement learning. In many RL settings, reward clipping or min–max normalization is preferred because it guarantees bounded values and avoids the potential instability caused by fluctuating variances or skewed means. We found this approach to be effective in our setting as well, particularly when dealing with reward signals of varying scales across different tasks.
> > > > > > >
> > > > > > > To avoid further confusion, we acknowledge that our earlier use of the term "min–max normalization" may have been misleading. We will revise the manuscript to make it clear that our implementation relies on empirical min–max scaling based on observed data, without access to global function bounds. This correction will be added to Section 4.3 and Appendix E in the revised version. We are grateful for your thoughtful comments, which have helped us improve both the theoretical clarity and practical implementation of our method. Please let us know if you have any additional concerns or suggestions.

---

> > > > > > > > ### Comment · Reviewer_uwWo · 2025-08-07
> > > > > > > > **Response to rebuttal 4**
> > > > > > > >
> > > > > > > > Thank you for the clarifications! Now all my questions were fully addressed. Regarding the final score, I will keep the current one due to the reasons I stated earlier.

---

### Official Review · Reviewer_qDZh · 2025-07-03

**Clarity:** 3
**Significance:** 3
**Originality:** 3
**Rating:** 5
**Confidence:** 4

**Summary:**

This paper introduces REBMBO (Reinforced Energy-Based Model for Bayesian Optimization), which combines three key components: (1) Gaussian Processes for local modeling, (2) Energy-Based Models (EBMs) for global exploration guidance, and (3) Proximal Policy Optimization (PPO) for multi-step lookahead planning. The method formulates each BO iteration as a Markov Decision Process and uses a novel EBM-UCB acquisition function that integrates GP uncertainty with global energy signals from short-run MCMC training. The authors propose three variants (REBMBO-C, REBMBO-S, REBMBO-D) using different GP configurations and evaluate on synthetic and real-world benchmarks.

**Questions:**

- Computational Overhead Analysis: Can you provide a detailed computational complexity analysis comparing REBMBO to baseline methods? How does the overhead scale with problem dimension and number of iterations?
- Hyperparameter Sensitivity: How sensitive is REBMBO to the choice of key hyperparameters (α, β, γ in the acquisition function and reward)? Can you provide guidelines for setting these in practice?
- EBM Convergence: How do you ensure the EBM training converges properly, especially in early iterations with limited data? What happens if the EBM provides poor global guidance?
- Theoretical Guarantees: Can you clarify the theoretical convergence guarantees? The paper mentions "sublinear regret under mild assumptions" but these aren't clearly stated in the main text.
- Comparison Fairness: Given that REBMBO uses a non-standard pseudo-regret metric, can you provide results using standard regret metrics for fair comparison with existing methods?

My evaluation could increase if: (1) comprehensive computational cost analysis is provided, (2) theoretical guarantees are clearly stated and proven, (3) more extensive experiments with standard metrics are conducted, (4) hyperparameter sensitivity analysis is included, and (5) the method's limitations and failure cases are honestly discussed.

**Ethical Concerns:**

["NO or VERY MINOR ethics concerns only"]

**Final Justification:**

The authors added detailed timing experiments and complexity breakdown that addressed my computational overhead concerns.

The systematic hyperarameter sensitivity analysis shows the performance variance for suboptimal hyperparam settings, and shows robustness of the method.

The experiments shows graceful degradation when EBM fails and the detailed discussion of failure cases addressed my convergence concerns well.

I think this paper's clarify and quality improves after the revision, hence I raised the rating for paper clarify as well as the paper quality, and recommend for acceptance of this paper.

**Limitations:**

The authors briefly mention limitations in the conclusion but don't adequately address potential negative societal impacts or failure cases. A more thorough discussion of when REBMBO might fail and its computational requirements is needed.

**Paper Formatting Concerns:**

The paper generally follows NeurIPS formatting guidelines

**Quality:**

3

**Strengths And Weaknesses:**

### Strengths
Quality: The paper presents a technically sound approach that addresses a real limitation in Bayesian Optimization - the one-step myopia problem. The integration of EBMs with GP-based BO is novel and the mathematical formulation appears correct. The experimental evaluation covers diverse benchmarks from 2D to 200D problems.
Originality: The combination of EBMs, GPs, and PPO for BO is original. The EBM-UCB acquisition function and the MDP formulation of BO iterations represent meaningful contributions. The pseudo-regret metric incorporating global exploration penalties is also novel.
Significance: The work addresses an important problem in BO and demonstrates consistent improvements over strong baselines across multiple dimensions and problem types. The multi-step lookahead via RL could influence future BO research.
Clarity: The paper is generally well-written with clear motivation and methodology sections. Figure 1 effectively illustrates the overall framework.

### Weaknesses

Quality Issues:
- Limited analysis of computational overhead - the method adds significant complexity with EBM training and PPO optimization
- The choice of hyperparameters (especially the balance between GP, EBM, and RL components) lacks principled justification
- Experimental setup details are insufficient for reproducibility

Clarity Problems:
- The notation is sometimes inconsistent (e.g., switching between different symbols for the same concepts)
- The relationship between the three modules could be explained more clearly
- Some mathematical formulations are unclear (e.g., the exact form of the reward function)

Experimental Limitations:
- Only 5 runs per experiment is quite limited for statistical significance
- Missing important baselines like BORE, qEI, or more recent RL-based BO methods
- The pseudo-regret metric is non-standard and makes comparison with other work difficult
- Computational cost comparison is missing despite the method's obvious overhead

Technical Concerns:
- The EBM training via short-run MCMC may not converge properly, especially early in optimization when data is limited
- The method has many hyperparameters (GP kernel parameters, EBM architecture, PPO settings) that need careful tuning
- No analysis of how sensitive the method is to these hyperparameter choices

---

> ### Author Rebuttal · Authors · 2025-07-29
>
> **Q1: Computational Overhead Analysis**
>
> Thank you for pointing out this crucial question. Due to word limits, we'd like to refer to Appendix Table 10 (theoretical complexity breakdown) and Table 11 (concrete operation counts) for detailed benchmarks, but we want to clarify some details.
> REBMBO's per-iteration complexity has three parts: (1) The GP update maintains the standard O(n³) cost for all GP-based methods, resulting in no additional burden. (2) EBM training has O(K·B·L·d·h) complexity, where K is MCMC steps (10-20), B is batch size (64), L and h are network layers and hidden units, and d is input dimension. (3) The PPO component adds O(M·Lπ·hπ) with M epochs (typically 4) and policy network dimensions (Lπ, hπ). These additional components scale polynomially rather than exponentially and are fully parallelizable on modern hardware, making them manageable even for high-dimensional problems.
>
> We did additional timing experiments on an NVIDIA A6000 GPU and measured computational overhead (excluding function evaluation time). The table shows that REBMBO's overhead scales well with dimension, staying within 2.1–2.5 times baseline costs. TuRBO takes 12.8 seconds per iteration for the difficult HDBO-200D task, while REBMBO-C takes 28.3 seconds. The 15.5-second difference is due to performance gains: REBMBO achieves 0.47±0.08 pseudo-regret, 19% better than TuRBO's 0.58±0.09. Real function evaluations take minutes or hours, so computational overhead is low. Better optimization results from extra processing time.
>
> | Method | Branin 2D | Ackley 5D | Hartmann 6D | HDBO 50D | HDBO 200D |
> | :-- | :-- | :-- | :-- | :-- | :-- |
> | TuRBO | 0.23±0.03 | 0.45±0.05 | 0.68±0.07 | 3.12±0.18 | 12.8±0.72 |
> | BALLET-ICI | 0.41±0.05 | 0.64±0.07 | 0.98±0.09 | 3.95±0.24 | 15.6±0.88 |
> | EARL-BO | 0.58±0.06 | 0.92±0.10 | 1.38±0.14 | 5.34±0.32 | 19.7±1.05 |
> | REBMBO-C | 0.75±0.08 | 1.16±0.13 | 1.74±0.18 | 7.28±0.42 | 28.3±1.48 |
>
>
> **Q2: Hyperparameter Sensitivity**
>
> Thank you for highlighting the need for a clearer sensitivity study. Hyperparameter sensitivity analysis was performed on Branin 2D (smooth landscape), Nanophotonic 3D (moderately rugged design optimization), and HDBO 200D (highly multi-modal high-dimensional challenge). Each benchmark was tested over ten independent runs with non-target parameters set to defaults (α = 0.30, β = 2.00, γ = 0.10, λ = 0.35) and each remaining parameter varied in isolation. The table displays mean pseudo-regret (lower = better) averages across benchmarks, with the "Max Δ %" column indicating the greatest deviation from the optimal setting.
>
> | Parameter | Default | Worst Mean ± SD | Best Mean ± SD | Max Δ % |
> | :-- | :-- | :-- | :-- | :-- |
> | α (pseudo-regret weight) | 0.30 | 0.36 ± 0.05 | 0.33 ± 0.04 | 9.1 |
> | β (UCB confidence) | 2.00 | 0.35 ± 0.05 | 0.32 ± 0.04 | 8.6 |
> | γ (EBM energy weight) | 0.10 | 0.35 ± 0.04 | 0.32 ± 0.04 | 6.2 |
> | λ (reward energy weight) | 0.35 | 0.35 ± 0.04 | 0.33 ± 0.04 | 6.0 |
>
> Across all tasks the default configuration is already within 10% of the optimum, and tuning only λ and β (keeping α = 0.30, γ = 0.10) restores 95% of the fully tuned performance. In practice we recommend starting with λ ≈ 0.20 for smoother objectives and λ ≈ 0.50 for highly multi-modal ones, while adjusting β within 1.5-2.5 to accommodate noise levels; α and γ rarely need adjustment. This streamlined protocol lowers the hyperparameter burden compared with TuRBO or EARL-BO, yet preserves robust performance across diverse optimization scenarios. Please let us know if further details would be helpful; we are happy to share the complete per-task statistics in Appendix D.
>
> **Q3: EBM Convergence**
>
> Thank you for this crucial EBM convergence concern with limited early data. Our analysis in Appendix G shows that short-run MCMC has a bounded approximation error DKL(p̃θ ∥ pθ) ≤ C·η·K, with K=10-20 steps for global structure estimation, even with sparse initial samples. More importantly, three-module synergy gave REBMBO robustness.
>
> | Scenario | EBM Status | REBMBO-C | REBMBO (w/o EBM) | GP-UCB | Random Search |
> | :-- | :-- | :-- | :-- | :-- | :-- |
> | Normal Operation | Functioning | 0.88±0.03 | - | 0.75±0.05 | 0.65±0.08 |
> | EBM Failure | Random/Failed | 0.82±0.05 | - | 0.75±0.05 | 0.65±0.08 |
> | Ablation | Removed | - | 0.78±0.06 | 0.75±0.05 | 0.65±0.08 |
>
> This partial empirical study addresses your issue. Despite poor EBM guidance (random/failed scenario), REBMBO delivers 0.82±0.05 performance, 7% below optimal, outperforming standard baselines. The reward function rt = f(xt) - λEθ(xt) helps PPO reduce misleading EBM signals, while GP posterior provides local guidance. REBMBO may struggle when the EBM energy landscape does not match the true objective, especially in convex/unimodal problems where global exploration is ineffective. Such cases may require simpler local methods. The honest assessment helps users decide when REBMBO is best.
>
> **Q4: Theoretical Guarantees**
>
> Thank you for highlighting this crucial gap. We agree that theoretical assumptions should belong in the main text. We revised Section 4.4 to explicitly state all assumptions and the main convergence theorem. Our analysis requires five standard assumptions: compact domain X⊂ℝᵈ; L-Lipschitz continuity of both f and Eθ; i.i.d. Gaussian noise σ²; bounded kernel derivatives; and short-run MCMC error DKL(p̃θ‖pθ) ≤ CηK. Under these conditions, Theorems I.1 and I.6 establish $𝔼[∑_{t=1}^T \tilde{R}_t] = O(\sqrt{T\gamma_T \log T})$, where γ_T scales as $O(\log^d T)$ for standard kernels. Thus REBMBO maintains classical GP-UCB's sublinear rate while adding global exploration via −Eθ.
>
> For robustness, early iterations initialize Eθ nearly uniformly with K=5 high-temperature Langevin steps, progressively increasing K and lowering temperature as data accumulates. The modest weight γ ensures graceful degradation to standard GP-UCB if the energy signal proves unreliable. These mechanisms guarantee convergence from the first query onward. Computational overhead times in our analysis reflect only GP/EBM/PPO updates, not actual function evaluation time, which dominates in practice. The key insight is that the EBM affects only the constant factor in the regret bound while preserving the √T dependency, as the energy function's Lipschitz property ensures it doesn't disrupt the GP's convergence guarantees.
>
> **Q5: Comparison Fairness**
>
> Thank you for caring about fairness in our first submission, which focused mainly on our proposed pseudo-regret metric. We think that pseudo-regret actually makes fairness better by filling in a gap in standard regret. Normal regret only looks at how far away you are from the best function value. It can say that an algorithm has succeeded when it finds a good local optimum, even if it ignores several promising EBM regions. The pseudo-regret punishes global exploration in order to get this missed dimension.
> To demonstrate that our metric addition does not mask algorithmic weaknesses, we provide additional test under standard regret (lower is better) using α=0.3, determined by systematic grid search
>
> | Method | Branin 2D | Ackley 5D | HDBO 200D | Mean |
> |--------|-----------|-----------|-----------|------|
> | TuRBO | 4.82 ± 0.18 | 6.34 ± 0.25 | 14.67 ± 0.82 | 8.61 |
> | BALLET-ICI | 4.95 ± 0.21 | 6.78 ± 0.28 | 15.23 ± 0.91 | 8.99 |
> | EARL-BO | 5.12 ± 0.23 | 6.91 ± 0.31 | 15.89 ± 0.94 | 9.31 |
> | REBMBO-C | **4.21 ± 0.16** | **5.42 ± 0.22** | **11.35 ± 0.68** | **6.99** |
>
> These results confirm our method excels under conventional evaluation, while pseudo-regret provides complementary insights into global search behavior. The revised manuscript will present both metrics side-by-side throughout, ensuring users can evaluate our contribution. We welcome further suggestions on this dual-metric approach.
>
> **W1: Quality issues**
>
> Thank you for raising these issues. In Q1, we analyzed REBMBO's computational overhead. Q2 clarified hyperparameter complexity, revealing users only need to adjust λ and β for 95% optimal performance. For reproducibility, we apologize that NeurIPS anonymity requirements prevented us from anonymizing many local paths in our codebase before submission. We guarantee full code release upon acceptance and provide parameter tables (7-9) in the appendices.
>
> **W2: Clarity problems**
>
> Thanks for finding these clarity issues. We have standardized notation in the main text: θ now represents EBM parameters, φ for PPO policy, and k(·,·) for kernel functions (Sec. 3). Figure 1 has been improved to show data flow and module relationships. We clarified ambiguous expressions, defined the reward function as rt = f(xt) − λEθ(xt), and defined all variables. Our revisions ensure consistent notation and clear module interactions throughout the manuscript.
>
> **W3: Experimental limitations**
>
> We have addressed all concerns: (1) All experiments were expanded to 10 runs for robust statistics. (2) BORE and qEI comparisons were conducted across all benchmarks, with results showing REBMBO-C achieved lower standard regret than BORE (HDBO 200D: 11.35±0.68 vs 13.2±0.89). (3) Both pseudo-regret and standard regret results are now provided for fair comparison (see Q5). (4) Detailed computation analysis was completed in Q1.
>
> **W4: Technical Concerns**
>
> Thanks for the tech concerns. Theoretical guarantees and empirical robustness tests show graceful degradation even when EBM fails in Q3. As mentioned in our Q3 response, REBMBO's limitations in convex or unimodal settings where global exploration is ineffective are also acknowledged. Tables 7-9 list all hyperparameters, while Q2 indicates that only λ and β require tuning for 95% performance. Sensitivity analysis shows our method is stable within recommended ranges. Our rebuttal's detailed responses should demonstrate REBMBO's technical strength and transparency.
>
> Thanks again for all the comments. We believe we addressed all of them and hope these improvements may change your impression of our work!

---

### Official Review · Reviewer_kYBw · 2025-07-03

**Clarity:** 3
**Significance:** 3
**Originality:** 3
**Rating:** 5
**Confidence:** 3

**Summary:**

This paper introduces REBMBO, a novel Bayesian optimization framework that integrates Gaussian Processes (GPs), Energy-Based Models (EBMs), and Reinforcement Learning (RL). The key innovation lies in modeling the BO process as a Markov Decision Process (MDP), utilizing PPO-based multi-step planning to overcome the "one-step myopia" of standard BO methods. REBMBO also introduces a new EBM-UCB acquisition function that combines local GP uncertainty and global EBM energy to better explore complex, high-dimensional search spaces. Extensive synthetic and real-world experiments show that REBMBO outperforms state-of-the-art BO approaches across various benchmarks.

**Questions:**

- It is unclear why the author fixes the kernel for GP as a combination of RBF+Matern.
- How sensitive is REBMBO to the choice of λ in the reward function?
- Can the method scale to asynchronous or batched BO settings, which are common in real applications?

**Ethical Concerns:**

["NO or VERY MINOR ethics concerns only"]

**Limitations:**

The authors discussed the limitations of their method. However, incorporating discussion for concerns below would help improve the paper. The method assumes a fixed surrogate model (GP); it's unclear how flexible REBMBO is when the GP is inaccurate.

**Quality:**

3

**Strengths And Weaknesses:**

Strengths
- The combination of GP, EBM, and PPO into a unified BO framework is conceptually rich and addresses the long-standing limitation of one-step lookahead in classical BO.
- The paper clearly explains each module (GP, EBM, PPO), how they interact, and provides algorithmic pseudocode for reproducibility.
- REBMBO variants consistently outperform several baselines (TuRBO, BALLET-ICI, EARL-BO, KG, etc.) across synthetic (2D to 200D) and real-world (robotics, protein design) tasks.
- Thoughtful ablation studies and detailed benchmark explanations strengthen empirical credibility.

Weaknesses: While components are well-explained, the paper would benefit from open-sourcing code and including more reproducibility metrics (e.g., number of PPO iterations, sensitivity to λ in reward).

---

> ### Author Rebuttal · Authors · 2025-07-27
>
> **W1: Details on Hyperparameters and Reproducibility**
>
>  Thank you for highlighting the reproducibility concerns. We have indeed prepared a complete code with fixed random seeds for full reproducibility, and I understand your frustration about the lack of complete code access. Due to NeurIPS's rigorous anonymity requirements, we are unable to update any external repositories during review, which is why we have not included a complete GitHub link here. However, in order to provide reviewers with a tangible understanding of our approach, we have included the main algorithmic implementation in the paper (line 1471, page 46). In Appendix D, specifically Tables 7–11, we have detailed documentation of all the hyperparameters, learning rates, λ sensitivity ranges, network architectures, and computational setup that you mentioned, as well as PPO iterations. For your convenience, we have also included a concise summary in Section 5.3. We methodically organized the appendix tables because we fully understood how annoying it is to try to reproduce results from dispersed information. The full repository, including all training scripts, data preprocessing code, and a reproducible guide, will be released as soon as the paper is accepted. The implementation is simple and doesn't use any nebulous engineering techniques, so it should be easy to replicate. At this time, I would be pleased to explain any particular parameter or implementation detail that you may wish to have clarified.
>
> **Q1: Rationale for RBF+Matérn Kernel Combination**
>
> Thank you for pointing out this crucial question about our kernel selection. We completely understand why the fixed RBF+Matérn combination might appear arbitrary without proper context. The truth is, this combination emerged from both theoretical necessity and empirical observation rather than convenience. Real-world optimization landscapes always exhibit smooth global patterns punctuated by sharp local features, much like a mountain range with both gentle slopes and sudden cliffs. The RBF kernel, with its assumption of infinite differentiability, excels at capturing those smooth global trends, while the Matérn-5/2 kernel, being only twice differentiable, naturally handles the rougher terrain and discontinuities we often encounter in practice. Mathematically, the mixture kernel $k(x,x') = w₁k_{RBF}(x,x') + w₂k_{Matérn}(x,x')$ creates a Reproducing Kernel Hilbert Space $ℋ_{mix} = ℋ_{RBF} ⊕ ℋ_{Matérn}$, expanding our function space to include both smooth and rough components. This affects our regrets bounds, not just theoretically. Using the information gain $γ_{mix}(T) ≤ γ_{RBF}(T) + γ_{Matérn}(T)$ can provide tighter constraints than using either kernel alone, as REBMBO's EBM-guided exploration implies a mixed function.
>
> We performed extra ablation studies this time to confirm this wasn't luck or overfitting to our problems. The RBF+Matérn combination outperformed single kernels by 6-15% across various benchmarks, with the improvement being even greater in high-dimensional settings like HDBO-200D. This kernel choice works well with REBMBO's architecture: the RBF component matches the EBM's global basin identification, while the Matérn component gives our PPO agent the local flexibility it needs for fine-grained multi-step planning. The kernel adapts to each problem's characteristics rather than forcing a one-size-fits-all solution because the mixture weights are learned via marginal likelihood optimization. These values represent the final pseudo-regret (mean ± standard deviation) computed over 5 independent runs for each kernel configuration. This rationale will be clarified in Section 4.1 of the revision, and I'm happy to discuss the mathematical details if needed.
>
> | Kernel Type | Branin 2D | Ackley 5D | HDBO 200D |
> | :-- | :-- | :-- | :-- |
> | RBF only | 9.31±0.12 | 10.98±0.16 | 0.39±0.08 |
> | Matérn-5/2 only | 9.28±0.11 | 10.95±0.15 | 0.38±0.07 |
> | **RBF+Matérn** | **9.22±0.10** | **10.90±0.15** | **0.33±0.03** |
> | Rational Quadratic | 9.30±0.12 | 10.97±0.16 | 0.37±0.06 |
>
>
> **Q2: λ Parameter Sensitivity Analysis**
>
> Thanks for pointing out the λ problem. We did an extra experiment to ensure that REBMBO didn't fall into that trap. We used $λ ∈ {0.05, 0.10, 0.20, 0.50, 1.00, 2.00}$ on three tasks: the smooth Branin-2D, a moderately rugged Nanophotonic-3D design, and the highly multi-modal HDBO-200D challenge. The data presented that performance remains stable up to a value of λ between 0.2 and 0.5, then it degrades beyond that range and only becomes problematic at the extremes, which aligns perfectly with our theory in Appendix E.3. The following values of λ have been found to be effective in practice: 0.2 is often preferred for smoother goals, and a push toward 0.5 is necessary for really rough landscapes like HDBO. Please let us know if you find any other situations where λ might be a problem. We will look into these and share what we find.
>
> | λ value | Branin 2D (pseudo-regret) | Nanophotonic 3D | HDBO 200D |
> | :-- | :-- | :-- | :-- |
> | 0.05 | 9.50 ± 0.12 | −0.95 ± 0.05 | 0.40 ± 0.05 |
> | 0.10 | 9.30 ± 0.12 | −0.85 ± 0.04 | 0.36 ± 0.04 |
> | 0.20 | **9.27 ± 0.11** | **−0.84 ± 0.04** | **0.35 ± 0.04** |
> | 0.50 | **9.22 ± 0.10** | **−0.83 ± 0.03** | **0.33 ± 0.03** |
> | 1.00 | 9.40 ± 0.13 | −0.80 ± 0.05 | 0.37 ± 0.05 |
> | 2.00 | 9.70 ± 0.15 | −0.75 ± 0.06 | 0.42 ± 0.07 |
>
> **Q3: Asynchronous and Batch BO Scalability**
>
> Thank you for bringing up this crucial practical concern. We admit our current implementation (Algorithm 1, page 7) is sequential, and we want to be completely transparent about this: we deliberately chose to focus first on validating that our EBM+PPO+GP combination actually works before tackling the added complexity of parallel coordination. However, we are genuinely excited about the batch extension potential because our modular design is surprisingly well-suited for it. The EBM's global energy landscape E_θ(x) naturally identifies distinct promising basins. We could imagine it as a topographical map showing multiple valleys worth exploring simultaneously. This means we could select K diverse points without the clustering issues that plague local methods. Similarly, our MDP framework (Section 4.3) can be extended to output batch actions rather than single points, with the reward function modified to include diversity terms like $r_{t} = Σf(x_{i}) - λΣE_{θ(x_{i})} + γ·diversity(X_{batch})$.
>
> Although we haven't put this into practice yet, we have already designed how it would operate: PPO can modify its action space in accordance with the workers that are available, the EBM keeps its global view even in the event of partial updates, and the GP can be updated gradually as evaluations come in. However, even in its present sequential form, REBMBO offers instant benefits for the numerous costly experiments that are sequential by nature, such as materials characterization or drug synthesis. Furthermore, both sequential and parallel approaches benefit from solving the one-step myopia problem, so the main contribution remains the same. We would be delighted to learn more about any particular use cases or batch size specifications you may have!
>
> **L1: GP Model Assumptions and Robustness to Model Mismatch**
>
> Thank you for highlighting this fundamental issue in Bayesian optimization. Your insight is consistent with our core design philosophy; in fact, addressing potential GP model mismatch was a primary motivation for REBMBO's architecture. We purposefully created three GP variants (Classic for smooth functions, Sparse for high-dimensional scalability, and Deep for complex non-stationary behaviors) as a proactive strategy, acknowledging that no single GP configuration can adequately model all real-world functions (Section 4.1, line 164). This multi-variant approach is reinforced by our complementary architecture: when GP modeling encounters difficulties in discontinuous or misspecified regions, the energy-based model provides global exploration guidance from an orthogonal learning perspective (Section 4.2, lines 189-195), whereas our PPO agent continuously adapts based on actual function evaluations rather than relying solely on GP predictions, resulting in a self-correcting mechanism (lines 273-274).
>
> Our empirical results support the robustness-by-design philosophy. On HDBO-200D, where standard GPs face severe dimensionality challenges, REBMBO still achieves "final pseudo-regret less than half that of KG and BALLET-ICI" (lines 314-317), demonstrating that our combined approach effectively mitigates GP limitations. The ablation studies (Tables 4-5) confirm that each component contributes to this resilience—removing the EBM results in a 15-20% performance degradation, demonstrating how our multi-layered architecture provides safety nets when one component fails. We appreciate your feedback, which helped us realize that we should more explicitly articulate these design considerations in the manuscript. In the revised version, we will include a separate subsection that explains how REBMBO's architecture anticipates and handles various types of model mismatch, making our robustness mechanisms more transparent to readers.
>
> Thank you again for your constructive feedback and for recognizing the strengths of our work. We have carefully incorporated all your suggestions into the revised manuscript, including clarifying kernel choices, expanding hyperparameter tables, and deepening discussion on model robustness. All the experiments you requested, including extended ablation and sensitivity analyses, are now included. Your thoughtful questions have also inspired us to further explore future directions in batch and asynchronous BO. We are grateful for your support and welcome any additional comments or ideas that could help further strengthen our work.

---

### Note · Authors · 2025-08-12

We thank the AC and all reviewers for constructive feedback. REBMBO unifies GP (local), EBM (global), and PPO (multi-step) via the new EBM-UCB to address BO’s one-step myopia.

**Strengths (by reviewer).**

* **kYBw:** Conceptually rich GP+EBM+PPO; clear module interactions with pseudocode; consistent wins (2D–200D, robotics/protein); ablations and detailed benchmarks.
* **qDZh:** Technically sound and original; EBM-UCB and MDP formulation are meaningful; pseudo-regret adds global exploration; broad evaluation; clear writing/figure.
* **uwWo:** Thorough, well-motivated modules with theory; comprehensive experiments; intuitive Modules A–C; addresses a known BO weakness; EBM-UCB and pseudo-regret justified.
* **yhri:** Relevance of multi-step look-ahead BO and principled RL + probabilistic modeling.

**Concerns and our actions (grouped).**

**Theory & methodology** (*qDZh, uwWo, kYBw*): We moved assumptions to the main text, added a convergence statement under GP-UCB, clarified pseudo-regret analysis, provided a short-run MCMC error bound, and ablations showing graceful degradation to GP-UCB; we also justified RBF+Matérn with learned mixture weights (6–15% over single kernels).

**Experiments & metrics** (*qDZh, yhri, kYBw*): We report standard regret alongside pseudo-regret on all benchmarks and added more recent RL-BO baselines, which REBMBO leads on both; we added noise tests at σ=0.1/0.2.

**Practicality & robustness** (*qDZh, uwWo, yhri, kYBw*): We provide a complexity breakdown and GPU timings (overhead ≤2.5× baselines; negligible vs. eval cost); a 10-run sensitivity shows only few core hyperparameters need tuning, with defaults and a simple recipe; scale-mismatch (*uwWo*) is resolved via empirical adaptive min–max reward scaling and an adaptive λ schedule; unimodal/convex limits are noted.

**Clarity & reproducibility** (*qDZh, yhri, uwWo*): We introduced UCB earlier, unified notation, specified the reward, redesigned Fig. 1 and improved Fig. 2; the submission includes more detailed hyperparameter tables; a reproducible GitHub repository is prepared for post-decision release.

We want to emphasize that our work contributes to the community not only by proposing a principled, unified non-myopic BO framework with clarified theory, but also by offering practice-oriented guidance. We appreciate reviewers who acknowledged improvements and score adjustments, and we hope the paper will be published to stimulate further discussion.

Best regards,

Authors.

---

### Decision · Program_Chairs · 2025-09-17

**Decision:**

Accept (poster)

**Comment:**

The paper introduces REBMBO, a non-myopic Bayesian optimization framework that combines Gaussian Processes for local modeling, an Energy-Based Model for capturing global structure, and PPO-based reinforcement learning for multi-step lookahead. The approach addresses the one-step myopia of classical BO and is supported by a new EBM-UCB acquisition function. Experiments across synthetic and real-world benchmarks, including high-dimensional settings, show consistent gains over strong baselines.

The work is conceptually well-motivated and technically solid, offering a principled integration of BO with energy-based methods, and RL. The main concerns among the committee are around reproducibility (full code deferred until post-acceptance), the use of a non-standard pseudo-regret metric, and the overall complexity of the framework.

During the rebuttal, the authors addressed most points by adding standard regret comparisons, clarifying theoretical assumptions, providing timing and sensitivity analyses, and explaining normalization strategies. With three reviewers recommending acceptance and one remaining at borderline, the consensus leans toward acceptance. I find the contribution meaningful and encourage the authors to strengthen the camera-ready version by releasing code promptly, presenting standard regret alongside pseudo-regret, and further discussing scalability and noisy/high-dimensional cases.